# INCOMPLETE DATA, COMPLETE DYNAMICS:
# A DIFFUSION APPROACH

**Zihan Zhou**[1,2]**, Chenguang Wang**[1,2]**, Hongyi Ye**[1]**, Yongtao Guan**[1]**, Tianshu Yu**[1,2*]
[1]School of Data Science, The Chinese University of Hong Kong, Shenzhen
[2]Shanghai Artificial Intelligence Laboratory
`{zihanzhou1, chenguangwang, hongyiye}@link.cuhk.edu.cn`
`{guanyongtao, yutianshu}@cuhk.edu.cn`

## ABSTRACT

Learning physical dynamics from data is a fundamental challenge in machine learning and scientific modeling. Real-world observational data are inherently incomplete and irregularly sampled, posing significant challenges for existing data-driven approaches. In this work, we propose a principled diffusion-based framework for learning physical systems from *incomplete training samples*. To this end, our method strategically partitions each such sample into observed context and unobserved query components through a carefully designed splitting strategy, then trains a conditional diffusion model to reconstruct the missing query portions given available contexts. This formulation enables accurate imputation across arbitrary observation patterns without requiring complete data supervision. Specifically, we provide theoretical analysis demonstrating that our diffusion training paradigm on incomplete data achieves asymptotic convergence to the true complete generative process under mild regularity conditions. Empirically, we show that our method significantly outperforms existing baselines on synthetic and real-world physical dynamics benchmarks, including fluid flows and weather systems, with particularly strong performance in limited and irregular observation regimes. These results demonstrate the effectiveness of our theoretically principled approach for learning and imputing partially observed dynamics.

## 1 INTRODUCTION

Learning physical dynamics from observational data represents a cornerstone challenge in machine learning and scientific computing, with applications spanning weather forecasting (Conti, 2024; Zhang et al., 2025b), fluid dynamics (Wang et al., 2024; Brunton & Kutz, 2024), biological systems modeling (Qi et al., 2024; Goshisht, 2024), and beyond. Classical physics-based approaches require explicit specification of governing equations and boundary conditions, while data-driven methods offer the promise of discovering hidden dynamics directly from observations (Luo et al., 2025; Meng et al., 2025). However, a fundamental bottleneck persists: real-world observational data are inherently incomplete, irregularly sampled, and subject to various forms of missing information, making it difficult for existing approaches to learn accurate representations of the underlying dynamics.

**Inherent sparsity of physical measurements.** Physical science data fundamentally differs from typical computer vision datasets. Unlike natural images, where complete pixel grids are the norm, real-world physical measurements are inherently sparse and incomplete. Sensor networks provide observations only at discrete spatial locations, satellite imagery suffers from cloud occlusion, and experimental measurements are constrained by instrumental limitations. This incompleteness is not a temporary inconvenience to be resolved through better data collection—it is an intrinsic characteristic of how we observe physical systems.

**Structured observation patterns.** Prior approaches to learning from incomplete physical data have largely adopted simplistic assumptions about observation patterns. Most existing methods assume pixel-level independent and identically distributed (i.i.d.) missing patterns, where each spatial

---
*corresponding author

location has an equal probability of being observed (Daras et al., 2023; Dai et al., 2024; Simkus & Gutmann, 2025). While some recent works have explored alternative missing patterns in their experimental evaluations, such as row/column missing for tabular data, they still employ the same training strategies regardless of the observation structure (Ouyang et al., 2023). This one-size-fits-all approach fails to leverage the specific characteristics of different mask distributions. In reality, observation patterns exhibit strong spatial structure: weather stations capture measurements within their local coverage areas, creating contiguous blocks of observations; satellite instruments observe swaths determined by orbital paths; underwater sensor arrays monitor volumes dictated by acoustic propagation. These structured patterns fundamentally differ from random pixel dropout and demand specialized training strategies. Our work addresses this gap by developing context-query partitioning strategies specifically tailored to the underlying mask distribution, ensuring effective learning across diverse observation patterns.

**Lack of theoretical foundations.** While recent works have proposed various heuristic approaches for handling missing data in generative modeling, they lack rigorous theoretical foundations. Existing methods typically rely on empirical design choices without providing convergence guarantees or understanding of learning dynamics (Ouyang et al., 2023; Daras et al., 2023; Dai et al., 2024; Barth et al., 2024; Simkus & Gutmann, 2025; Majid et al., 2026). Moreover, some theoretically-motivated approaches suffer from prohibitive computational costs, requiring multiple complete model retraining cycles or complex importance weighting schemes that limit their applicability to low-dimensional toy problems (Chen et al., 2024b; Givens et al., 2025; Zhang et al., 2025a).

**Our solution.** To address these challenges, we develop a theoretically principled diffusion-based framework that provides rigorous convergence guarantees while maintaining computational efficiency for high-dimensional physical dynamics problems. Our approach answers critical questions about whether diffusion models trained solely on incomplete data can recover complete data distributions, how observation patterns affect diffusion training efficiency, and under what conditions successful reconstruction of unobserved regions is guaranteed. In summary, our contributions are:

- **Methodical design**: We propose a novel conditional diffusion training paradigm that works directly with incomplete training samples, featuring a strategically designed context-query partitioning scheme tailored for physical dynamics.

- **Theoretical guarantee**: We provide the first theoretical analysis proving that diffusion-based training on incomplete data with our paradigm asymptotically recovers the true complete dynamical process under mild regularity conditions.

- **Strong results**: We conduct comprehensive experiments on both synthetic and real-world datasets, demonstrating substantial improvements in imputation accuracy over competitive baselines, particularly in challenging sparse observation regimes.

## 2 PRELIMINARIES

In Appendix A, we present a review of imputation methods and generative modeling approaches for missing data, which provides the broader context for our contributions. We also provide a detailed introduction to diffusion models in Appendix B, covering both standard *noise matching* and the *data matching* formulation that our method primarily employs. In this section, we formally define the problem of learning physical dynamics from incomplete observations. We establish the mathematical framework and notation that will be used throughout the paper.

We formalize the problem as follows. Let $\mathcal{X} \subset \mathbb{R}^d$ denote the space of complete data samples following an unknown distribution $p_{\text{data}}(\boldsymbol{x}_0)$. Binary masks $\boldsymbol{M} \in \{0, 1\}^d$ are drawn from distribution $p_{\text{mask}}(\boldsymbol{M})$, where 1 indicates observed elements. We assume that masks are conditionally independent of the data given the observation process: $p_{\text{mask}}(\boldsymbol{M} \mid \boldsymbol{x}_0) = p_{\text{mask}}(\boldsymbol{M})$. In practice, we have prior knowledge about the mask distribution $p_{\text{mask}}(\boldsymbol{M})$ based on the data collection process (e.g., sensor placement patterns, measurement protocols).

For each training instance $i$, we have $\boldsymbol{x}_{\text{obs}}^{(i)} = \boldsymbol{M}^{(i)} \odot \boldsymbol{x}_0^{(i)}$ denoting partially observed data and $\boldsymbol{x}_{\text{unobs}}^{(i)} = (1 - \boldsymbol{M}^{(i)}) \odot \boldsymbol{x}_0^{(i)}$ representing missing values. Crucially, our training dataset

$\mathcal{D} = \{(\boldsymbol{x}_{\text{obs}}^{(i)}, \boldsymbol{M}^{(i)})\}_{i=1}^{N}$ contains only partial observations, no complete samples $\boldsymbol{x}_0^{(i)}$ are available during training. This setting reflects realistic scenarios where complete ground truth is unavailable. The objective is to learn a conditional generative model $p_\theta(\boldsymbol{x}_0 \mid \boldsymbol{x}_{\text{obs}}, \boldsymbol{M})$ that generates complete samples consistent with the observed elements, despite being trained solely on incomplete data.

## 3 METHOD

In this section, we present our approach for learning physical dynamics directly from incomplete observations using diffusion models. Our method addresses the fundamental challenge of training generative models when both training and test data are partially observed, without access to complete ground truth during training. Our approach consists of three key components: **(1) Denoising data matching on incomplete training data** (Sec. 3.1): We formulate a theoretically grounded training loss and establish conditions under which the model learns meaningful conditional expectations for all dimensions. **(2) Strategic context-query partitioning** (Sec. 3.2): We develop a principled strategy for partitioning incomplete samples into context and query components, enabling reconstruction of originally missing dimensions. **(3) Ensemble sampling for complete data reconstruction** (Sec. 3.3): We bridge the gap between training on context masks and inference on full observations through ensemble averaging with theoretical convergence guarantees. This unified framework enables robust learning from incomplete observations while providing strong theoretical foundations for reconstructing complete data from partial observations.

### 3.1 DENOISING DATA MATCHING ON INCOMPLETE TRAINING DATA

The fundamental challenge in learning from incomplete data is ensuring that training on incomplete observations enables the model to recover the complete underlying data distribution, despite never having access to complete samples during training. To address this, we formulate a training objective that learns a conditional generative model $p_\theta(\boldsymbol{x}_0 \mid \boldsymbol{x}_{\text{obs}}, \boldsymbol{M})$[1] that generates complete samples consistent with the observed elements, despite being trained solely on incomplete data. Our approach strategically partitions incomplete samples and deliberately withholds information during training through a theoretically grounded framework.

Our key insight is to reframe the learning problem through a hierarchical masking strategy. For each incomplete sample $(\boldsymbol{x}_{\text{obs}}, \boldsymbol{M})$ in our training dataset, we treat the partially observed data $\boldsymbol{x}_{\text{obs}}$ as "complete" within the scope of available observations. We then sample context masks $\boldsymbol{M}_{\text{ctx}} \subseteq \boldsymbol{M}$ to represent "observable" portions and query masks $\boldsymbol{M}_{\text{qry}} \subseteq \boldsymbol{M}$ to represent "query" portions relative to $\boldsymbol{x}_{\text{obs}}$. Given the noisy sample $\boldsymbol{x}_{\text{obs},t} = \boldsymbol{M} \odot (\alpha_t \boldsymbol{x}_{\text{obs}} + \sigma_t \boldsymbol{\epsilon})$ at time $t$ in the diffusion process, we train the neural network $\boldsymbol{x}_\theta$ to predict the *complete* clean data $\boldsymbol{x}_0$ from the timestep $t$, context-masked noisey observations $\boldsymbol{M}_{\text{ctx}} \odot \boldsymbol{x}_{\text{obs},t}$, and context mask $\boldsymbol{M}_{\text{ctx}}$, with our training loss is formulated as:

$$\mathcal{L}(t, \boldsymbol{x}_{\text{obs}}, \boldsymbol{M}_{\text{ctx}}, \boldsymbol{M}_{\text{qry}}) = \|\boldsymbol{M}_{\text{qry}} \odot (\boldsymbol{x}_\theta(t, \boldsymbol{M}_{\text{ctx}} \odot \boldsymbol{x}_{\text{obs},t}, \boldsymbol{M}_{\text{ctx}}) - \boldsymbol{x}_{\text{obs}})\|^2 \tag{1}$$

The key architectural choice is that the model receives only the context mask $\boldsymbol{M}_{\text{ctx}}$ and corresponding observed values, trying to provide the best estimate for the queried dimensions. A natural question arises:

*Since we train exclusively on incomplete data, how can this training loss enable the model to predict the originally missing portions of the data—regions that were never observed during training?*

Through the following theoretical analysis, we provide key insights into how training on incomplete observations can still lead to models capable of reconstructing complete data distributions.

**Theorem 1** (Optimal solution under context masking without query information). *Let $\boldsymbol{x}_\theta^*$ be the optimal solution by minimizing the loss in equation 1. Under the conditional independence of masks and data, we have the following results:*

---

[1]For clarity, we note that in the full diffusion imputation setting, the ideal model would output $\mathbb{E}[\boldsymbol{x}_0 \mid \boldsymbol{x}_t, \boldsymbol{x}_{\text{obs}}, \boldsymbol{M}]$, conditioning on both the noisy state $\boldsymbol{x}_t$ and observations $\boldsymbol{x}_{\text{obs}}$. Our single-step sampling approach (Sec. 3.3) simplifies this by using minimal noise ($t = \delta \approx 0$), effectively approximating $\mathbb{E}[\boldsymbol{x}_0 \mid \boldsymbol{x}_{\text{obs}}, \boldsymbol{M}]$. For cases requiring diversity generation, we provide a multi-step sampling procedure in Appendix E that combines both sources of information through weighted averaging.

*(i) Optimal solution: The optimal solution is given by*

$$(\boldsymbol{x_\theta}\left(t, \boldsymbol{M}_{ctx} \odot \boldsymbol{x}_{obs,t}, \boldsymbol{M}_{ctx}\right))_i = \begin{cases} \mathbb{E}\left[(\boldsymbol{x}_0)_i \mid \boldsymbol{M}_{ctx} \odot \boldsymbol{x}_{obs,t}, \boldsymbol{M}_{ctx}\right], & P((\boldsymbol{M}_{qry})_i = 1 \mid \boldsymbol{M}_{ctx}) > 0 \\ \textit{an arbitrary value}, & P((\boldsymbol{M}_{qry})_i = 1 \mid \boldsymbol{M}_{ctx}) = 0 \end{cases} \quad (2)$$

*where $i$ indicates the $i$-th entry of the vector. Specially, given the context mask $\boldsymbol{M}_{ctx}$, if the union of all possible query mask $\boldsymbol{M}_{qry}$ supports covers all spatial dimensions, we have*

$$\boldsymbol{x_\theta}\left(t, \boldsymbol{M}_{ctx} \odot \boldsymbol{x}_{obs,t}, \boldsymbol{M}_{ctx}\right) = \mathbb{E}\left[\boldsymbol{x}_0 \mid \boldsymbol{M}_{ctx} \odot \boldsymbol{x}_{obs,t}, \boldsymbol{M}_{ctx}\right] \quad (3)$$

*(ii) Gradient magnitude scaling: The expected squared gradient magnitude with respect to the network output for dimension $i$ scales linearly with the query probability $p_i := P((\boldsymbol{M}_{qry})_i = 1 \mid \boldsymbol{M}_{ctx})$:*

$$\mathbb{E}\left[\left(\frac{\partial \mathcal{L}}{\partial (\boldsymbol{x_\theta})_i}\right)^2\right] = 4 p_i \mathbb{E}\left[\left((\boldsymbol{x_\theta})_i - (\boldsymbol{x}_{obs})_i\right)^2 \Big| (\boldsymbol{M}_{qry})_i = 1\right] \quad (4)$$

*(iii) Parameter update frequency: The frequency of non-zero parameter updates for dimension $i$ is exactly $p_i$:*

$$P(\textit{dimension } i \textit{ contributes to parameter update}) = P((\boldsymbol{M}_{qry})_i = 1 \mid \boldsymbol{M}_{ctx}) = p_i \quad (5)$$

The proof can be found in Appendix F. This theorem reveals a critical insight: the model learns meaningful conditional expectations $\mathbb{E}\left[(\boldsymbol{x}_0)_i \mid \boldsymbol{M}_{ctx} \odot \boldsymbol{x}_{obs,t}, \boldsymbol{M}_{ctx}\right]$ for dimension $i$ only when $P((\boldsymbol{M}_{qry})_i = 1 \mid \boldsymbol{M}_{ctx}) > 0$. When this probability is zero, the model's output for dimension $i$ becomes arbitrary since it never receives gradient updates for that dimension. This theoretical finding directly drives the need for strategic context-query partitioning: given any context mask $\boldsymbol{M}_{ctx}$ (without information of $\boldsymbol{M}$), we must ensure that every dimension outside the context, including dimensions that were originally missing in the raw data (i.e., dimensions $i$ where $\boldsymbol{M}_i = 0$), has a positive probability of being selected as a query point. In the following section, we detail how to design the context-query mask sampling strategy to guarantee this requirement.

## 3.2 STRATEGIC CONTEXT-QUERY PARTITIONING

Building on our theoretical analysis, we now address the crucial question: how should we design the context-query partitioning strategy to ensure effective learning? Our approach is guided by a principled design framework that guarantees positive query probabilities for all observable dimensions.

**Design principle.** Based on Theorem 1, we establish the core design principle for effective context-query partitioning:

**Principle 1** (Principle of uniform query exposure). *For effective learning from incomplete data, the context-query partitioning strategy must satisfy:*

1. **Non-zero query probability:** *For all unobserved dimensions $i$, i.e., $(\boldsymbol{M}_{ctx})_i = 0$,*

$$P((\boldsymbol{M}_{qry})_i = 1 \mid \boldsymbol{M}_{ctx}) > 0 \quad (6)$$

2. **Uniform exposure:** *The query probabilities should be approximately uniform across all observed dimensions to achieve balanced learning:*

$$P((\boldsymbol{M}_{qry})_i = 1 \mid \boldsymbol{M}_{ctx}) \approx P((\boldsymbol{M}_{qry})_j = 1 \mid \boldsymbol{M}_{ctx}) \quad \forall i, j : (\boldsymbol{M}_{ctx})_i = (\boldsymbol{M}_{ctx})_j = 0 \quad (7)$$

To implement this principle, we can decompose the query probability using the law of total probability over all possible observation masks:

$$P((\boldsymbol{M}_{qry})_i = 1 \mid \boldsymbol{M}_{ctx}) = \sum_{\boldsymbol{M}} P((\boldsymbol{M}_{qry})_i = 1 \mid \boldsymbol{M}_{ctx}, \boldsymbol{M}) \cdot P(\boldsymbol{M} \mid \boldsymbol{M}_{ctx}) \quad (8)$$

This decomposition reveals that the query probability depends on two factors: (1) the conditional query sampling strategy given both context and observation masks $P((\boldsymbol{M}_{qry})_i = 1 \mid \boldsymbol{M}_{ctx}, \boldsymbol{M})$, and (2) the posterior distribution of observation masks given the context $P(\boldsymbol{M} \mid \boldsymbol{M}_{ctx})$. Since $\boldsymbol{M}_{ctx} \subseteq \boldsymbol{M}$ by construction, we can always find observation masks $\boldsymbol{M}$ such that $P(\boldsymbol{M} \mid \boldsymbol{M}_{ctx}) >$

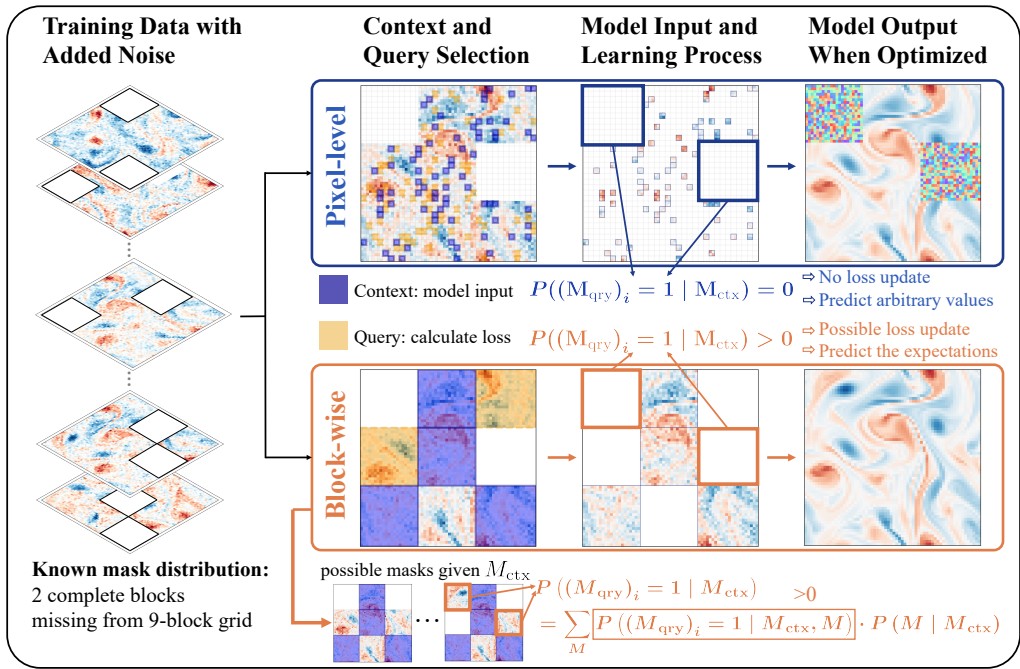

Figure 1: Impact of context-query partitioning strategies on learning effectiveness. Blue regions indicate context (model input), orange regions indicate query (loss calculation). **Top:** Problematic uniform sampling creates zero-query regions. **Bottom:** Effective block-structured sampling ensures balanced learning across all dimensions. See Fig. 10 for the resulting imputation failures.

0. However, $P((M_{\text{qry}})_i = 1 \mid M_{\text{ctx}}, M)$ is not guaranteed to be positive for all dimensions $i$. Fortunately, we can strategically design the sampling mechanism for $M_{\text{ctx}}$ to ensure that there indeed exist observation masks $M$ where both terms are simultaneously positive, thereby guaranteeing $P((M_{\text{qry}})_i = 1 \mid M_{\text{ctx}}) > 0$ for all observed dimensions. To illustrate how different context mask selection strategies affect the query probability $P((M_{\text{qry}})_i = 1 \mid M_{\text{ctx}})$, we first present a concrete example that demonstrates the critical impact of this design choice.

**Illustrate example.** Fig. 1 demonstrates how different partitioning strategies affect learning effectiveness using block-structured observation patterns. Consider a scenario where observation masks $M$ randomly mask 2 out of 9 spatial blocks:

- Problematic strategy (top): For each partially observed sample $x_{\text{obs}}$, context points are selected by uniform sampling across all observable dimensions, yielding $M_{\text{ctx}}^{\text{uni}}$. This strategy is problematic because for such given $M_{\text{ctx}}^{\text{uni}}$, there exists only one observation mask $M$ that contains it (i.e., the specific $M$ from which $x_{\text{obs}}$ was derived). Consequently, for masked dimensions $i$ where $M_i = 0$, we have $P((M_{\text{qry}})_i = 1 \mid M_{\text{ctx}}^{\text{uni}}, M) = 0$ because these dimensions are never available for query selection. The regions marked with rectangles represent such zero-probability areas, leading to incomplete learning.

- Effective strategy (bottom): Context masks are sampled by selecting entire blocks according to the same block-structured pattern as the observation masks, yielding $M_{\text{ctx}}^{\text{block}}$ that typically contains 4 complete blocks. Crucially, for a given $M_{\text{ctx}}^{\text{block}}$, there exist multiple possible observation masks $M$ that contain it (see two visualized possible masks in the figure). This multiplicity ensures that for any masked dimensions $i$, there always exists at least one possible observation mask $M$ such that $P((M_{\text{qry}})_i = 1 \mid M_{\text{ctx}}^{\text{block}}, M) > 0$, thereby guaranteeing positive query probabilities across all observable dimensions.

**Implementation strategy.** Generally speaking, we typically have knowledge about how the data becomes masked (e.g., sensor placement patterns, measurement protocols), which provides us with either explicit estimates or reasonable prior knowledge about $p_{\text{mask}}(M)$. In practice, to satisfy Prin-

ciple 1, a viable strategy is to design the context mask sampling mechanism $M_{\text{ctx}}$ based on the observation mask distribution $p_{\text{mask}}(M)$. Specifically, we sample $M_{\text{ctx}}$ from $M$ following the same structural pattern as $p_{\text{mask}}(M)$: for i.i.d. pixel-level observations, we independently sample each observed pixel; for block-structured observations, we sample complete blocks from available blocks in $M$. This distribution-preserving strategy ensures that every observed dimension can potentially be excluded from context (and included in query), guaranteeing $P\left((M_{\text{qry}})_i = 1 \mid M_{\text{ctx}}\right) > 0$ for all $i$. Under this design paradigm, the model learns during training to recover $x_0$ from context observations while leveraging the structural knowledge embedded in $p_{\text{mask}}(M)$. At inference time, the model can then utilize the complete observation $x_{\text{obs}}$ along with the same distributional knowledge $p_{\text{mask}}(M)$ to reconstruct the full data $x_0$.

**Training algorithm and practical considerations.** Building on our theoretical foundation and design principles, we present our training algorithm in Alg. 1. While ensuring $P((M_{\text{qry}})_i = 1 \mid M_{\text{ctx}}) > 0$ is necessary, the choice of context-query ratio involves two critical trade-offs:

- Information gap trade-off (Theorem 2): When $M_{\text{ctx}}$ contains few observed points relative to $M$, the large information gap increases approximation variance and slows down convergence.

- Parameter update frequency trade-off (equation 5): When $M_{\text{ctx}}$ contains too many observed points, query probabilities $p_i$ become small, leading to infrequent parameter updates for reconstructing missing information.

---

**Algorithm 1** Diffusion-based training for missing data imputation

---

**Require:** dataset $\mathcal{D} = \{(x_{\text{obs}}^{(i)}, M^{(i)})\}_{i=1}^N$
**Ensure:** trained neural network parameters $\theta$
1: **while** not converged **do**
2:    **for** each batch $(x_{\text{obs}}, M) \in \mathcal{D}$ **do**
3:       sample $t \sim \text{Uni}.(0, 1)$, $\epsilon \sim \mathcal{N}(\mathbf{0}, \mathbf{I})$
4:       $x_{\text{obs},t} \leftarrow M \odot (\alpha_t x_{\text{obs}} + \sigma_t \epsilon)$
5:       sample $M_{\text{ctx}}, M_{\text{qry}} \subseteq M$ (Princ. 1)
6:       $\hat{x} \leftarrow x_\theta(t, M_{\text{ctx}} \odot x_{\text{obs},t}, M_{\text{ctx}})$
7:       $\mathcal{L} \leftarrow \|M_{\text{qry}} \odot (\hat{x} - x_{\text{obs}})\|^2$
8:       update $\theta$ using gradient descent on $\mathcal{L}$
9:    **end for**
10: **end while**

---

These theoretical considerations suggest that moderate context ratios should achieve optimal performance by balancing both trade-offs. Our experimental results confirm this theoretical prediction, with detailed analysis provided in Appendix H.4 and Tab. 7.

### 3.3 ENSEMBLE SAMPLING FOR COMPLETE DATA RECONSTRUCTION

Our trained model approximates the conditional expectation $\mathbb{E}[x_0 \mid M_{\text{ctx}} \odot x_{\text{obs},t}, M_{\text{ctx}}]$ given a randomly sampled context mask, rather than the desired full conditional expectation $\mathbb{E}[x_0 \mid x_{\text{obs},t}, M]$ that conditions on the complete observation. To bridge this gap and enable complete data reconstruction, we leverage ensemble averaging across multiple context masks. This section presents our sampling procedures and their theoretical guarantees.

**Single-step sampling.** In many scientific applications, the observed data are sufficiently informative to constrain the solutions to a relatively concentrated region in the solution space (Alberti & Santacesaria, 2021). When the posterior distribution $p(x_0 \mid x_{\text{obs}}, M)$ is highly concentrated with solutions clustered closely together, it can be well-approximated by a narrow distribution centered at $x^*$, where $x^*$ represents the mean of the tightly clustered solutions consistent with the observations. In such cases, $\mathbb{E}[x_0 \mid x_{\text{obs}}, M] \approx x^*$ provides a good representative solution, reducing the need for extensive iterative denoising steps.

To leverage this property, we implement a single-step sampling procedure. We apply minimal noise at timestep $t = \delta$ where $0 < \delta \ll 1$:

$$x_\delta = \alpha_\delta x_{\text{obs}} + \sigma_\delta \epsilon, \quad \epsilon \sim \mathcal{N}(\mathbf{0}, \mathbf{I}) \tag{9}$$

The small noise level ensures $M \odot x_\delta \approx x_{\text{obs}}$. We then approximate the desired conditional expectation using ensemble averaging over $K$ randomly sampled context masks:

$$x^* = \mathbb{E}[x_0 \mid x_{\text{obs}}, M] \approx \frac{1}{K} \sum_{k=1}^K x_\theta\left(\delta, M_{\text{ctx}}^{(k)} \odot x_{\text{obs},\delta}, M_{\text{ctx}}^{(k)}\right), \tag{10}$$

where $M_{\text{ctx}}^{(1)}, \ldots, M_{\text{ctx}}^{(K)} \subseteq M$ are conditionally independent given $M$. This approach enables direct reconstruction in a single denoising step, making it particularly suitable for well-posed inverse problems where the observations strongly constrain the solution space.

We then establish the theoretical foundation that justifies our ensemble averaging approach. Let $\text{obs} = [M \odot \boldsymbol{x}_{\text{obs},t}, M]$ and $\text{ctx} = [M_{\text{ctx}} \odot \boldsymbol{x}_{\text{obs},t}, M_{\text{ctx}}]$ denote the full observation and context observation respectively.

**Theorem 2** (Ensemble approximation convergence). *Let* $\mathbb{E}[\boldsymbol{x}_0 \mid \text{obs}] := \mathbb{E}[\boldsymbol{x}_0 \mid \boldsymbol{x}_{\text{obs},t}, M]$ *be the ground truth conditional expectation and* $\mathbb{E}[\boldsymbol{x}_0 \mid \text{ctx}] = \mathbb{E}[\boldsymbol{x}_0 \mid M_{\text{ctx}} \odot \boldsymbol{x}_{obs,t}, M_{ctx}]$ *be the context-conditioned expectation. The expected squared error between these quantities is:*

$$\mathbb{E}\left[\|\mathbb{E}[\boldsymbol{x}_0 \mid \text{ctx}] - \mathbb{E}[\boldsymbol{x}_0 \mid \text{obs}]\|^2\right] = \mathbb{E}[\text{Var}[\boldsymbol{x}_0 \mid \text{ctx}]] - \mathbb{E}[\text{Var}[\boldsymbol{x}_0 \mid \text{obs}]] \tag{11}$$

*Consider a practical model with output* $\boldsymbol{x}_{\boldsymbol{\theta}}(t, \text{ctx}) = \mathbb{E}[\boldsymbol{x}_0 \mid \text{ctx}] + \boldsymbol{b}(\text{ctx}) + \boldsymbol{\epsilon}_{bias}(\text{ctx})$, *where* $\boldsymbol{b}(\text{ctx})$ *represents context-dependent deterministic bias and* $\boldsymbol{\epsilon}_{bias}(\text{ctx})$ *is random error with* $\mathbb{E}[\boldsymbol{\epsilon}_{bias}] = \boldsymbol{0}$. *Given the ensemble prediction in equation 10 as:*

$$\hat{\boldsymbol{\mu}}_K = \frac{1}{K} \sum_{k=1}^{K} \boldsymbol{x}_{\boldsymbol{\theta}}(t, \text{ctx}^{(k)}), \tag{12}$$

*the expected squared error between the ensemble prediction and ground truth is:*

$$\mathbb{E}\left[\|\hat{\boldsymbol{\mu}}_K - \mathbb{E}[\boldsymbol{x}_0 \mid \text{obs}]\|^2\right] = \|\underbrace{\mathbb{E}[\mathbb{E}[\boldsymbol{x}_0 \mid \text{ctx}]] - \mathbb{E}[\boldsymbol{x}_0 \mid \text{obs}]}_{\text{information gap}} + \underbrace{\mathbb{E}[\boldsymbol{b}(\text{ctx})]}_{\text{model bias}}\|^2$$
$$+ \frac{1}{K}\left(\underbrace{\text{Var}[\mathbb{E}[\boldsymbol{x}_0 \mid \text{ctx}]]}_{\text{data variance}} + \underbrace{\text{Var}[\boldsymbol{b}(\text{ctx})] + \text{Var}[\boldsymbol{\epsilon}_{bias}]}_{\text{model variance}}\right) \tag{13}$$

*As* $K \to \infty$, *the ensemble converges to:*

$$\lim_{K \to \infty} \mathbb{E}\left[\|\hat{\boldsymbol{\mu}}_K - \mathbb{E}[\boldsymbol{x}_0 \mid \text{obs}]\|^2\right] = \|\mathbb{E}[\mathbb{E}[\boldsymbol{x}_0 \mid \text{ctx}]] - \mathbb{E}[\boldsymbol{x}_0 \mid \text{obs}] + \mathbb{E}[\boldsymbol{b}(\text{ctx})]\|^2 \tag{14}$$

This theorem demonstrates that ensemble averaging eliminates the variance terms, with the remaining error determined by the information gap between context and full observations, plus any systematic model bias. The proof is provided in Appendix F.2.

**Multi-step sampling.** While single-step sampling suffices when observations nearly determine a unique solution, generating diverse imputed samples or handling cases with significant uncertainty requires a multi-step sampling procedure. This approach follows the standard diffusion sampling process but replaces each denoising step with ensemble averaging over multiple context masks. Note that multi-step sampling involves repeated application of the model to generated content, which can lead to slight accumulation of errors compared to the single-step approach (Xu et al., 2023). The detailed multi-step sampling algorithm is provided in Appendix E.

## 4 EXPERIMENTS

### 4.1 BASELINES

We compare against established baselines, including traditional imputation methods (Temporal Consistency (Huang et al., 2016), Fast Marching (Telea, 2004), Navier-Stokes inpainting (Bertalmio et al., 2001)) and recent diffusion-based approaches, MissDiff (Ouyang et al., 2023), AmbientDiff (Daras et al., 2023). To ensure a fair comparison, we modified MissDiff to use data matching instead of its original noise matching approach, which improves its performance (see Appendix H.3 for more detailed discussion). As a result, all three diffusion-based approaches, MissDiff, AmbientDiff, and our method, now employ the data matching paradigm. For all baseline methods, we experimented with both single-step sampling and multi-step sampling strategies, and report the best performance results between these two approaches to ensure optimal baseline comparisons. We also exclude methods by Chen et al. (2024b); Givens et al. (2025); Zhang et al. (2025a) due to their computational limitations: these approaches are designed for low-dimensional data and incur prohibitively high training costs when scaled to high dimensions. See Appendix G for details.

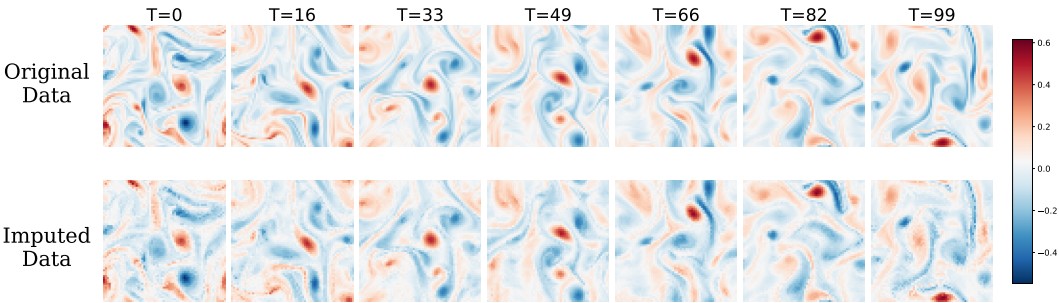

Figure 3: Comparison of original and imputed data from the Navier-Stokes dataset (60% observed points). The upper row shows the original data, while the bottom row shows the results after data imputation. Each sample consists of 100 frames at $64 \times 64$ resolution.

## 4.2 DATASETS

We conduct comprehensive experiments to validate our theoretical framework and demonstrate the effectiveness of our approach across diverse scientific domains. Our evaluation encompasses both synthetic PDE datasets: Shallow Water (Klöwer et al., 2018), Advection (Klöwer et al., 2018), and Navier-Stokes (Cao, 2024), and real-world climate data (ERA5) (Hersbach et al., 2020), under varying levels of data sparsity, ranging from 80% to as low as 1% observed points. To simulate realistic scientific measurement scenarios, we construct datasets where each sample contains only a subset of spatial locations with known values, while the remaining locations are permanently unobserved. This reflects the fundamental challenge in scientific applications where complete ground truth data is never available during training, distinguishing our setting from conventional imputation tasks that artificially mask complete observations.

**Shallow Water and Advection Equations.** We consider two fundamental geophysical PDE systems: the shallow water equations governing fluid dynamics with rotation, and the linear advection equation describing scalar transport. Each dataset contains 5k training, 1k validation, and 1k test samples with $32 \times 32$ spatial resolution and 50 temporal frames, generated with randomized physical parameters and initial conditions.

**Navier-Stokes Equations.** We use incompressible Navier-Stokes simulations of isotropic turbulence, featuring the characteristic Kolmogorov energy cascade. The dataset comprises 1,152 samples at $64 \times 64$ resolution with 100-frame sequences, generated using spectral or finite volume methods.

**ERA5 Reanalysis.** For real-world evaluation, we utilize the ERA5 atmospheric reanalysis from ECMWF, incorporating nine essential meteorological variables. We process one year of hourly data at $103 \times 120$ spatial resolution, segmented into 3-hour windows with sparse observations.

We assess reconstruction quality using physically meaningful metrics: PDE residual errors for shallow water, forward propagation accuracy for advection, and direct MSE for Navier-Stokes and ERA5. Detailed dataset specifications and evaluation protocols are provided in Appendix H.1.



(a) Visualization of data imputation results on Shallow Water and Advection datasets. The training set contains 30% observed data points. Each example shows a complete sample reconstructed by the model based on partial observations. Observed entries are marked with rectangles, while missing entries are filled in by the model.

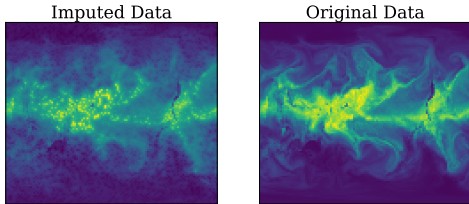

(b) Visualization of data imputation results for the "total column water vapor" variable in the ERA5 dataset (20% observed data points).

Figure 2: Data imputation visualization

Table 1: Performance comparison on physical dynamics imputation tasks, where masks are sampled pixel-wisely. Column headers indicate the percentage of spatial points observed in the dataset.

| Method | Navier-Stokes ($\times 10^{-3}$) | | | ERA5 ($\times 10^{-2}$) | | |
|---|---|---|---|---|---|---|
| | 80% | 60% | 20% | 20% | 10% | 1% |
| Temporal Consistency | 1.341 | 2.709 | 5.709 | 0.967 | 1.179 | 9.735 |
| Fast Marching | 0.486 | 1.220 | 3.737 | 0.710 | 0.978 | 3.053 |
| Navier-Stokes | 0.263 | 0.656 | 2.989 | 0.600 | 0.942 | 3.074 |
| MissDiff | $0.251 \pm 0.025$ | $0.611 \pm 0.077$ | $3.077 \pm 1.046$ | $0.416 \pm 0.004$ | $0.676 \pm 0.088$ | $1.653 \pm 0.296$ |
| AmbientDiff | $0.238 \pm 0.017$ | $0.538 \pm 0.024$ | $2.043 \pm 0.089$ | $0.256 \pm 0.002$ | $0.414 \pm 0.031$ | $1.234 \pm 0.437$ |
| **Ours** | $\mathbf{0.223 \pm 0.016}$ | $\mathbf{0.507 \pm 0.026}$ | $\mathbf{1.931 \pm 0.092}$ | $\mathbf{0.250 \pm 0.002}$ | $\mathbf{0.408 \pm 0.030}$ | $\mathbf{1.229 \pm 0.437}$ |

Table 2: Performance comparison on PDE imputation tasks, where masks are sampled block-wisely. Column headers indicate the fraction of observed blocks. The last two rows demonstrate our method using incorrect pixel-level versus correct block-wise context-query partitioning strategies.

| Method | | Shallow Water | | Advection | | Navier-Stokes |
|---|---|---|---|---|---|---|
| | | 8/9 | 5/9 | 8/9 | 5/9 | 8/9 |
| Temporal Consistency | | 0.6974 | 2.5486 | 0.4758 | 1.0940 | 1.4287 |
| Fast Marching | | 0.8454 | 3.3718 | 0.5042 | 1.4434 | 1.7391 |
| Navier-Stokes | | 0.4753 | 1.7565 | 0.4418 | 1.3594 | 1.7274 |
| MissDiff | | $0.0285 \pm 0.0024$ | $0.1166 \pm 0.0066$ | $0.1202 \pm 0.0047$ | $0.1979 \pm 0.0228$ | $1.4357 \pm 0.1132$ |
| AmbientDiff | | $0.0217 \pm 0.0063$ | $0.0925 \pm 0.0017$ | $0.1077 \pm 0.0009$ | $0.1524 \pm 0.0137$ | $1.4954 \pm 0.2609$ |
| Ours | pixel-level (incorrect) | $0.0215 \pm 0.0035$ | $0.0989 \pm 0.0007$ | $0.1171 \pm 0.0041$ | $0.1894 \pm 0.0179$ | $1.4925 \pm 0.2609$ |
| | block-wise (correct) | $\mathbf{0.0203 \pm 0.0059}$ | $\mathbf{0.0865 \pm 0.0014}$ | $\mathbf{0.1065 \pm 0.0009}$ | $\mathbf{0.1407 \pm 0.0116}$ | $\mathbf{0.7592 \pm 0.0386}$ |

## 4.3 EXPERIMENT RESULTS

**Pixel-level observation.** Individual spatial points are randomly masked throughout the domain, simulating sparse sensor networks or measurement failures. We test observation rates from 80% down to 1%, challenging the model to reconstruct scattered missing points using local spatial correlations. Tab. 1 and Tab. 10 demonstrate that our method achieves superior performance in the majority of evaluation scenarios.

**Block-wise observation.** This more challenging setting masks entire contiguous spatial regions, reflecting realistic constraints such as sensor placement limitations, regional measurement failures, or structured occlusions in observational systems. For instance, in the 5/9 block configuration, only 5 out of 9 spatial blocks contain observations, while 4 complete blocks remain entirely unobserved. This requires the model to reconstruct entire spatial regions without any local observations, relying solely on distant context and learned physical priors. Our results, shown in Tab. 2, demonstrate that the proposed strategic context-query partitioning, which adapts to the observation pattern during training, is essential. When the partitioning strategy matches the observation structure (block masks), our method effectively learns to reconstruct complete fields. Conversely, mismatched strategies lead to degraded performance, validating our theoretical analysis.

Table 3: Cross-distribution generalization on the Navier-Stokes dataset. Each column represents a model trained with a specific observation ratio, and each row represents the test observation ratio. Values indicate MSE between reconstructed and ground truth fields. The diagonal entries represent matched train-test distributions, while off-diagonal entries measure generalization under distribution shift. Models maintain reasonable performance when test-time observations are close to training conditions, but degrade gracefully when trained on fewer observation datasets.

| Test Set \ Training Set | 80% | 60% | 20% |
|---|---|---|---|
| 80% | $0.2229 \pm 0.0162$ | $0.2362 \pm 0.0121$ | $0.3363 \pm 0.0082$ |
| 60% | $0.4990 \pm 0.0260$ | $0.5071 \pm 0.0257$ | $0.6980 \pm 0.0188$ |
| 20% | - | - | $1.9315 \pm 0.0921$ |

**Cross-distribution generalization.** To evaluate the robustness of our method under distribution shift between training and testing, we conduct experiments where the observation ratio differs between training and inference. Specifically, we investigate whether a model trained on data with a certain observation density can generalize to test scenarios with different observation patterns. The key challenge lies in maintaining consistent model behavior when the available information at test time deviates from the training distribution. Our implementation addresses this through adaptive context mask sampling: during training with observation ratio $r_{\text{train}}$ (e.g., 80%), we sample context masks containing a fraction $\alpha$ of the observed points (e.g., 50%), resulting in the model receiving $r_{\text{train}} \times \alpha$ of the total pixels as input (e.g., 40%). At test time with a different observation ratio $r_{\text{test}}$ (e.g., 60%), we maintain the same effective input ratio by sampling $\frac{r_{\text{train}} \times \alpha}{r_{\text{test}}}$ of the available observations as context (e.g., $\frac{40\%}{60\%} = 66.7\%$). This strategy ensures the model operates within its learned input distribution while adapting to varying observation densities. Tab. 3 presents results across different train-test observation ratio combinations, demonstrating that our method maintains reasonable performance even under significant distribution shifts, though performance naturally degrades when the test-time observation ratio substantially deviates from the training distribution.

## 4.4 ABLATION STUDY

We conduct comprehensive ablation studies to analyze the contribution of key components in our framework: **(1)** test-time gap introduced by replacing $M_{\text{ctx}}$ with $M$, which evaluates the distribution mismatch between training and inference masks and its effect on reconstruction accuracy; **(2)** the choice of backbone architecture, where we investigate whether the proposed training paradigm generalizes across different neural architectures; **(3)** the context and query mask ratio selection guided by our theoretical analysis, examining the trade-off between information availability and parameter update frequency during training; and **(4)** the influence of the ensemble size $K$, which controls the variance reduction effect in our ensemble approximation during sampling. Detailed results and analysis are provided in Appendix H.4. Through systematic ablation studies, we validate our key theoretical insights. The results demonstrate that our method consistently outperforms existing approaches while providing theoretical guarantees for convergence to the desired conditional expectations.

## 5 CONCLUSION

We presented a principled framework for learning physical dynamics from incomplete observations using diffusion models trained directly on partial data through strategic context-query partitioning. Our approach enables diffusion models to learn consistent conditional expectations without requiring access to fully observed training samples, making it particularly suitable for realistic scientific datasets where missing data is unavoidable. Our theoretical analysis proves that training on incomplete data recovers the complete distribution with convergence guarantees, validated empirically with substantial improvements over baselines on synthetic PDEs and ERA5 climate data, especially in sparse regimes (1–20% coverage). The method's effectiveness across diverse physical systems demonstrates practical applicability for real-world scenarios where complete observations are inherently unavailable. More broadly, our results highlight the potential of diffusion-based generative models as a powerful tool for modeling partially observed physical systems and open promising directions for future research on integrating generative modeling with scientific data analysis.

### REPRODUCIBILITY STATEMENT

To ensure the reproducibility of our work, all experimental code and datasets are publicly available at `https://github.com/LOGO-CUHKSZ/incomplete-data-complete-dynamics`. The complete implementation, including model architectures, training procedures, and evaluation scripts, is provided in the repository with detailed documentation. All datasets used in our experiments are released, with the exception of the ERA5 dataset, which requires individual registration and download from the official ECMWF website due to licensing restrictions.

### ACKNOWLEDGMENTS

This work was supported by the National Science and Technology Major Project under Grant 2022ZD0116408.

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

## A  RELATED WORK

### A.1  IMPUTATION

Imputation, the task of filling missing or corrupted regions in data with plausible content, has been revolutionized by diffusion models across various modalities (Corneanu et al., 2024; Lemercier et al., 2025; Duan et al., 2024). Current imputation approaches follow three primary paradigms. Palette (Saharia et al., 2022) established conditioning on partially observed data by incorporating known regions at each denoising timestep. RePaint (Lugmayr et al., 2022) introduced a training-free method that leverages pretrained unconditional models by resampling known regions while generating content only for masked areas. Bridge-based methods (Liu et al., 2023; Yue et al., 2023; Albergo et al., 2023) design specialized diffusion processes between original and masked data distributions, requiring models trained to condition directly on masked inputs. DiffusionPDE (Huang et al., 2024) introduces a diffusion model to solve PDEs under partial observation by learning the joint distribution of coefficient and solution spaces. A critical limitation shared by all these approaches is their reliance on complete, unmasked data during training to learn the underlying data distribution before performing inference-time imputation on partially observed inputs. This assumption fundamentally conflicts with scientific applications where training data itself consists only of partial observations.

### A.2  GENERATIVE MODELING WITH MISSING DATA

Deep generative models tackle missing data through various approaches, including VAE-based methods (Ipsen et al., 2020; Ma et al., 2020) and GAN-based methods (Li et al., 2019; Li & Marlin, 2020). Some diffusion-based (Ouyang et al., 2023; Daras et al., 2023; Dai et al., 2024; Simkus & Gutmann, 2025) generative models generate clean samples from missing data, though they rely on heuristic intuition and lack rigorous convergence analysis. DiffLight (Chen et al., 2024a) leverages a partial rewards conditioned diffusion model to prevent missing rewards from interfering with the learning process. Zhang et al. (2025a) presents a theoretically sound framework combining diffusion models with EM algorithm for imputation, but its requirement of multiple complete model retraining cycles limits its scalability to complex and large datasets. More recently, Givens et al. (2025) proposed score matching with missing data, providing theoretical guarantees but facing computational scalability challenges due to high complexity in their importance weighting method and requiring auxiliary network training for their variational approach. Their experimental validation is limited to low-dimensional synthetic data and simple graphical models, raising questions about scalability to high-dimensional real-world scenarios.

## B  DATA MATCHING DIFFUSION MODELS

Diffusion models (Song & Ermon, 2019; Song et al., 2020; Ho et al., 2020) generate samples from a target data distribution by defining a forward process that gradually adds noise to data $\boldsymbol{x}_0 \sim p_0$ according to the stochastic differential equation:

$$\mathrm{d}\boldsymbol{x}_t = f(t)\boldsymbol{x}_t\mathrm{d}t + g(t)\mathrm{d}\boldsymbol{w}_t, \quad \boldsymbol{x}_0 \sim p_0, \tag{15}$$

where $\boldsymbol{w}_t \in \mathbb{R}^d$ represents standard Brownian motion, $f(t) = \frac{\mathrm{d}\log\alpha_t}{\mathrm{d}t}$, $g^2(t) = \frac{\mathrm{d}\sigma_t^2}{\mathrm{d}t} - 2\frac{\mathrm{d}\log\alpha_t}{\mathrm{d}t}\sigma_t^2$, and $\alpha_t$, $\sigma_t$ are predefined time-dependent functions. This process has the analytical solution $p_t(\boldsymbol{x}_t \mid \boldsymbol{x}_0) = \mathcal{N}\left(\boldsymbol{x}_t \mid \alpha_t\boldsymbol{x}_0, \sigma_t^2\boldsymbol{I}\right)$.

The reverse process generates samples by integrating backward from noise to data using:

$$\frac{\mathrm{d}\boldsymbol{x}_t}{\mathrm{d}t} = f(t)\boldsymbol{x}_t - \frac{1}{2}g^2(t)\nabla_{\boldsymbol{x}}\log p_t(\boldsymbol{x}_t), \quad \boldsymbol{x}_T \sim p_T(\boldsymbol{x}_T). \tag{16}$$

Since the terminal distribution $p_T(\boldsymbol{x}_T)$ becomes approximately Gaussian through appropriate parameter choices, sampling from it and reversing the process yields samples from $p_0$. The score function $\nabla_{\boldsymbol{x}} \log p_t(\boldsymbol{x}_t)$ is computationally intractable but can be estimated using neural networks via noise matching and data matching approaches (Zheng et al., 2023):

$$\mathcal{J}_{\text{noise}}(\boldsymbol{\theta}) = \mathbb{E}_{t,\boldsymbol{x}_0,\boldsymbol{\epsilon}} \left[ w(t) \left\| \boldsymbol{\epsilon}_{\boldsymbol{\theta}}(\boldsymbol{x}_t, t) - \boldsymbol{\epsilon} \right\|^2 \right], \qquad \boldsymbol{\epsilon}_{\boldsymbol{\theta}}^*(\boldsymbol{x}_t, t) = -\sigma_t \nabla_{\boldsymbol{x}} \log p_t(\boldsymbol{x}_t); \tag{17a}$$

$$\mathcal{J}_{\text{data}}(\boldsymbol{\theta}) = \mathbb{E}_{t,\boldsymbol{x}_0,\boldsymbol{\epsilon}} \left[ w(t) \left\| \boldsymbol{x}_{\boldsymbol{\theta}}(\boldsymbol{x}_t, t) - \boldsymbol{x}_0 \right\|^2 \right], \quad \boldsymbol{x}_{\boldsymbol{\theta}}^*(\boldsymbol{x}_t, t) = \frac{1}{\alpha_t} \boldsymbol{x}_t + \frac{\sigma_t^2}{\alpha_t} \nabla_{\boldsymbol{x}} \log p_t(\boldsymbol{x}_t), \tag{17b}$$

where $\boldsymbol{x}_t = \alpha_t \boldsymbol{x}_0 + \sigma_t \boldsymbol{\epsilon}$ with $\boldsymbol{\epsilon} \sim \mathcal{N}(\boldsymbol{0}, \boldsymbol{I})$, and $w(t)$ is a loss weight function. By Tweedie's formula (Efron, 2011; Kim & Ye, 2021; Chung et al., 2022), we also have $\boldsymbol{x}_{\boldsymbol{\theta}}^*(\boldsymbol{x}_t, t) = \mathbb{E}[\boldsymbol{x}_0 \mid \boldsymbol{x}_t]$.

## C  METHOD COMPARISON AND THEORETICAL ANALYSIS

### C.1  DIFFUSION PROBABILISTIC FIELDS

Several prior studies have proposed training generative models using field representations (Du et al., 2021; Dupont et al., 2022; Zhuang et al., 2023). Similarly, our approach trains a model to predict $\mathbb{E}[\boldsymbol{x}_0 \mid \boldsymbol{x}_{\text{obs},t}, \boldsymbol{M}]$ and applies the loss function to several randomly selected points across the entire set of coordinates. However, three main differences distinguish our approach from DPF: (1) Loss objective: Our method uses data matching while DPF uses noise matching, providing more flexible input requirements; (2) Model architecture: DPF requires both context and query inputs, whereas our method trains using only context, leading to different optimal solutions; (3) Theoretical foundation: DPF relies on heuristic designs without convergence guarantees, while our approach provides rigorous theoretical analysis. As shown in Appendix D, DPF's optimal solution depends on specific context-query mask combinations and may predict values differing from target predictions, whereas our method guarantees convergence to desired objectives through sufficient context mask sampling.

### C.2  AMBIENT DIFFUSION

While ambient diffusion (Daras et al., 2023) shares the fundamental principle of our approach—incorporating masks during training to predict clean data—several critical distinctions emerge upon closer examination. First, ambient diffusion lacks a theoretical analysis of how different mask distributions affect the learning dynamics, whereas our work provides a rigorous characterization through Theorem 1 parts (ii) and (iii). The most significant difference lies in the sampling methodology. Ambient diffusion employs a fixed mask sampling strategy and directly approximates $\mathbb{E}[\boldsymbol{x}_0 \mid \boldsymbol{M}_{\text{ctx}} \odot \boldsymbol{x}_{\text{obs},t}, \boldsymbol{M}_{\text{ctx}}]$ to approximate $\mathbb{E}[\boldsymbol{x}_0 \mid \boldsymbol{x}_{\text{obs},t}, \boldsymbol{M}]$. As we demonstrate in Theorem 2, this approximation introduces a distribution gap proportional to the variance of information provided by the conditioning terms, leading to suboptimal sample quality. In contrast, our method leverages the Martingale convergence theorem to approximate the true conditional expectation, providing theoretical convergence guarantees and eliminating the distribution gap inherent in the ambient diffusion approach.

### C.3  SCORE MATCHING WITH MISSING DATA

Our approach differs from this prior work (Givens et al., 2025) in a key way: we employ data matching rather than score matching due to its superior input flexibility and computational efficiency. In data matching, the optimal solution is $\mathbb{E}[\boldsymbol{x}_0 \mid \boldsymbol{x}_{\text{obs},t}, \boldsymbol{M}]$, which requires only the observed data $\boldsymbol{x}_{\text{obs},t}$ and mask $\boldsymbol{M}$ as inputs. By contrast, noise matching and velocity matching require computing the score term $\nabla_{\boldsymbol{x}_{\text{unobs},t}} \log p_t(\boldsymbol{x}_{\text{unobs},t} \mid \boldsymbol{x}_{\text{obs}}, \boldsymbol{M})$, which depends on the unobserved data $\boldsymbol{x}_{\text{unobs},t}$ that is unavailable during training. To address this limitation, the score matching approach relies on a Monte Carlo approximation, significantly increasing computational cost. Additionally, prior work (Zhou et al., 2024) has shown that data matching achieves superior performance compared to noise matching for PDE solution generation. Therefore, we focus exclusively on the data matching framework.

## C.4 Extension to other score-based models

Our approach can be extended to other score-based generative models through established theoretical connections. The equivalence between diffusion models and flow matching (Albergo et al., 2023), as well as between diffusion models and Bayesian flow networks (Xue et al., 2024), provides a foundation for this extension. Through Tweedie's formula (Efron, 2011), we can establish the connection between the score function $\nabla_{\boldsymbol{x}} \log p_t(\boldsymbol{x}_t)$ and the conditional expectation $\mathbb{E}[\boldsymbol{x}_0 \mid \boldsymbol{x}_t]$. Leveraging this connection via the parameterization trick, we can train a model to learn $\mathbb{E}[\boldsymbol{x}_0 \mid \boldsymbol{x}_{\text{obs},t}, \boldsymbol{M}]$ using our proposed method, which can then be converted to output the score function for the denoising process (Zhou et al., 2024).

## C.5 Comparison to masked self-supervised learning

Our context-query partitioning strategy is conceptually related to masked signal modeling, such as Masked Autoencoders (MAEs) (He et al., 2022). However, our approach differs fundamentally in its problem setting, data assumptions, and theoretical goals.

- **Data Completeness Assumption (Most Critical):** MAEs and related self-supervised methods train on *complete data* that is *artificially* masked. The full ground truth is always available during training. Our framework is designed for a different, and common, scientific scenario: the training data is **inherently incomplete** (e.g., from sparse sensors or cloud occlusion). No complete ground truth samples exist in our training dataset.

- **Primary Objective:** MAE uses masking as a pretext task to learn *robust representations* for downstream applications (Feichtenhofer et al., 2022). Our goal is to learn the *complete generative distribution* $p_{\text{data}}(\boldsymbol{x}_0)$ from these partial observations to perform accurate imputation of the true physical fields.

- **Masking Strategy and Theory:** MAE's random masking is an empirical choice for representation learning. Our *strategic* context-query partitioning is a direct consequence of our theoretical analysis in Theorem. 1. It is specifically designed to solve the core challenge of our setting: how to ensure that dimensions *permanently missing* in the training set still receive a positive query probability $(P((\boldsymbol{M}_{\text{qry}})_i = 1 \mid \boldsymbol{M}_{\text{ctx}}) > 0)$ and meaningful gradient updates. As shown in Theorem. 1, without this, the model learns arbitrary values for these unobserved regions. This is a problem MAE does not encounter, as it always has access to the complete ground truth for its loss calculation.

## C.6 Method comparison and theoretical analysis

We provide additional method comparisons in Tab. 4 and summarize the key feature comparisons with the most related methods, contrasting our approach with three existing methods: (1) AmbientDiff (Daras et al., 2023), (2) Diffusion Probabilistic Fields (DPF) (Zhuang et al., 2023), and (3) Score Matching with Missing Data (Givens et al., 2025).

Table 4: Comparison of the most related methods.

| Aspect | Ours | Ambient Diffusion | DPF | MissDiff |
|---|---|---|---|---|
| **Training Objective** | Data matching | Data matching | Noise matching | Noise matching |
| **Model Input** | Context only | Fixed mask sampling | Both context and query | Masked tabular data |
| **Query Mask Usage** | Hidden during training | Hidden during training | Provided to model | Provided to model |
| **Mask Sampling** | Random subsets: $\boldsymbol{M}_{\text{ctx}} \subseteq \boldsymbol{M}$ | Fixed distribution | Both $\boldsymbol{M}_{\text{ctx}}$ and $\boldsymbol{M}_{\text{qry}}$ | $\boldsymbol{M}_{\text{ctx}} = \boldsymbol{M}_{\text{qry}} = \boldsymbol{M}$ |
| **Expectation Approx.** | Ensemble: $\frac{1}{n}\sum_i \boldsymbol{x}_\theta(t, \text{ctx})$ | Direct: $\mathbb{E}[\boldsymbol{x}_0 \mid \text{ctx}]$ | Incorrect | Direct: $\mathbb{E}[\boldsymbol{x}_0 \mid \text{obs}]$ |
| **Theoretical Guarantees** | ✓ Convergence proofs (Thm. 1& 2) | ✗ Lacks rigorous analysis | ✗ Heuristic design | ✗ Lacks rigorous analysis |
| **Distribution Gap** | ✓ Minimized via ensemble | ✗ Gap $\propto$ Var[conditioning] | ✗ Not addressed | ✗ Not addressed |
| **Learning Dynamics** | ✓ Gradient scaling analysis | ✗ No analysis | ✗ No analysis | ✗ No analysis |

## D Method comparison: diffusion probabilistic fields

We summarize the training and sampling algorithms for diffusion probabilistic fields (DPF) (Zhuang et al., 2023) in Alg. 2, 3. We cannot directly adopt DPF for our dynamic completion tasks because

we lack access to the ground truth value of qry during training. However, we can still compare several high-level ideas with those in DPF.

---

**Algorithm 2** DPF training process (Zhuang et al., 2023)

---
1: **repeat**
2:    $\boldsymbol{x}_0 \sim p_{\text{data}}, t \sim \text{Uniform}(0, 1)$
3:    $\boldsymbol{\epsilon}_{\text{ctx}} \sim \mathcal{N}(\mathbf{0}, \mathbf{I}), \boldsymbol{\epsilon}_{\text{qry}} \sim \mathcal{N}(\mathbf{0}, \boldsymbol{I})$
4:    Sample $\boldsymbol{M}_{\text{ctx}}, \boldsymbol{M}_{\text{qry}}$
5:    $\text{ctx} = [\boldsymbol{M}_{\text{ctx}}, \boldsymbol{M}_{\text{ctx}} \odot (\alpha_t \boldsymbol{x}_0 + \sigma_t \boldsymbol{\epsilon}_{\text{ctx}})]$
6:    $\text{qry} = [\boldsymbol{M}_{\text{qry}}, \boldsymbol{M}_{\text{qry}} \odot (\alpha_t \boldsymbol{x}_0 + \sigma_t \boldsymbol{\epsilon}_{\text{qry}})]$
7:    Optimize the loss function
     $\mathcal{L} = \|\boldsymbol{M}_{\text{qry}} \odot (\boldsymbol{\epsilon}_{\boldsymbol{\theta}}(t, \text{ctx}, \text{qry}) - \boldsymbol{\epsilon}_{\text{qry}})\|^2$
8: **until** converged

---

**Algorithm 3** DPF sampling process (Zhuang et al., 2023)

---
1: Sample $\boldsymbol{M}_{\text{ctx}} \subseteq \boldsymbol{M}_{\text{qry}}$
2: $\boldsymbol{x}_T \sim \mathcal{N}(\mathbf{0}, \boldsymbol{I})$
3: $\text{ctx} = [\boldsymbol{M}_{\text{ctx}}, \boldsymbol{M}_{\text{ctx}} \odot \boldsymbol{x}_T]$
4: $\text{qry} = [\boldsymbol{M}_{\text{qry}}, \boldsymbol{M}_{\text{qry}} \odot \boldsymbol{x}_T]$
5: **for** $t = T, \ldots, 1$ **do**
6:    $\boldsymbol{x}_{t-1} = \text{ProbabilityFlowODE}(t, \boldsymbol{x}_t, \boldsymbol{\epsilon}_{\boldsymbol{\theta}}(t, \text{ctx}, \text{qry}))$
7:    $\text{ctx} = [\boldsymbol{M}_{\text{ctx}}, \boldsymbol{M}_{\text{ctx}} \odot \boldsymbol{x}_{t-1}]$
8:    $\text{qry} = [\boldsymbol{M}_{\text{qry}}, \boldsymbol{M}_{\text{qry}} \odot \boldsymbol{x}_{t-1}]$
9: **end for**
10: **return** sample values evaluated on $\boldsymbol{M}_{\text{qry}}$

---

There are three main difference between our proposed method and DPF:

1. Diffusion loss objective: our methods use data matching while DPF uses noise matching. Using data matching to predict conditional expectation has a more flexible requirement on model input (see Sec. C.6 for details).

2. The DPF method suggests we should take $\boldsymbol{M}_{\text{qry}}$ to be the full observed mask, which is impossible to implement in our task.

3. DPF trains a model that takes both ctx and qry as inputs. In contrast, our method trains a model using only ctx as input. This difference leads the model to converge to a different optimal solution.

In the following analysis, we will assume that we have the fully observed sample during the training session. We will analyze the output of the model optimized by Alg. 2 and demonstrate that it does not yield the desired solution for the denoising process.

We reparameterize the loss function as

$$\mathcal{L}(\boldsymbol{\theta}) = \|\boldsymbol{M}_{\text{qry}} \odot (\boldsymbol{x}_{\boldsymbol{\theta}}(t, \text{ctx}, \text{qry}) - \boldsymbol{x}_0)\|^2 \tag{18}$$

When optimized,

$$\boldsymbol{M}_{\text{qry}} \odot (\alpha_t \boldsymbol{x}_{\boldsymbol{\theta}}^*(t, \text{ctx}, \text{qry}) + \sigma_t \boldsymbol{\epsilon}_{\boldsymbol{\theta}}^*(t, \text{ctx}, \text{qry}) - \text{qry}[1]) = \mathbf{0} \tag{19}$$

In the following, we will analyze the optimal solution given by equation 18. The conditional expectation $\mathbb{E}[\boldsymbol{x}_0 \mid \text{ctx}, \text{qry}]$ minimizes the expected squared error (Bishop & Nasrabadi, 2006). Hence, we have

$$\boldsymbol{M}_{\text{qry}} \odot \boldsymbol{x}_{\boldsymbol{\theta}}^*(t, \text{ctx}, \text{qry}) = \boldsymbol{M}_{\text{qry}} \odot \mathbb{E}[\boldsymbol{x}_0 \mid \text{ctx}, \text{qry}] \tag{20}$$

For simplicity, we consider a simple distribution $\boldsymbol{x}_0 \sim p_{\text{data}} = \mathcal{N}(\boldsymbol{\mu}, \boldsymbol{\Sigma})$. The noisy observations are defined as: $\boldsymbol{z}_{\text{ctx}} = \alpha_t \boldsymbol{x}_0 + \sigma_t \boldsymbol{\epsilon}_{\text{ctx}}, \boldsymbol{z}_{\text{qry}} = \alpha_t \boldsymbol{x}_0 + \sigma_t \boldsymbol{\epsilon}_{\text{qry}}$. We have a joint Gaussian distribution:

$$\begin{bmatrix} \boldsymbol{x}_0 \\ \boldsymbol{z}_{\text{ctx}} \\ \boldsymbol{z}_{\text{qry}} \end{bmatrix} \sim \mathcal{N}\left(\begin{bmatrix} \boldsymbol{\mu} \\ \alpha_t \boldsymbol{\mu} \\ \alpha_t \boldsymbol{\mu} \end{bmatrix}, \begin{bmatrix} \boldsymbol{\Sigma} & \alpha_t \boldsymbol{\Sigma} & \alpha_t \boldsymbol{\Sigma} \\ \alpha_t \boldsymbol{\Sigma} & \alpha_t^2 \boldsymbol{\Sigma} + \sigma_t^2 \boldsymbol{I} & \alpha_t^2 \boldsymbol{\Sigma} \\ \alpha_t \boldsymbol{\Sigma} & \alpha_t^2 \boldsymbol{\Sigma} & \alpha_t^2 \boldsymbol{\Sigma} + \sigma_t^2 \boldsymbol{I} \end{bmatrix}\right) \tag{21}$$

Let $\boldsymbol{y}$ denote the observed entries selected by the masks: $\boldsymbol{y} = \boldsymbol{S} \begin{bmatrix} \boldsymbol{z}_{\text{ctx}} \\ \boldsymbol{z}_{\text{qry}} \end{bmatrix}$, where $\boldsymbol{S}$ is the selection matrix that extracts the masked entries from $\boldsymbol{z}_{\text{ctx}}$ and $\boldsymbol{z}_{\text{qry}}$. By the property of linear transformation of multivariate Gaussian, suppose we have a joint Gaussian distribution of the form:

$$\begin{bmatrix} \boldsymbol{x}_0 \\ \boldsymbol{z} \end{bmatrix} \sim \mathcal{N}\left(\begin{bmatrix} \boldsymbol{\mu}_x \\ \boldsymbol{\mu}_z \end{bmatrix}, \begin{bmatrix} \boldsymbol{\Sigma}_{xx} & \boldsymbol{\Sigma}_{xz} \\ \boldsymbol{\Sigma}_{zx} & \boldsymbol{\Sigma}_{zz} \end{bmatrix}\right) \tag{22}$$

then the joint distribution of $\boldsymbol{x}_0$ and $\boldsymbol{y}$ is given by

$$\begin{bmatrix} \boldsymbol{x}_0 \\ \boldsymbol{y} \end{bmatrix} = \begin{bmatrix} \boldsymbol{x}_0 \\ \boldsymbol{S}\boldsymbol{z} \end{bmatrix} \sim \mathcal{N}\left(\begin{bmatrix} \boldsymbol{\mu}_x \\ \boldsymbol{S}\boldsymbol{\mu}_z \end{bmatrix}, \begin{bmatrix} \boldsymbol{\Sigma}_{xx} & \boldsymbol{\Sigma}_{xz}\boldsymbol{S}^\top \\ \boldsymbol{S}\boldsymbol{\Sigma}_{zx} & \boldsymbol{S}\boldsymbol{\Sigma}_{zz}\boldsymbol{S}^\top \end{bmatrix}\right) \tag{23}$$

Using the above property, the joint distribution of $\boldsymbol{x}_0$ and $\boldsymbol{y}$ is then:

$$\begin{bmatrix} \boldsymbol{x}_0 \\ \boldsymbol{y} \end{bmatrix} = \begin{bmatrix} \boldsymbol{x}_0 \\ \boldsymbol{S} \begin{bmatrix} \boldsymbol{z}_{\text{ctx}} \\ \boldsymbol{z}_{\text{qry}} \end{bmatrix} \end{bmatrix} \tag{24a}$$

$$\sim \mathcal{N}\left( \begin{bmatrix} \boldsymbol{\mu} \\ \alpha_t \boldsymbol{S} \left( \mathbf{1}_2 \otimes \boldsymbol{I} \right) \boldsymbol{\mu} \end{bmatrix}, \begin{bmatrix} \boldsymbol{\Sigma}_{xx} & \alpha_t \left( \mathbf{1}_2^\top \otimes \boldsymbol{\Sigma} \right) \boldsymbol{S}^\top \\ \alpha_t \boldsymbol{S} \left( \mathbf{1}_2 \otimes \boldsymbol{\Sigma} \right) & \boldsymbol{S} \left( \alpha_t^2 \left( \mathbf{1}_2 \mathbf{1}_2^\top \otimes \boldsymbol{\Sigma} \right) + \sigma_t^2 \boldsymbol{I} \right) \boldsymbol{S}^\top \end{bmatrix} \right) \tag{24b}$$

Thus, the conditional expectation is:

$$\mathbb{E}\left[ \boldsymbol{x}_0 \mid \boldsymbol{y} \right] = \boldsymbol{\mu} + \alpha_t \left( \mathbf{1}_2^\top \otimes \boldsymbol{\Sigma} \right) \boldsymbol{S}^\top \left[ \boldsymbol{S} \left( \alpha_t^2 \left( \mathbf{1}_2 \mathbf{1}_2^\top \otimes \boldsymbol{\Sigma} \right) + \sigma_t^2 \boldsymbol{I} \right) \boldsymbol{S}^\top \right]^{-1} \left( \boldsymbol{y} - \alpha_t \boldsymbol{S} \left( \mathbf{1}_2 \otimes \boldsymbol{I} \right) \boldsymbol{\mu} \right) \tag{25}$$

Therefore, DPF's optimal solution depends on the specific context and query masks chosen, and may predict values that differ from target predictions. In contrast, our proposed method has a theoretical guarantee: with sufficient sampling of context masks, the model's output will converge to the desired objective (see Theorem 2).

## E  MULTI-STEP SAMPLING

In the following, we will consider two specific settings of the mask $\boldsymbol{M}$ and diffusion time $t$, and use these constructions to approximate $\mathbb{E}\left[ \boldsymbol{x}_0 \mid \boldsymbol{x}_t, \boldsymbol{x}_{\text{obs}}, \boldsymbol{M} \right]$.

**Diffusion expectation approximation.**  We randomly generate multiple masks $\{ \boldsymbol{M}_{\text{rnd}}^{(i)} \}_{i=1}^K$, where $\boldsymbol{M}_{\text{rnd}}$ follows the same marginal distribution as $\boldsymbol{M}_{\text{ctx}}$ in the training process, but not necessarily being a subset of $\boldsymbol{M}$ and take the average across all samples yields:

$$\mathbb{E}\left[ \boldsymbol{x}_0 \mid \boldsymbol{x}_t \right] \approx \frac{1}{K} \sum_{k=1}^K \boldsymbol{x}_{\boldsymbol{\theta}} \left( t, \boldsymbol{M}_{\text{rnd}}^{(k)} \odot \boldsymbol{x}_t, \boldsymbol{M}_{\text{rnd}}^{(k)} \right) \tag{26}$$

This Monte Carlo estimation demonstrates that our model, despite being trained exclusively on masked data, can recover the full data distribution. The averaging process allows us to obtain the same distributional modeling capability as standard diffusion models trained on complete datasets.

**Imputation expectation approximation.**  Given partially observed samples $\boldsymbol{x}_{\text{obs}}$, we apply the forward diffusion process to a small timestamp $t = \delta$ and generate random masks $\{ \boldsymbol{M}_{\text{ctx}}^{(k)} \}_{k=1}^K \subseteq \boldsymbol{M}$ to approximate:

$$\mathbb{E}\left[ \boldsymbol{x}_0 \mid \boldsymbol{x}_{\text{obs}}, \boldsymbol{M} \right] \approx \mathbb{E}\left[ \boldsymbol{x}_0 \mid \boldsymbol{x}_{\text{obs},\delta}, \boldsymbol{M} \right] \approx \frac{1}{K} \sum_{k=1}^K \boldsymbol{x}_{\boldsymbol{\theta}} \left( \delta, \boldsymbol{M}_{\text{ctx}}^{(k)} \odot \boldsymbol{x}_{\text{obs},\delta}, \boldsymbol{M}_{\text{ctx}}^{(k)} \right) \tag{27}$$

The optimal denoiser $\mathbb{E}\left[ \boldsymbol{x}_0 \mid \boldsymbol{x}_t, \boldsymbol{x}_{\text{obs}}, \boldsymbol{M} \right]$ requires the expectation of $\boldsymbol{x}_0$ conditional on all three information sources: the noisy state $\boldsymbol{x}_t$, the clean observations $\boldsymbol{x}_{\text{obs}}$, and the observation mask $\boldsymbol{M}$. Our approach decomposes this complex conditioning into two manageable components: the *diffusion expectation* $\mathbb{E}\left[ \boldsymbol{x}_0 \mid \boldsymbol{x}_t \right]$ captures the denoising information from the current noisy state, while the *imputation expectation* $\mathbb{E}\left[ \boldsymbol{x}_0 \mid \boldsymbol{x}_{\text{obs}}, \boldsymbol{M} \right]$ incorporates the structural information from the observed values and their locations. We then heuristically combine these two sources of information through a weighted average:

$$\hat{\boldsymbol{x}}_{\boldsymbol{\theta}} \left( t, \boldsymbol{x}_t, \boldsymbol{x}_{\text{obs}}, \boldsymbol{M} \right) := \mathbb{E}\left[ \boldsymbol{x}_0 \mid \boldsymbol{x}_t, \boldsymbol{x}_{\text{obs}}, \boldsymbol{M} \right] \approx \omega_t \mathbb{E}\left[ \boldsymbol{x}_0 \mid \boldsymbol{x}_t \right] + \left( 1 - \omega_t \right) \mathbb{E}\left[ \boldsymbol{x}_0 \mid \boldsymbol{x}_{\text{obs}}, \boldsymbol{M} \right]$$
$$\approx \omega_t \mathbb{E}_{\boldsymbol{M}_{\text{rnd}}} \left[ \boldsymbol{x}_{\boldsymbol{\theta}} \left( t, \boldsymbol{M}_{\text{rnd}} \odot \boldsymbol{x}_t, \boldsymbol{M}_{\text{rnd}} \right) \right] + \left( 1 - \omega_t \right) \mathbb{E}_{\boldsymbol{M}_{\text{ctx}} \subseteq \boldsymbol{M}} \left[ \boldsymbol{x}_{\boldsymbol{\theta}} \left( \delta, \boldsymbol{M}_{\text{ctx}} \odot \boldsymbol{x}_{\text{obs},\delta}, \boldsymbol{M}_{\text{ctx}} \right) \right] \tag{28}$$

where $\omega_t$ is a monotonically increasing weight function that transitions from 0 to 1 as the diffusion process progresses and $\delta$ is a sufficiently small positive number.

We present our proposed sampling algorithm in Alg. 4. At the implementation level, we precompute the *imputation expectation* once before the denoising process begins. During each denoising step, we approximate the *diffusion expectation* by sampling a single random mask $\boldsymbol{M}_{\text{rnd}}$ and making one model evaluation. This single-sample approximation is analogous to using a batch size of 1 in stochastic gradient descent, where we accept the variance from using only one sample in exchange for computational efficiency.

Following Lugmayr et al. (2022), our sampling procedure applies different denoising strategies to observed and unobserved regions. For unobserved elements, we estimate the noise using our trained model, while for observed elements, we directly compute the noise from the known clean observations. This approach ensures that the observed values remain consistent with their true underlying data throughout the denoising process. A more advanced setting proposed in Huang et al. (2024) uses guided diffusion sampling that starts from Gaussian noise and iteratively denoises it while being guided by two loss terms: an observation loss (matching sparse measurements) and a PDE loss (satisfying the governing equation), ultimately generating complete solutions that are consistent with both the partial observations and the underlying physics. How-

---

**Algorithm 4** Diffusion-based sampling for data imputation

---

**Require:** partially observed data $\boldsymbol{x}_{\text{obs}}$, mask $\boldsymbol{M}$, trained model $\boldsymbol{x}_{\boldsymbol{\theta}}$
**Ensure:** imputed complete data $\boldsymbol{x}_0$
1: initialize: $\boldsymbol{x}_T \sim \mathcal{N}(\boldsymbol{0}, \boldsymbol{I})$
2: **for** diffusion steps from $s$ to $t$ **do**
3: $\quad \boldsymbol{\epsilon}_{\text{unobs}} \leftarrow \frac{\boldsymbol{x}_s - \alpha_s \hat{\boldsymbol{x}}_{\boldsymbol{\theta}}(s, \boldsymbol{x}_s, \boldsymbol{x}_{\text{obs}}, \boldsymbol{M})}{\sigma_s}$ (Eq. 28)
4: $\quad \boldsymbol{\epsilon}_{\text{obs}} \leftarrow \frac{\boldsymbol{x}_s - \alpha_s \boldsymbol{x}_{\text{obs}}}{\sigma_s}$
5: $\quad \boldsymbol{\epsilon}_{\text{full}} \leftarrow \boldsymbol{M} \odot \boldsymbol{\epsilon}_{\text{obs}} + (1 - \boldsymbol{M}) \odot \boldsymbol{\epsilon}_{\text{unobs}}$
6: $\quad \boldsymbol{x}_t \leftarrow \text{DiffusionODE}(s, t, \boldsymbol{x}_s, \boldsymbol{\epsilon}_{\text{full}})$
7: **end for**
8: $\boldsymbol{x}_0 \leftarrow \boldsymbol{M} \odot \boldsymbol{x}_{\text{obs}} + (1 - \boldsymbol{M}) \odot \boldsymbol{x}_t$
9: **return** $\boldsymbol{x}_0$

---

ever, we did not implement this approach in our work, and combining our mask-based denoising strategy with physics-informed guided diffusion remains an interesting direction for future research.

# F PROOFS

This section provides detailed mathematical proofs for the main theoretical results presented in the paper. We begin by establishing key assumptions and notation that will be used throughout the proofs.

**Assumption 1** (Uncorrelated decomposition). *Decompose the model output as* $\boldsymbol{x}_{\boldsymbol{\theta}}(t, \text{ctx}) = \mathbb{E}[\boldsymbol{x}_0 \mid \text{ctx}] + \boldsymbol{b}(\text{ctx}) + \boldsymbol{\epsilon}_{\text{bias}}(\text{ctx})$. *Given a context* ctx, *the following three components are mutually uncorrelated:*

- *data component:* $\mathbb{E}[\boldsymbol{x}_0 \mid \text{ctx}] - \mathbb{E}[\mathbb{E}[\boldsymbol{x}_0 \mid \text{ctx}]]$,

- *bias component:* $\boldsymbol{b}(\text{ctx}) - \mathbb{E}[\boldsymbol{b}(\text{ctx})]$,

- *random error:* $\boldsymbol{\epsilon}(\text{ctx})$.

*We also assume that* $\boldsymbol{\epsilon}(\text{ctx}^{(i)})$ *and* $\boldsymbol{\epsilon}(\text{ctx}^{(j)})$ *are independent for* $i \neq j$.

For notation simplicity, we denote

$$\text{obs} = [\boldsymbol{M} \odot \boldsymbol{x}_{\text{obs},t}, \boldsymbol{M}] \tag{29a}$$

$$\text{ctx} = [\boldsymbol{M}_{\text{ctx}} \odot \boldsymbol{x}_{\text{obs},t}, \boldsymbol{M}_{\text{ctx}}] \tag{29b}$$

$$\text{qry} = [\boldsymbol{M}_{\text{qry}} \odot \boldsymbol{x}_{\text{obs},t}, \boldsymbol{M}_{\text{qry}}] \tag{29c}$$

when it is clear from the context.

## F.1 ANALYSIS OF MODEL OUTPUTS UNDER OPTIMAL LOSS CONDITIONS

We begin the proof with a foundational lemma that establishes the key relationship between the optimal model output and the conditional expectations.

**Lemma 1** (Optimal Function for Element-wise Weighted MSE). *Let* $\boldsymbol{X}$, $\boldsymbol{Y}$, *and* $\boldsymbol{Z}$ *be random vectors in* $\mathbb{R}^d$ *such that the relevant second moments are finite. Let* $\boldsymbol{g} : Space(\boldsymbol{Y}) \to \mathbb{R}^d$ *be a deterministic function and let the objective function* $L(\boldsymbol{g})$ *be defined as*

$$L(\boldsymbol{g}) = \mathbb{E}\left[\|\boldsymbol{Z} \odot \boldsymbol{g}(\boldsymbol{Y}) - \boldsymbol{Z} \odot \boldsymbol{X}\|^2\right], \tag{30}$$

*where* $\odot$ *denotes the Hadamard (element-wise) product. If each component of the vector* $\mathbb{E}[\boldsymbol{Z} \odot \boldsymbol{Z} \mid \boldsymbol{Y}]$ *is strictly positive almost surely, then the unique function* $\boldsymbol{g}^*$ *that minimizes* $L(\boldsymbol{g})$ *is given by*

$$\boldsymbol{g}^*(\boldsymbol{Y}) = \frac{\mathbb{E}[\boldsymbol{Z} \odot \boldsymbol{Z} \odot \boldsymbol{X} \mid \boldsymbol{Y}]}{\mathbb{E}[\boldsymbol{Z} \odot \boldsymbol{Z} \mid \boldsymbol{Y}]}, \tag{31}$$

*where the division is performed element-wise.*

*Proof.* The objective function $L(\boldsymbol{g})$ can be decomposed by writing the squared Euclidean norm as a sum over its components.

$$L(\boldsymbol{g}) = \mathbb{E}\left[\|\boldsymbol{Z} \odot (\boldsymbol{g}(\boldsymbol{Y}) - \boldsymbol{X})\|^2\right] \tag{32a}$$

$$= \mathbb{E}\left[\sum_{i=1}^{d} (\boldsymbol{Z}_i(\boldsymbol{g}_i(\boldsymbol{Y}) - \boldsymbol{X}_i))^2\right] \tag{32b}$$

$$= \sum_{i=1}^{d} \mathbb{E}\left[\boldsymbol{Z}_i^2(\boldsymbol{g}_i(\boldsymbol{Y}) - \boldsymbol{X}_i)^2\right] \tag{32c}$$

The final step follows from the linearity of expectation. The total loss is a sum of non-negative terms, so $L(\boldsymbol{g})$ is minimized if and only if each term in the summation is minimized independently. Let $L_i(\boldsymbol{g}_i) = \mathbb{E}[\boldsymbol{Z}_i^2(\boldsymbol{g}_i(\boldsymbol{Y}) - \boldsymbol{X}_i)^2]$ be the $i$-th term. The optimization problem is thus reduced to finding the function $\boldsymbol{g}_i$ that minimizes $L_i$ for each component $i \in \{1, \ldots, d\}$. By the law of total expectation, $L_i(\boldsymbol{g}_i)$ can be written as:

$$L_i(\boldsymbol{g}_i) = \mathbb{E}_{\boldsymbol{Y}}\left[\mathbb{E}\left[\boldsymbol{Z}_i^2(\boldsymbol{g}_i(\boldsymbol{Y}) - \boldsymbol{X}_i)^2 \mid \boldsymbol{Y}\right]\right] \tag{33}$$

The outer expectation is minimized by minimizing the inner conditional expectation for any given realization $\boldsymbol{y}$ from the space of $\boldsymbol{Y}$. For a fixed $\boldsymbol{y}$, let $\boldsymbol{v}_i = \boldsymbol{g}_i(\boldsymbol{y})$ be a deterministic scalar. The inner expectation becomes:

$$\mathbb{E}\left[\boldsymbol{Z}_i^2(\boldsymbol{v}_i - \boldsymbol{X}_i)^2 \mid \boldsymbol{Y} = \boldsymbol{y}\right] = \mathbb{E}\left[\boldsymbol{Z}_i^2(\boldsymbol{v}_i^2 - 2\boldsymbol{v}_i\boldsymbol{X}_i + \boldsymbol{X}_i^2) \mid \boldsymbol{Y} = \boldsymbol{y}\right] \tag{34}$$

Applying the linearity of conditional expectation yields a quadratic function of $\boldsymbol{v}_i$:

$$\boldsymbol{v}_i^2\mathbb{E}[\boldsymbol{Z}_i^2 \mid \boldsymbol{Y} = \boldsymbol{y}] - 2\boldsymbol{v}_i\mathbb{E}[\boldsymbol{Z}_i^2\boldsymbol{X}_i \mid \boldsymbol{Y} = \boldsymbol{y}] + \mathbb{E}[\boldsymbol{Z}_i^2\boldsymbol{X}_i^2 \mid \boldsymbol{Y} = \boldsymbol{y}] \tag{35}$$

This is a convex quadratic in $\boldsymbol{v}_i$, since its leading coefficient $\mathbb{E}[\boldsymbol{Z}_i^2 \mid \boldsymbol{Y} = \boldsymbol{y}]$ is strictly positive by hypothesis. The unique minimum is found by setting the derivative with respect to $\boldsymbol{v}_i$ to zero:

$$2\boldsymbol{v}_i\mathbb{E}[\boldsymbol{Z}_i^2 \mid \boldsymbol{Y} = \boldsymbol{y}] - 2\mathbb{E}[\boldsymbol{Z}_i^2\boldsymbol{X}_i \mid \boldsymbol{Y} = \boldsymbol{y}] = 0 \tag{36}$$

Solving for $\boldsymbol{v}_i$ gives the optimal value for the component function at $\boldsymbol{y}$:

$$\boldsymbol{v}_i = \frac{\mathbb{E}[\boldsymbol{Z}_i^2\boldsymbol{X}_i \mid \boldsymbol{Y} = \boldsymbol{y}]}{\mathbb{E}[\boldsymbol{Z}_i^2 \mid \boldsymbol{Y} = \boldsymbol{y}]} \tag{37}$$

This establishes the optimal form for each component $\boldsymbol{g}_i^*(\boldsymbol{y})$ of the function $\boldsymbol{g}^*(\boldsymbol{y})$. Since this holds for all $\boldsymbol{y}$, the optimal function $\boldsymbol{g}_i^*$ for the $i$-th component is:

$$\boldsymbol{g}_i^*(\boldsymbol{Y}) = \frac{\mathbb{E}[\boldsymbol{Z}_i^2\boldsymbol{X}_i \mid \boldsymbol{Y}]}{\mathbb{E}[\boldsymbol{Z}_i^2 \mid \boldsymbol{Y}]} \tag{38}$$

Assembling the components for $i = 1, \ldots, d$ into a single vector equation gives the expression for the optimal function $\boldsymbol{g}^*$:

$$\boldsymbol{g}^*(\boldsymbol{Y}) = \frac{\mathbb{E}[\boldsymbol{Z} \odot \boldsymbol{Z} \odot \boldsymbol{X} \mid \boldsymbol{Y}]}{\mathbb{E}[\boldsymbol{Z} \odot \boldsymbol{Z} \mid \boldsymbol{Y}]} \tag{39}$$

where the division is understood to be element-wise. $\qquad\square$

Having established the fundamental relationship between optimal model outputs and query probabilities in the lemma, we now proceed to prove our main theorem.

**Theorem 1** (Optimal solution under context masking without query information). *Let $\boldsymbol{x}_{\boldsymbol{\theta}}^*$ be the optimal solution by minimizing the loss in equation 1. Under the conditional independence of masks and data, we have the following results:*

*(i) Optimal solution: The optimal solution is given by*

$$(\boldsymbol{x}_{\boldsymbol{\theta}}(t, \boldsymbol{M}_{ctx} \odot \boldsymbol{x}_{obs,t}, \boldsymbol{M}_{ctx}))_i = \begin{cases} \mathbb{E}\left[(\boldsymbol{x}_0)_i \mid \boldsymbol{M}_{ctx} \odot \boldsymbol{x}_{obs,t}, \boldsymbol{M}_{ctx}\right], & P((\boldsymbol{M}_{qry})_i = 1 \mid \boldsymbol{M}_{ctx}) > 0 \\ \textit{an arbitrary value}, & P((\boldsymbol{M}_{qry})_i = 1 \mid \boldsymbol{M}_{ctx}) = 0 \end{cases} \tag{2}$$

where $i$ indicates the $i$-th entry of the vector. Specially, given the context mask $\boldsymbol{M}_{ctx}$, if the union of all possible query mask $\boldsymbol{M}_{qry}$ supports covers all spatial dimensions, we have

$$\boldsymbol{x}_{\boldsymbol{\theta}}\left(t, \boldsymbol{M}_{ctx} \odot \boldsymbol{x}_{obs,t}, \boldsymbol{M}_{ctx}\right) = \mathbb{E}\left[\boldsymbol{x}_0 \mid \boldsymbol{M}_{ctx} \odot \boldsymbol{x}_{obs,t}, \boldsymbol{M}_{ctx}\right] \tag{3}$$

*(ii) Gradient magnitude scaling: The expected squared gradient magnitude with respect to the network output for dimension $i$ scales linearly with the query probability $p_i := P((\boldsymbol{M}_{qry})_i = 1 \mid \boldsymbol{M}_{ctx})$:*

$$\mathbb{E}\left[\left(\frac{\partial \mathcal{L}}{\partial (\boldsymbol{x}_{\boldsymbol{\theta}})_i}\right)^2\right] = 4p_i \mathbb{E}\left[((\boldsymbol{x}_{\boldsymbol{\theta}})_i - (\boldsymbol{x}_{obs})_i)^2 \,\big|\, (\boldsymbol{M}_{qry})_i = 1\right] \tag{4}$$

*(iii) Parameter update frequency: The frequency of non-zero parameter updates for dimension $i$ is exactly $p_i$:*

$$P(\text{dimension } i \text{ contributes to parameter update}) = P((\boldsymbol{M}_{qry})_i = 1 \mid \boldsymbol{M}_{ctx}) = p_i \tag{5}$$

*Proof.* Given the training algorithm in Alg. 1, we have

$$\boldsymbol{\theta}^* = \arg\min_{\boldsymbol{\theta}} \mathbb{E}_{t, \boldsymbol{x}_{\text{obs}}, (\boldsymbol{M}_{\text{ctx}}, \boldsymbol{M}_{\text{qry}} \subseteq \boldsymbol{M})}\left[\|\boldsymbol{M}_{\text{qry}} \odot (\boldsymbol{x}_{\boldsymbol{\theta}}(t, \text{ctx}) - \boldsymbol{x}_{\text{obs}})\|^2\right] \tag{40a}$$

$$= \arg\min_{\boldsymbol{\theta}} \mathbb{E}_t \mathbb{E}_{\boldsymbol{x}_{\text{obs}}, (\boldsymbol{M}_{\text{ctx}}, \boldsymbol{M}_{\text{qry}} \subseteq \boldsymbol{M})}\left[\|\boldsymbol{M}_{\text{qry}} \odot (\boldsymbol{x}_{\boldsymbol{\theta}}(t, \text{ctx}) - \boldsymbol{M} \odot \boldsymbol{x}_0)\|^2\right] \tag{40b}$$

$$= \arg\min_{\boldsymbol{\theta}} \mathbb{E}_t \mathbb{E}_{\boldsymbol{x}_{\text{obs}}, (\boldsymbol{M}_{\text{ctx}}, \boldsymbol{M}_{\text{qry}} \subseteq \boldsymbol{M})}\left[\|\boldsymbol{M}_{\text{qry}} \odot \boldsymbol{x}_{\boldsymbol{\theta}}(t, \text{ctx}) - \boldsymbol{M}_{\text{qry}} \odot \boldsymbol{x}_0\|^2\right] \tag{40c}$$

**(i) Optimal solution:** When optimized, given $\forall \boldsymbol{M}_{\text{ctx}} \subseteq \boldsymbol{M}$, for any index $i$ such that $P((\boldsymbol{M}_{\text{qry}})_i = 1 \mid \boldsymbol{M}_{\text{ctx}}) > 0$ under the sampling distribution, we have

$$\boldsymbol{M}_{\text{qry}} \odot \boldsymbol{x}_{\boldsymbol{\theta}}(t, \text{ctx}) = \mathbb{E}\left[\boldsymbol{M}_{\text{qry}} \odot \boldsymbol{x}_0 \mid \text{ctx}, \boldsymbol{M}_{\text{qry}}\right] = \boldsymbol{M}_{\text{qry}} \odot \mathbb{E}\left[\boldsymbol{x}_0 \mid \text{ctx}\right] \tag{41}$$

Thus, $\boldsymbol{M}_{\text{qry}} \odot (\boldsymbol{x}_{\boldsymbol{\theta}}(t, \text{ctx}) - \mathbb{E}[\boldsymbol{x}_0 \mid \text{ctx}]) = \boldsymbol{0}$ for any $\boldsymbol{M}_{\text{qry}}$ in the support of the sampling distribution given $\boldsymbol{M}_{\text{ctx}}$.

**Case 1:** When $P((\boldsymbol{M}_{\text{qry}})_i = 1 \mid \boldsymbol{M}_{\text{ctx}}) > 0$. By applying Lemma 1 with $\boldsymbol{Z} = \boldsymbol{M}_{\text{qry}}$, $\boldsymbol{Y} = \text{ctx}$, $\boldsymbol{X} = \boldsymbol{x}_0$, and $\boldsymbol{g}(\cdot) = \boldsymbol{x}_{\boldsymbol{\theta}}(t, \cdot)$, the optimal solution is:

$$\boldsymbol{x}_{\boldsymbol{\theta}}^*(t, \text{ctx}) = \frac{\mathbb{E}[\boldsymbol{M}_{\text{qry}} \odot \boldsymbol{M}_{\text{qry}} \odot \boldsymbol{x}_0 \mid \text{ctx}]}{\mathbb{E}[\boldsymbol{M}_{\text{qry}} \odot \boldsymbol{M}_{\text{qry}} \mid \text{ctx}]} \tag{42}$$

Since $\boldsymbol{M}_{\text{qry}}$ is a binary mask where $(\boldsymbol{M}_{\text{qry}})_i \in \{0, 1\}$, we have $(\boldsymbol{M}_{\text{qry}})_i \odot (\boldsymbol{M}_{\text{qry}})_i = (\boldsymbol{M}_{\text{qry}})_i$. Thus:

$$\boldsymbol{x}_{\boldsymbol{\theta}}^*(t, \text{ctx}) = \frac{\mathbb{E}[\boldsymbol{M}_{\text{qry}} \odot \boldsymbol{x}_0 \mid \text{ctx}]}{\mathbb{E}[\boldsymbol{M}_{\text{qry}} \mid \text{ctx}]} \tag{43}$$

For component $i$ where $P((\boldsymbol{M}_{\text{qry}})_i = 1 \mid \boldsymbol{M}_{\text{ctx}}) > 0$:

$$(\boldsymbol{x}_{\boldsymbol{\theta}}^*(t, \text{ctx}))_i = \frac{\mathbb{E}[(\boldsymbol{M}_{\text{qry}})_i (\boldsymbol{x}_0)_i \mid \text{ctx}]}{\mathbb{E}[(\boldsymbol{M}_{\text{qry}})_i \mid \text{ctx}]} \tag{44a}$$

$$= \frac{\mathbb{E}[(\boldsymbol{M}_{\text{qry}})_i (\boldsymbol{x}_0)_i \mid \text{ctx}]}{P((\boldsymbol{M}_{\text{qry}})_i = 1 \mid \boldsymbol{M}_{\text{ctx}})} \tag{44b}$$

$$= \frac{\mathbb{E}[(\boldsymbol{x}_0)_i \mathbf{1}_{(\boldsymbol{M}_{\text{qry}})_i = 1} \mid \text{ctx}]}{P((\boldsymbol{M}_{\text{qry}})_i = 1 \mid \boldsymbol{M}_{\text{ctx}})} \tag{44c}$$

$$= \mathbb{E}[(\boldsymbol{x}_0)_i \mid \text{ctx}, (\boldsymbol{M}_{\text{qry}})_i = 1] \tag{44d}$$

$$= \mathbb{E}[(\boldsymbol{x}_0)_i \mid \text{ctx}] \tag{44e}$$

where the last equality holds because $\boldsymbol{x}_0$ is independent of $\boldsymbol{M}_{\text{qry}}$ given ctx.

**Case 2:** When $P((\boldsymbol{M}_{\text{qry}})_i = 1 \mid \boldsymbol{M}_{\text{ctx}}) = 0$. In this case, $(\boldsymbol{M}_{\text{qry}})_i = 0$ almost surely given $\boldsymbol{M}_{\text{ctx}}$. The contribution of index $i$ to the loss function is:

$$\mathbb{E}[|(\boldsymbol{M}_{\text{qry}})_i (\boldsymbol{x}_{\boldsymbol{\theta}}(t, \text{ctx}))_i - (\boldsymbol{M}_{\text{qry}})_i (\boldsymbol{x}_0)_i|^2 \mid \text{ctx}]$$

$$= \mathbb{E}[|0 \cdot (\boldsymbol{x}_{\boldsymbol{\theta}}(t, \text{ctx}))_i - 0 \cdot (\boldsymbol{x}_0)_i|^2 \mid \text{ctx}] \tag{45a}$$

$$= 0 \tag{45b}$$

Therefore, $(x_{\theta}(t, \text{ctx}))_i$ does not affect the loss function and can take any arbitrary value.

Thus, we have

$$(x_{\theta}(t, \text{ctx}))_i = \begin{cases} \mathbb{E}\left[(x_0)_i \mid \text{ctx}\right], & P((M_{\text{qry}})_i = 1 \mid M_{\text{ctx}}) > 0 \\ \text{an arbitrary value}, & P((M_{\text{qry}})_i = 1 \mid M_{\text{ctx}}) = 0 \end{cases} \quad (46)$$

If the union of all possible query mask $M_{\text{qry}}$ supports covers all spatial dimensions, we have $\bigcup_{M_{\text{qry}} \text{ possible}} \text{supp}(M_{\text{qry}}) = \{1, \ldots, \dim(x_0)\}$. Thus, $\forall i, P((M_{\text{qry}})_i = 1) > 0$ and

$$x_{\theta}(t, \text{ctx}) = \mathbb{E}\left[x_0 \mid \text{ctx}\right] \quad (47)$$

**(ii) Gradient magnitude scaling:** The gradient of the loss with respect to the $i$-th output component is:

$$\frac{\partial \mathcal{L}}{\partial (x_{\theta})_i} = 2(M_{\text{qry}})_i \cdot ((x_{\theta})_i - (x_{\text{obs}})_i) \quad (48)$$

Taking expectation over the sampling distribution:

$$\mathbb{E}\left[\left(\frac{\partial \mathcal{L}}{\partial (x_{\theta})_i}\right)^2\right] = \mathbb{E}\left[4(M_{\text{qry}})_i^2 \cdot ((x_{\theta})_i - (x_{\text{obs}})_i)^2\right] \quad (49a)$$

$$= 4\mathbb{E}\left[(M_{\text{qry}})_i \cdot ((x_{\theta})_i - (x_{\text{obs}})_i)^2\right] \quad (49b)$$

$$= 4p_i\mathbb{E}\left[((x_{\theta})_i - (x_{\text{obs}})_i)^2 \Big| (M_{\text{qry}})_i = 1\right] \quad (49c)$$

Let $C_i := \mathbb{E}\left[((x_{\theta})_i - (x_{\text{obs}})_i)^2 \Big| (M_{\text{qry}})_i = 1\right]$. Then:

$$\mathbb{E}\left[\left(\frac{\partial \mathcal{L}}{\partial (x_{\theta})_i}\right)^2\right] = 4p_i C_i \quad (50)$$

establishing linear scaling with $p_i$.

**(iii) Parameter update frequency:** At each training step, the gradient contribution from dimension $i$ is:

$$\left(\frac{\partial \mathcal{L}}{\partial \theta}\right)_i = \frac{\partial \mathcal{L}}{\partial (x_{\theta})_i} \frac{\partial (x_{\theta})_i}{\partial \theta} \quad (51)$$

This contribution is non-zero if and only if $(M_{\text{qry}})_i = 1$, which occurs with probability $p_i$. Therefore:

$$P(\text{dimension } i \text{ contributes to parameter update}) = P((M_{\text{qry}})_i = 1 \mid M_{\text{ctx}}) = p_i \quad (52)$$

$\square$

## F.2 PARTIALLY OBSERVED MASK CONVERGENCE THEOREM

**Theorem 2** (Ensemble approximation convergence). *Let $\mathbb{E}[x_0 \mid \text{obs}] := \mathbb{E}[x_0 \mid x_{\text{obs},t}, M]$ be the ground truth conditional expectation and $\mathbb{E}[x_0 \mid \text{ctx}] = \mathbb{E}[x_0 \mid M_{\text{ctx}} \odot x_{obs,t}, M_{ctx}]$ be the context-conditioned expectation. The expected squared error between these quantities is:*

$$\mathbb{E}\left[\|\mathbb{E}[x_0 \mid \text{ctx}] - \mathbb{E}[x_0 \mid \text{obs}]\|^2\right] = \mathbb{E}[\text{Var}[x_0 \mid \text{ctx}]] - \mathbb{E}[\text{Var}[x_0 \mid \text{obs}]] \quad (11)$$

*Consider a practical model with output $x_{\theta}(t, \text{ctx}) = \mathbb{E}[x_0 \mid \text{ctx}] + b(\text{ctx}) + \epsilon_{bias}(\text{ctx})$, where $b(\text{ctx})$ represents context-dependent deterministic bias and $\epsilon_{bias}(\text{ctx})$ is random error with $\mathbb{E}[\epsilon_{bias}] = 0$. Given the ensemble prediction in equation 10 as:*

$$\hat{\mu}_K = \frac{1}{K} \sum_{k=1}^{K} x_{\theta}(t, \text{ctx}^{(k)}), \quad (12)$$

*the expected squared error between the ensemble prediction and ground truth is:*

$$\mathbb{E}\left[\|\hat{\mu}_K - \mathbb{E}[x_0 \mid \text{obs}]\|^2\right] = \|\underbrace{\mathbb{E}[\mathbb{E}[x_0 \mid \text{ctx}]] - \mathbb{E}[x_0 \mid \text{obs}]}_{\text{information gap}} + \underbrace{\mathbb{E}[b(\text{ctx})]}_{\text{model bias}}\|^2$$

$$+ \frac{1}{K}\left(\underbrace{\text{Var}[\mathbb{E}[x_0 \mid \text{ctx}]]}_{\text{data variance}} + \underbrace{\text{Var}[b(\text{ctx})] + \text{Var}[\epsilon_{bias}]}_{\text{model variance}}\right) \quad (13)$$

*As $K \to \infty$, the ensemble converges to:*

$$\lim_{K \to \infty} \mathbb{E}\left[\|\hat{\boldsymbol{\mu}}_K - \mathbb{E}[\boldsymbol{x}_0 \mid \text{obs}]\|^2\right] = \|\mathbb{E}[\mathbb{E}[\boldsymbol{x}_0 \mid \text{ctx}]] - \mathbb{E}[\boldsymbol{x}_0 \mid \text{obs}] + \mathbb{E}[\boldsymbol{b}(\text{ctx})]\|^2 \tag{14}$$

*Proof.* Define random variables $\boldsymbol{\mu} = \mathbb{E}[\boldsymbol{x}_0 \mid \text{obs}]$ and $\boldsymbol{\mu}_{\text{c}} = \mathbb{E}[\boldsymbol{x}_0 \mid \text{ctx}]$. Our goal is to compute $\mathbb{E}\left[\|\boldsymbol{\mu} - \boldsymbol{\mu}_{\text{c}}\|^2\right]$.

Applying the Law of Total Variance, we have

$$\text{Var}[\boldsymbol{x}_0 \mid \text{ctx}] = \mathbb{E}[\text{Var}[\boldsymbol{x}_0 \mid \text{obs}] \mid \text{ctx}] + \text{Var}[\mathbb{E}[\boldsymbol{x}_0 \mid \text{obs}] \mid \text{ctx}] \tag{53a}$$

$$= \mathbb{E}[\text{Var}[\boldsymbol{x}_0 \mid \text{obs}] \mid \text{ctx}] + \text{Var}[\boldsymbol{\mu} \mid \text{ctx}] \tag{53b}$$

Rearranging:

$$\text{Var}[\boldsymbol{\mu} \mid \text{ctx}] = \text{Var}[\boldsymbol{x}_0 \mid \text{ctx}] - \mathbb{E}[\text{Var}[\boldsymbol{x}_0 \mid \text{obs}] \mid \text{ctx}] \tag{54}$$

Then:

$$\mathbb{E}\left[\|\boldsymbol{\mu} - \boldsymbol{\mu}_{\text{c}}\|^2\right]$$

$$= \mathbb{E}\left[\mathbb{E}\left[\|\boldsymbol{\mu} - \boldsymbol{\mu}_{\text{c}}\|^2 \mid \text{ctx}\right]\right] \qquad \text{Law of total expectation} \tag{55a}$$

$$= \mathbb{E}\left[\mathbb{E}\left[\|\boldsymbol{\mu} - \mathbb{E}[\boldsymbol{x}_0 \mid \text{ctx}]\|^2 \mid \text{ctx}\right]\right] \qquad \text{By definition of } \boldsymbol{\mu}_{\text{c}} \tag{55b}$$

$$= \mathbb{E}\left[\mathbb{E}\left[\|\boldsymbol{\mu} - \mathbb{E}[\mathbb{E}[\boldsymbol{x}_0 \mid \text{obs}] \mid \text{ctx}]\|^2 \mid \text{ctx}\right]\right] \qquad \text{Tower rule} \tag{55c}$$

$$= \mathbb{E}\left[\mathbb{E}\left[\|\boldsymbol{\mu} - \mathbb{E}[\boldsymbol{\mu} \mid \text{ctx}]\|^2 \mid \text{ctx}\right]\right] \qquad \text{By definition of } \boldsymbol{\mu} \tag{55d}$$

$$= \mathbb{E}[\text{Var}[\boldsymbol{\mu} \mid \text{ctx}]] \qquad \text{Definition of conditional variance} \tag{55e}$$

$$= \mathbb{E}[\text{Var}[\boldsymbol{x}_0 \mid \text{ctx}] - \mathbb{E}[\text{Var}[\boldsymbol{x}_0 \mid \text{obs}] \mid \text{ctx}]] \qquad \text{equation 54} \tag{55f}$$

$$= \mathbb{E}[\text{Var}[\boldsymbol{x}_0 \mid \text{ctx}]] - \mathbb{E}[\text{Var}[\boldsymbol{x}_0 \mid \text{obs}]] \qquad \text{Linearity of expectation} \tag{55g}$$

Further define $\boldsymbol{\mu}_k = \boldsymbol{x}_{\boldsymbol{\theta}}\left(t, \text{ctx}^{(k)}\right) = \mathbb{E}\left[\boldsymbol{x}_0 \mid \text{ctx}^{(k)}\right] + \boldsymbol{b}\left(\text{ctx}^{(k)}\right) + \boldsymbol{\epsilon}_{\text{bias}}\left(\text{ctx}^{(k)}\right)$, where $\boldsymbol{b}$ is the systematic bias and $\boldsymbol{\epsilon}_{\text{bias}}$ is the random error with $\mathbb{E}[\boldsymbol{\epsilon}_{\text{bias}}] = \boldsymbol{0}$. The ensemble average is:

$$\hat{\boldsymbol{\mu}}_K = \frac{1}{K}\sum_{k=1}^{K}\boldsymbol{\mu}_k = \frac{1}{K}\sum_{k=1}^{K}\mathbb{E}[\boldsymbol{x}_0 \mid \text{ctx}^{(k)}] + \frac{1}{K}\sum_{k=1}^{K}\boldsymbol{b}(\text{ctx}^{(k)}) + \frac{1}{K}\sum_{k=1}^{K}\boldsymbol{\epsilon}(\text{ctx}^{(k)}) \tag{56}$$

Computing the expectation:

$$\mathbb{E}[\hat{\boldsymbol{\mu}}_K] = \mathbb{E}\left[\frac{1}{K}\sum_{k=1}^{K}\boldsymbol{\mu}_k\right] \tag{57a}$$

$$= \frac{1}{K}\sum_{k=1}^{K}\mathbb{E}[\mathbb{E}[\boldsymbol{x}_0 \mid \text{ctx}^{(k)}] + \boldsymbol{b}(\text{ctx}^{(k)}) + \boldsymbol{\epsilon}(\text{ctx}^{(k)})] \tag{57b}$$

$$= \mathbb{E}[\mathbb{E}[\boldsymbol{x}_0 \mid \text{ctx}]] + \mathbb{E}[\boldsymbol{b}(\text{ctx})] + \mathbb{E}[\boldsymbol{\epsilon}(\text{ctx})] \tag{57c}$$

$$= \mathbb{E}[\mathbb{E}[\boldsymbol{x}_0 \mid \text{ctx}]] + \mathbb{E}[\boldsymbol{b}(\text{ctx})] \tag{57d}$$

The bias of the ensemble estimator is:

$$\text{Bias}(\hat{\boldsymbol{\mu}}_K) = \mathbb{E}[\hat{\boldsymbol{\mu}}_K] - \mathbb{E}[\boldsymbol{x}_0 \mid \text{obs}] = \mathbb{E}[\mathbb{E}[\boldsymbol{x}_0 \mid \text{ctx}]] - \mathbb{E}[\boldsymbol{x}_0 \mid \text{obs}] + \mathbb{E}[\boldsymbol{b}(\text{ctx})] \tag{58}$$

For the variance, decompose each component:

$$\boldsymbol{\mu}_k - \mathbb{E}[\boldsymbol{\mu}_k] = \mathbb{E}[\boldsymbol{x}_0 \mid \text{ctx}^{(k)}] + \boldsymbol{b}(\text{ctx}^{(k)}) + \boldsymbol{\epsilon}(\text{ctx}^{(k)}) - \mathbb{E}[\mathbb{E}[\boldsymbol{x}_0 \mid \text{ctx}]] - \mathbb{E}[\boldsymbol{b}(\text{ctx})] \tag{59a}$$

$$= \underbrace{(\mathbb{E}[\boldsymbol{x}_0 \mid \text{ctx}^{(k)}] - \mathbb{E}[\mathbb{E}[\boldsymbol{x}_0 \mid \text{ctx}]])}_{\text{data variation}} + \underbrace{(\boldsymbol{b}(\text{ctx}^{(k)}) - \mathbb{E}[\boldsymbol{b}(\text{ctx})])}_{\text{bias variation}} + \underbrace{\boldsymbol{\epsilon}(\text{ctx}^{(k)})}_{\text{random error}} \tag{59b}$$

The variance of the ensemble average:

$$\text{Var}(\hat{\boldsymbol{\mu}}_K) = \mathbb{E}\left[\|\hat{\boldsymbol{\mu}}_K - \mathbb{E}[\hat{\boldsymbol{\mu}}_K]\|^2\right] \tag{60a}$$

$$= \mathbb{E}\left[\left\|\frac{1}{K}\sum_{k=1}^{K}(\boldsymbol{\mu}_k - \mathbb{E}[\boldsymbol{\mu}_k])\right\|^2\right] \tag{60b}$$

$$= \frac{1}{K^2}\mathbb{E}\left[\sum_{k=1}^{K}\|\boldsymbol{\mu}_k - \mathbb{E}[\boldsymbol{\mu}_k]\|^2 + 2\sum_{i<j}\langle\boldsymbol{\mu}_i - \mathbb{E}[\boldsymbol{\mu}_i], \boldsymbol{\mu}_j - \mathbb{E}[\boldsymbol{\mu}_j]\rangle\right] \tag{60c}$$

Since $\text{ctx}^{(1)}, \ldots, \text{ctx}^{(K)}$ are conditionally independent given $\boldsymbol{x}_{\text{obs},t}$ and $\boldsymbol{M}$, the data variations and bias variations are independent across different $k$. Additionally, under the assumption that $\boldsymbol{\epsilon}(\text{ctx}^{(i)})$ and $\boldsymbol{\epsilon}(\text{ctx}^{(j)})$ are independent for $i \neq j$:

$$\mathbb{E}[\langle \boldsymbol{\mu}_i - \mathbb{E}[\boldsymbol{\mu}_i], \boldsymbol{\mu}_j - \mathbb{E}[\boldsymbol{\mu}_j] \rangle] = 0 \quad \forall i \neq j \tag{61}$$

Therefore:

$$\text{Var}(\hat{\boldsymbol{\mu}}_K) = \frac{1}{K^2} \sum_{k=1}^{K} \mathbb{E}[\|\boldsymbol{\mu}_k - \mathbb{E}[\boldsymbol{\mu}_k]\|^2] \tag{62a}$$

$$= \frac{1}{K} \mathbb{E}\left[\|(\mathbb{E}[\boldsymbol{x}_0 \mid \text{ctx}] - \mathbb{E}[\mathbb{E}[\boldsymbol{x}_0 \mid \text{ctx}]]) + (\boldsymbol{b}(\text{ctx}) - \mathbb{E}[\boldsymbol{b}(\text{ctx})]) + \boldsymbol{\epsilon}(\text{ctx})\|^2\right] \tag{62b}$$

$$= \frac{1}{K} \left(\text{Var}[\mathbb{E}[\boldsymbol{x}_0 \mid \text{ctx}]] + \text{Var}[\boldsymbol{b}(\text{ctx})] + \text{Var}[\boldsymbol{\epsilon}]\right) \tag{62c}$$

Here, we used the independence between the three components to separate the variances. Combining bias and variance using the bias-variance decomposition:

$$\mathbb{E}\left[\|\hat{\boldsymbol{\mu}}_K - \mathbb{E}[\boldsymbol{x}_0 \mid \text{obs}]\|^2\right]$$

$$= \|\text{Bias}(\hat{\boldsymbol{\mu}}_K)\|^2 + \text{Var}(\hat{\boldsymbol{\mu}}_K) \tag{63a}$$

$$= \|\mathbb{E}[\mathbb{E}[\boldsymbol{x}_0 \mid \text{ctx}]] - \mathbb{E}[\boldsymbol{x}_0 \mid \text{obs}] + \mathbb{E}[\boldsymbol{b}(\text{ctx})]\|^2$$

$$+ \frac{1}{K} \left(\text{Var}[\mathbb{E}[\boldsymbol{x}_0 \mid \text{ctx}]] + \text{Var}[\boldsymbol{b}(\text{ctx})] + \text{Var}[\boldsymbol{\epsilon}]\right) \tag{63b}$$

Taking the limit as $K \to \infty$:

$$\lim_{K \to \infty} \mathbb{E}\left[\|\hat{\boldsymbol{\mu}}_K - \mathbb{E}[\boldsymbol{x}_0 \mid \text{obs}]\|^2\right] = \|\mathbb{E}[\mathbb{E}[\boldsymbol{x}_0 \mid \text{ctx}]] - \mathbb{E}[\boldsymbol{x}_0 \mid \text{obs}] + \mathbb{E}[\boldsymbol{b}(\text{ctx})]\|^2 \tag{64}$$

This establishes that:

- The average bias $\mathbb{E}[\boldsymbol{b}(\text{ctx})]$ across all contexts is not reduced by ensemble averaging.

- The variance of the context-dependent bias $\text{Var}[\boldsymbol{b}(\text{ctx})]$ is reduced by a factor of $1/K$.

- Both data variance and random error variance are also reduced by $1/K$.

- The asymptotic error includes both the data bias and the model's average bias.

$\square$

## G  DISCUSSION ON INAPPLICABILITY OF BASELINES

### G.1  INAPPLICABILITY OF KNEWIMP

We discuss the inapplicability of the primary method proposed in Chen et al. (2024b), Kernelized Negative Entropy-regularized Wasserstein gradient flow Imputation (KnewImp), as a baseline for our high-dimensional PDE dynamics task.

**Principle.** KnewImp is an approach explicitly designed for the imputation of numerical tabular datasets. The method reformulates the imputation problem within the framework of Wasserstein Gradient Flow (WGF). Its core contribution is to derive a closed-form, implementable imputation procedure by optimizing a novel Negative Entropy-Regularized (NER) cost functional within a Reproducing Kernel Hilbert Space (RKHS).

The final imputation procedure involves simulating an ODE $\frac{d\boldsymbol{X}^{(\text{miss})}}{d\tau} = u(\boldsymbol{X}^{(\text{joint})}, \tau)$, where the velocity field $u(\boldsymbol{X}^{(\text{joint})}, \tau)$ is defined using a kernel function:

$$u(\boldsymbol{X}^{(\text{joint})}, \tau) = \mathbb{E}_{r(\tilde{\boldsymbol{X}}^{(\text{joint})}, \tau)} \left\{ \begin{array}{l} -\lambda \nabla_{\tilde{\boldsymbol{X}}^{(\text{miss})}} \mathcal{K}(\boldsymbol{X}^{(\text{joint})}, \tilde{\boldsymbol{X}}^{(\text{joint})}) \\ +[\nabla_{\tilde{\boldsymbol{X}}^{(\text{miss})}} \log \hat{p}(\tilde{\boldsymbol{X}}^{(\text{joint})})]^\top \mathcal{K}(\boldsymbol{X}^{(\text{joint})}, \tilde{\boldsymbol{X}}^{(\text{joint})}) \end{array} \right\} \tag{65}$$

This velocity field depends on two components: (1) a kernel function $\mathcal{K}$, which the authors specify is a Radial Basis Function (RBF) kernel, $\mathcal{K}(\boldsymbol{X}, \tilde{\boldsymbol{X}}) := \exp(-\frac{||\boldsymbol{X}-\tilde{\boldsymbol{X}}||^2}{2h^2})$, and (2) an estimated score function $\nabla_{\boldsymbol{X}^{(\mathrm{miss})}} \log \hat{p}(\boldsymbol{X}^{(\mathrm{joint})})$, which is trained separately using DSM (Song et al., 2020).

**Reason for Unsuitability.** This method is mathematically and computationally infeasible for our task for two primary reasons:

- **Mathematical Infeasibility (Curse of Dimensionality):** The entire derivation of the implementable velocity field hinges on the use of an RBF kernel. This kernel's computation is based on the squared Euclidean distance $||\boldsymbol{X} - \tilde{\boldsymbol{X}}||^2$ between data points. Our PDE dynamics data has a dimensionality of $D = 100 \times 64 \times 64 = 409,600$. In such an extremely high-dimensional space, the concept of Euclidean distance becomes meaningless; all data points tend to become equidistant from one another. This "curse of dimensionality" would cause the RBF kernel to lose all discriminative power, rendering the velocity field calculation mathematically unstable and uninformative. The KnewImp method is fundamentally structured for low-dimensional tabular data, where distance metrics remain meaningful.

- **Prohibitive Computational Cost (VRAM and Time):** The method's two-stage process scales intractably with dimension $D$.
  - The "Estimate" phase requires training a score network $\nabla_{\boldsymbol{X}^{(\mathrm{joint})}} \log \hat{p}(\boldsymbol{X}^{(\mathrm{joint})})$ via DSM. Training any neural architecture (U-Net, Transformer, etc.) to model the score of a $D = 409,600$ dimensional vector would require prohibitive amounts of VRAM just to store the activations and gradients for a single batch.
  - The "Impute" phase requires simulating an ODE. Each step of this simulation necessitates a full computation of the velocity field. This computation involves a Monte Carlo estimation over the dataset, with each sample calculation requiring an expensive kernel evaluation in $D$-dimensional space.

This combination of a memory-intensive score network and a computationally-intensive, kernel-based ODE simulation makes the KnewImp approach computationally infeasible for our high-dimensional spatiotemporal task.

## G.2 Inapplicability of Score Matching with Missing Data

We discuss the inapplicability of the two primary methods proposed in Givens et al. (2025) as baselines for our high-dimensional PDE generation task.

### G.2.1 Method 1: marginal importance weighting (Marg-IW)

**Principle.** This approach (Algorithm 1) adapts the score matching objective to work with missing data by defining a marginal Fisher divergence. It then estimates the intractable marginal scores by applying importance weighting (IW) to approximate the high-dimensional integral over the missing coordinates using Monte Carlo samples.

**Reason for Unsuitability.** This method is unsuitable as it is not designed for high-dimensional data. The authors of the original paper explicitly state that the IW approach "will struggle in higher dimensional scenarios" and primarily demonstrate its efficacy in "lower dimensional settings". Given the dimensionality of our task, the variance and bias from the IW estimator would render the optimization intractable.

### G.2.2 Method 2: marginal variational score matching (Marg-Var)

**Principle.** This more complex approach (Algorithm 2) avoids direct IW estimation by first taking the gradient of the loss objective (a "gradient-first" approach). It then introduces a secondary variational neural network ($p'_\phi$) to approximate the conditional expectations and covariances over the missing data. The training involves a nested optimization, where the main score model ($s_\theta$) and the variational "helper" model ($p'_\phi$) are updated iteratively.

**Reason for Unsuitability.** This method is computationally infeasible for our task due to pro­hibitive VRAM and time costs. Our backbone model (analogous to $s_\theta$) already consumes $\sim$70GB of VRAM for a single forward-backward pass with a batch size of 8 on an 80GB A800 GPU. The Marg-Var algorithm would introduce, at a minimum:

- **Dual Model Memory Cost:** The algorithm requires maintaining two large neural net­works—the primary score model $s_\theta$ and the variational helper $p'_\phi$, simultaneously in VRAM. This alone would exceed the 80GB capacity.

- **Peak Gradient Memory Cost:** The gradient calculation for $s_\theta$ (Eq. 10/11) is exceptionally complex. It requires computing expectations and covariances that involve components from *both* models, necessitating that the computational graphs of both networks are active for the joint gradient estimation. This leads to a peak VRAM usage far exceeding the simple sum of the two models.

- **Multiplied Training Time Cost:** The algorithm employs a nested optimization loop. For *each* gradient step of the main model $s_\theta$, the helper model $p'_\phi$ must be trained for $L$ steps (e.g., $L = 10$ in the paper's experiments). This $L$-fold multiplication of an already lengthy training step makes the method impractical for large-scale generative modeling.

### G.2.3 DIMENSIONALITY MISMATCH

The core issue is the extreme discrepancy in data dimensionality. The experiments in Givens et al. (2025) are conducted on tasks with dimensions of 10 (Gaussian), up to 50 (Non-Gaussian), 100 (S&P 100), and 106 (Yeast). Our PDE dataset has a dimensionality of $100 \times 64 \times 64 = 409,600$. This is more than three orders of magnitude larger than the highest-dimensional task (106-dim) on which the Marg-Var method was validated. The computational and statistical challenges of this scale are not addressed by these methods.

## H  SUPPLEMENTARY EXPERIMENTS

### H.1  DATASET SETTINGS

**Shallow Water and Advection datasets.** We evaluate our approach on two fundamental geophys­ical PDE systems with distinct characteristics:

$$\text{Shallow Water:} \quad \partial_t u = fv - g\partial_x h, \quad \partial_t v = -fu - g\partial_y h, \quad \partial_t h = -H(\partial_x u + \partial_y v) \quad \text{(66a)}$$

$$\text{Advection:} \quad \partial_t u(t,x) + \beta \partial_x u(t,x) = 0 \quad \text{(66b)}$$

where for the shallow water system, $u$ and $v$ represent velocity components, $h$ denotes height field, $f$ is the Coriolis parameter, $g$ is gravitational acceleration, and $H$ is mean depth. For the advection equation, $u$ represents a scalar field being transported and $\beta$ is the advection velocity. We gener­ate synthetic solutions by randomly sampling physical parameters, along with randomized initial conditions to ensure dataset diversity. Each dataset contains 5k training, 1k validation, and 1k test samples with $32 \times 32$ spatial resolution and 50 temporal frames. We generate synthetic solutions by randomly sampling physical parameters and initial conditions. For evaluation, we employ comple­mentary metrics to comprehensively assess reconstruction quality across different physical aspects. For the Shallow Water dataset, we evaluate PDE feasibility loss, which measures how well the re­constructed solutions satisfy the underlying shallow water equations by computing the residual error when the reconstructed fields are substituted into the governing PDEs. For the Advection dataset, we reconstruct complete initial conditions from partial observations using our trained model, then propagate these reconstructions forward using traditional PDE solvers (finite difference schemes) to generate temporal sequences. Performance is measured by computing MSE between our recon­structed solutions and ground truth sequences over all 50 time steps. This dual evaluation strategy demonstrates our method's superiority across multiple physically meaningful criteria.

**Navier-Stokes dataset.** We evaluate our approach on the incompressible Navier-Stokes equations for isotropic turbulence. The governing equations are:

$$\partial_t \boldsymbol{u} + (\boldsymbol{u} \cdot \nabla)\boldsymbol{u} = -\nabla p + \nu \nabla^2 \boldsymbol{u} + \boldsymbol{f}, \quad \text{(67a)}$$

$$\nabla \cdot \boldsymbol{u} = 0 \quad \text{(67b)}$$

where $\boldsymbol{u} = (u, v)$ represents the velocity field, $p$ is pressure, $\nu$ is kinematic viscosity, and $\boldsymbol{f}$ is external forcing. The data are generated using either pseudo-spectral solvers with 4th-order Runge-Kutta or higher-order Finite Volume IMEX methods. Initial conditions with varying peak wavenumbers eventually evolve to exhibit the Kolmogorov energy cascade. The dataset contains 1152 samples with a spatial resolution of $64 \times 64$ and temporal sequences of 100 frames. For evaluation, the model generates complete field reconstructions. Performance is measured by computing MSE between reconstructed fields and ground truth across all spatial locations and time steps.

**ERA5 dataset.** We evaluate our approach on the ERA5 reanalysis dataset, which provides comprehensive atmospheric and surface meteorological variables. ERA5 represents the fifth generation atmospheric reanalysis from the European Centre for Medium-Range Weather Forecasts (ECMWF), combining model data with observations through data assimilation to produce a globally complete and consistent dataset. We utilize one year of hourly data sampled at a spatial resolution of $103 \times 120$ grid points in latitude-longitude coordinates, creating temporal sequences of dimension $(365 \times 24, 103, 120)$. The data are segmented into 3-hour windows, and we only select 20% / 10% / 1% observed pixels for each sample. Our experiments incorporate nine essential atmospheric variables. These variables capture both surface conditions and atmospheric dynamics critical for weather prediction tasks. Performance is evaluated by computing reconstruction errors across all spatial locations and temporal frames.

## H.2 MASK SELECTION IMPLEMENTATION

We now present concrete algorithmic implementations of our strategic context-query partitioning framework for both pixel-level and block-wise missing patterns. Algorithm 5 details the pixel-level procedure, where each observed location in the observation mask $\boldsymbol{M}$ is independently selected as context or query through Bernoulli sampling with ratios $r_{\text{ctx}}$ and $r_{\text{qry}}$, ensuring that $\boldsymbol{M}_{\text{ctx}} \subseteq \boldsymbol{M}$ and $\boldsymbol{M}_{\text{qry}} \subseteq \boldsymbol{M}$. For the more realistic block-wise scenario depicted in Algorithm 6, we operate on spatial blocks. This block-based sampling preserves spatial continuity while guaranteeing that every observable dimension maintains a positive probability of being selected as query, directly implementing the principle from Theorem 1. Both procedures maintain the crucial property that $P((\boldsymbol{M}_{\text{qry}})_i = 1 \mid \boldsymbol{M}_{\text{ctx}}) > 0$ for all unobserved dimensions, thereby enabling effective learning from incomplete training data.

---

**Algorithm 5** Pixel-Level Context-Query Partitioning

---

**Require:** Observation mask $\boldsymbol{M} \in \{0, 1\}^d$, context ratio $r_{\text{ctx}} \in (0, 1)$, query ratio $r_{\text{qry}} \in (0, 1)$
**Ensure:** Context mask $\boldsymbol{M}_{\text{ctx}}$, query mask $\boldsymbol{M}_{\text{qry}}$
1: Initialize $\boldsymbol{M}_{\text{ctx}} \leftarrow \boldsymbol{0}$, $\boldsymbol{M}_{\text{qry}} \leftarrow \boldsymbol{0}$
2: **for** each spatial index $i \in \{1, \dots, d\}$ **do**
3:     **if** $M_i = 1$ **then**
4:         Sample $u \sim \text{Uniform}(0, 1)$
5:         **if** $u < r_{\text{ctx}}$ **then**
6:             $(\boldsymbol{M}_{\text{ctx}})_i \leftarrow 1$
7:         **end if**
8:         Sample $v \sim \text{Uniform}(0, 1)$
9:         **if** $v < r_{\text{qry}}$ **then**
10:            $(\boldsymbol{M}_{\text{qry}})_i \leftarrow 1$
11:         **end if**
12:     **end if**
13: **end for**
14: **return** $\boldsymbol{M}_{\text{ctx}}, \boldsymbol{M}_{\text{qry}}$

---

---

**Algorithm 6** Block-Wise Context-Query Partitioning (Integer-based)

---

**Require:** Observation mask grid $M_{\text{grid}} \in \{0,1\}^{3 \times 3}$ (e.g., 5 observed blocks out of 9), integer $k_{\text{ctx}}$ (number of context blocks, e.g., 4), integer $k_{\text{qry}}$ (number of query blocks, e.g., 1)
**Ensure:** Context mask $M_{\text{ctx}}$, query mask $M_{\text{qry}}$
    $\mathcal{B}_{\text{obs}} \leftarrow \{b \mid (M_{\text{grid}})_b = 1\}$
    Sample $\mathcal{B}_{\text{ctx}} \subseteq \mathcal{B}_{\text{obs}}$ uniformly with $|\mathcal{B}_{\text{ctx}}| = k_{\text{ctx}}$
    Sample $\mathcal{B}_{\text{qry}} \subseteq \mathcal{B}_{\text{obs}}$ uniformly with $|\mathcal{B}_{\text{qry}}| = k_{\text{qry}}$
    $M_{\text{ctx}} \leftarrow \text{BlocksToMask}(\mathcal{B}_{\text{ctx}})$
    $M_{\text{qry}} \leftarrow \text{BlocksToMask}(\mathcal{B}_{\text{qry}})$
    **return** $M_{\text{ctx}}, M_{\text{qry}}$

---

### H.3 ANALYSIS OF MISSDIFF BASELINE AND DATA MATCHING ADAPTATION

A notable aspect of our experimental setup is the adaptation of the MissDiff baseline (Ouyang et al., 2023) from its original noise matching objective to a data matching framework. This modification was empirically necessary, as the original objective proved ineffective in our experimental context. This adaptation facilitates a meaningful and fair comparison by ensuring the baseline can operate effectively on our challenging datasets.

**Initial failure of the noise matching objective.** An initial evaluation of MissDiff with its original noise matching objective showed that the training loss failed to decrease at all in our experimental setting. This failure is attributed to a fundamental difference between the data domains: the tabular data used in the original MissDiff paper and the spatiotemporal data used in our work.

- **Tabular data (original MissDiff domain):** The MissDiff paper focused on tabular data with relatively moderate missing ratios. In this context, each entry often represents an independent feature. Missing one entry means completely losing information about that specific feature.

- **Spatiotemporal data (our domain):** Our PDE datasets involve spatiotemporal fields (e.g., images/videos) characterized by much higher missing data ratios (down to 1% observed data). Critically, in these physical fields, a missing pixel does not represent the loss of an independent feature. Due to the inherent spatial smoothness and continuity of physical systems, neighboring pixels carry highly correlated information.

**Implications for diffusion objectives.** This fundamental data difference has profound implications for the suitability of noise matching versus data matching:

- **Data matching (our adaptation):** This objective (predicting $x_0$) can effectively leverage the spatial smoothness priors. Even from sparse observations, the model can learn to interpolate and predict reasonable values for missing regions by exploiting the correlated context.

- **Noise matching (original MissDiff):** This objective (predicting $\epsilon$) requires the model to predict fine-grained noise patterns. This task demands much denser observations to capture the necessary local structure. At extreme sparsity (e.g., 1% observed), the noise prediction task becomes ill-posed. There is simply insufficient local context to distinguish signal from noise, making the learning target ambiguous.

Our empirical findings show that in our setting, the original noise matching objective led to MissDiff completely failing to learn (e.g., outputting all zeros). The adaptation to a data matching framework allows MissDiff to produce meaningful predictions by leveraging the smoothness priors inherent in physical systems. Therefore, this modification was essential for a valid and fair comparison. Without this adaptation, MissDiff would be unable to generate meaningful results in our experimental scenarios, rendering the comparison ineffective.

### H.4 ABLATION STUDY

We conduct ablation studies to validate the effectiveness of key components in our proposed method.

**Test-time gap introduced by replacing $M_{\text{ctx}}$ with $M$.** Our sampling procedure requires multiple context masks $M_{\text{ctx}}$ to estimate $\mathbb{E}\left[x_0 \mid x_{\text{obs},t}, M\right] \approx \frac{1}{K} \sum_{k=1}^{K} x_{\theta}\left(t, M_{\text{ctx}}^{(k)} \odot x_t, M_{\text{ctx}}^{(k)}\right)$. This ablation study compares our method against directly computing $\mathbb{E}\left[x_0 \mid x_{\text{obs},t}, M\right]$ using $x_{\theta}\left(t, M \odot x_t, M\right)$. The direct approach creates a distributional mismatch: during training, the model's input mask follows the distribution of $M_{\text{ctx}}$, but during sampling, the input becomes $M$. This mismatch degrades model performance. Tab. 5 presents experimental results comparing both methods.

Table 5: Performance comparison of two approaches: (1) imputation with multiple time sampling of $M_{\text{ctx}}$ followed by ensemble prediction (Theorem 2), versus (2) directly using $M$ as $M_{\text{ctx}}$, which creates a distributional mismatch between training and testing inputs.

| Method | Shallow Water | | Advection | |
|---|---|---|---|---|
| | 80% | 30% | 80% | 30% |
| Ours ($M$) | $2.3983 \pm {\scriptstyle 0.7880}$ | $2.6717 \pm {\scriptstyle 1.4731}$ | $0.1320 \pm {\scriptstyle 0.0155}$ | $0.1655 \pm {\scriptstyle 0.0537}$ |
| Ours ($M_{\text{ctx}}$) | $\mathbf{0.1878} \pm {\scriptstyle \mathbf{0.0054}}$ | $\mathbf{0.7379} \pm {\scriptstyle \mathbf{0.1101}}$ | $\mathbf{0.1035} \pm {\scriptstyle \mathbf{0.0008}}$ | $\mathbf{0.1189} \pm {\scriptstyle \mathbf{0.0069}}$ |

**Backbone architecture.** To demonstrate the generalizability of our method across different architectures, we evaluate both our proposed approach and baseline methods using two distinct backbones: Karras UNet (Karras et al., 2024) and Fourier Neural Operator (FNO) (Li et al., 2020). For the FNO implementation, we concatenate diffusion time embeddings along the channel dimension. Results are presented in Tab. 6. Our findings show that U-Net and FNO achieve comparable performance on the Shallow Water and Advection datasets, while U-Net outperforms FNO on the Navier-Stokes and ERA5 datasets, where FNO fails to generate reasonable samples.

Table 6: Performance comparison across backbone architectures. Results for our method and baselines using Karras UNet (Karras et al., 2024) and FNO (Li et al., 2020) backbones across two PDE datasets.

| Method | Backbone | Shallow Water | | Advection | |
|---|---|---|---|---|---|
| | | 80% | 30% | 80% | 30% |
| MissDiff | UNet | $0.3963 \pm {\scriptstyle 0.0617}$ | $1.2570 \pm {\scriptstyle 0.2146}$ | $\mathbf{0.1030} \pm {\scriptstyle \mathbf{0.0004}}$ | $0.1197 \pm {\scriptstyle 0.0096}$ |
| | FNO | $0.2917 \pm {\scriptstyle 0.1683}$ | $0.7525 \pm {\scriptstyle 0.1529}$ | $0.1375 \pm {\scriptstyle 0.0063}$ | $0.4816 \pm {\scriptstyle 0.0187}$ |
| Ours | UNet | $0.3279 \pm {\scriptstyle 0.0655}$ | $0.9292 \pm {\scriptstyle 0.1963}$ | $0.1035 \pm {\scriptstyle 0.0008}$ | $\mathbf{0.1189} \pm {\scriptstyle \mathbf{0.0069}}$ |
| | FNO | $\mathbf{0.1869} \pm {\scriptstyle \mathbf{0.0015}}$ | $\mathbf{0.7379} \pm {\scriptstyle \mathbf{0.1101}}$ | $0.1240 \pm {\scriptstyle 0.0040}$ | $0.3527 \pm {\scriptstyle 0.0620}$ |

**Context and query mask ratio selection.** We conduct an ablation study examining how different choices of context and query mask ratios affect model performance. The results are presented in Table 7. We evaluate ratios ranging from 0.5 to 1.0 to understand the trade-offs between information availability and parameter update frequency identified in our theoretical analysis. As expected from our theoretical framework, intermediate ratios (0.5-0.9) achieve optimal performance by balancing the information gap and parameter update frequency trade-offs. Notably, when both context and query ratios are set to 100%, our proposed method reduces to the MissDiff baseline, providing a direct comparison point that validates our experimental setup.

**Optimal denoiser approximation.** We approximate the optimal denoiser $\mathbb{E}\left[x_0 \mid x_t, x_{\text{obs}}, M\right]$ through a weighted combination of diffusion expectation $\mathbb{E}\left[x_0 \mid x_t\right]$ and imputation expectation $\mathbb{E}\left[x_0 \mid x_{\text{obs}}, M\right]$ using empirical weight $\omega_t$ (equation 28). We investigate different weighting strategies to understand their impact on reconstruction quality during the multi-step generation process (200 steps). The results can be seen in Tab. 8.

Table 7: Performance comparison of context and query mask ratio.

| Context Ratio | Query Ratio | Navier-Stokes | | |
|---|---|---|---|---|
| | | 80% | 60% | 20% |
| 50% | 50% | 0.2383 | 0.5338 | 2.0924 |
| 70% | 70% | **0.2076** | 0.5336 | **2.0336** |
| 90% | 90% | 0.2252 | **0.5251** | 2.1309 |
| 70% | 100% | 0.2103 | 0.5276 | 2.1178 |
| 100% | 100% | 0.2444 | 0.7023 | 2.5599 |

Table 8: Impact of weighting strategies on optimal denoiser approximation.

| Method | Navier-Stokes | | |
|---|---|---|---|
| | 80% | 60% | 20% |
| w/o $\omega_t$ | $0.2334 \pm 0.0115$ | $0.5649 \pm 0.0329$ | $3.3820 \pm 0.1704$ |
| $\omega_t = t$ | $\mathbf{0.2331} \pm \mathbf{0.0117}$ | $\mathbf{0.5633} \pm \mathbf{0.0332}$ | $\mathbf{3.1881} \pm \mathbf{0.2170}$ |
| $\omega_t = t^2$ | $0.2334 \pm 0.0116$ | $0.5647 \pm 0.0333$ | $3.3557 \pm 0.1973$ |

**Influence of ensemble size $K$.** Tab. 9 shows that increasing $K$ consistently improves performance (see Theorem. 2). We use $K = 10$ by default to balance efficiency and accuracy.

**Training cost:** Our training procedure has a comparable computational cost to baseline diffusion methods (MissDiff, AmbientDiff) since the network architecture, input/output dimensions, and number of training steps are the same. The main difference is in our context-query partitioning strategy during training, which adds negligible overhead.

**Inference cost:** The additional computational cost comes from sampling:

- For *single sample generation* (common in scientific applications): The $K$ forward passes can be executed in parallel since they are independent. Wall-clock time increases sub-linearly with $K$ rather than $K$-fold. For example, on an A800 GPU, $K = 10$ requires $3.36\times$ the time of $K = 1$ for 50-frame $32 \times 32$ sequences, and $8.32\times$ for 100-frame $64 \times 64$ sequences. The overhead depends on hardware parallelization efficiency and batch size.

- For *batch generation* of multiple samples: The computational cost scales approximately $K$ times compared to baselines. This represents a fundamental trade-off: our method enables learning from realistically incomplete data, a necessity in many scientific domains where complete measurements are physically impossible.

Table 9: Impact of ensemble size $K$ on Navier-Stokes imputation. Errors decrease with larger $K$ but with diminishing returns. Time cost is measured for single-sample forward passes on a single GPU.

| Ensemble size $K$ | Navier-Stokes ($\times 10^{-3}$) | | | Time Ratio |
|---|---|---|---|---|
| | 80% | 60% | 20% | |
| $K = 1$ | 0.2239 | 0.5652 | 2.1446 | $1.00\times$ |
| $K = 2$ | 0.2147 | 0.5475 | 2.0822 | $1.81\times$ |
| $K = 3$ | 0.2119 | 0.5418 | 2.0640 | $2.65\times$ |
| $K = 5$ | 0.2094 | 0.5371 | 2.0462 | $4.26\times$ |
| $K = 10$ | 0.2076 | 0.5337 | 2.0343 | $8.32\times$ |
| $K = 20$ | 0.2068 | 0.5320 | 2.0277 | $16.48\times$ |
| $K = 50$ | 0.2062 | 0.5308 | 2.0240 | $41.06\times$ |

## H.5 COMPLETE RESULTS

We provide the complete experimental results, including standard deviations, to demonstrate the statistical significance and variance of our findings.

Table 10: Performance comparison on PDE imputation tasks. Each sample represents a temporal sequence of 50 frames, each with $32 \times 32$ spatial resolution. Results show the MSE between the reconstructed and the ground truth solutions from the PDE solver, averaged over all timesteps.

| Method | | Shallow Water (feasibility loss, $\times 10^{-8}$) | | Advection (simulation MSE, $\times 10^{-1}$) | |
|---|---|---|---|---|---|
| | | 80% | 30% | 80% | 30% |
| Temporal Consistency | | 3.0248 | 4.2742 | 0.5097 | 0.6911 |
| Fast Marching | | 2.5931 | 8.8631 | 0.2127 | 0.5222 |
| Navier-Stokes | | 0.7045 | 2.8244 | 0.1350 | 0.4805 |
| MissDiff | | $0.2917 \pm 0.1683$ | $0.7527 \pm 0.1530$ | $\mathbf{0.1030 \pm 0.0004}$ | $0.1197 \pm 0.0096$ |
| AmbientDiff | | $0.1927 \pm 0.0050$ | $0.7504 \pm 0.1119$ | $0.1039 \pm 0.0009$ | $0.1219 \pm 0.0075$ |
| **Ours** | 1 step | $0.1878 \pm 0.0054$ | $\mathbf{0.7379 \pm 0.1101}$ | $0.1035 \pm 0.0008$ | $\mathbf{0.1189 \pm 0.0069}$ |
| | 200 steps | $\mathbf{0.1869 \pm 0.0015}$ | $0.7502 \pm 0.1120$ | $0.1037 \pm 0.0009$ | $0.1231 \pm 0.0109$ |

## H.6 VISUALIZATION OF GENERATED SAMPLES

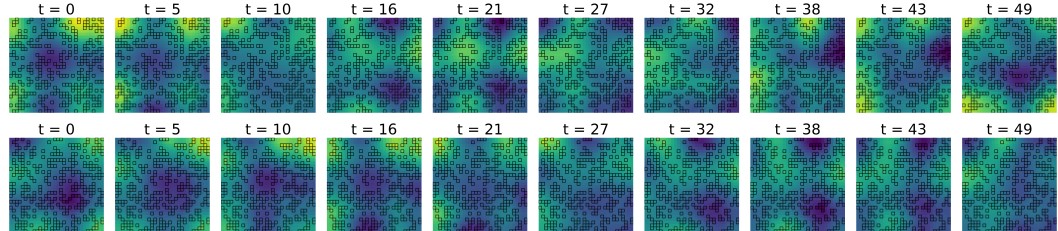

Figure 4: Imputed results on 2D Shallow Water dataset where 30% of the original data points are available for training.

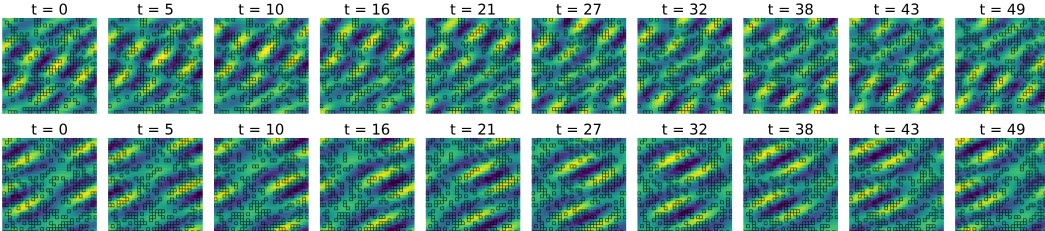

Figure 5: Imputed results on 2D Advection dataset where 30% of the original data points are available for training.

## I LIMITATIONS AND FUTURE WORK

Our work represents a first step towards systematically incorporating mask distribution priors into the training of generative models for incomplete data. A primary assumption in our current framework is that the mask distribution $p_{\mathrm{mask}}(\boldsymbol{M})$ is known *a priori* and is independent of the data $\boldsymbol{x}_0$. However, in certain real-world scenarios, the missingness mechanism can be data-dependent (e.g., a weather station failing due to the direct impact of a typhoon it is measuring) or follow complex patterns that are unknown. Our current methodology does not explicitly address these more complex cases. We believe that extending this framework to handle unknown or data-dependent mask distributions is a significant and important direction for future research.

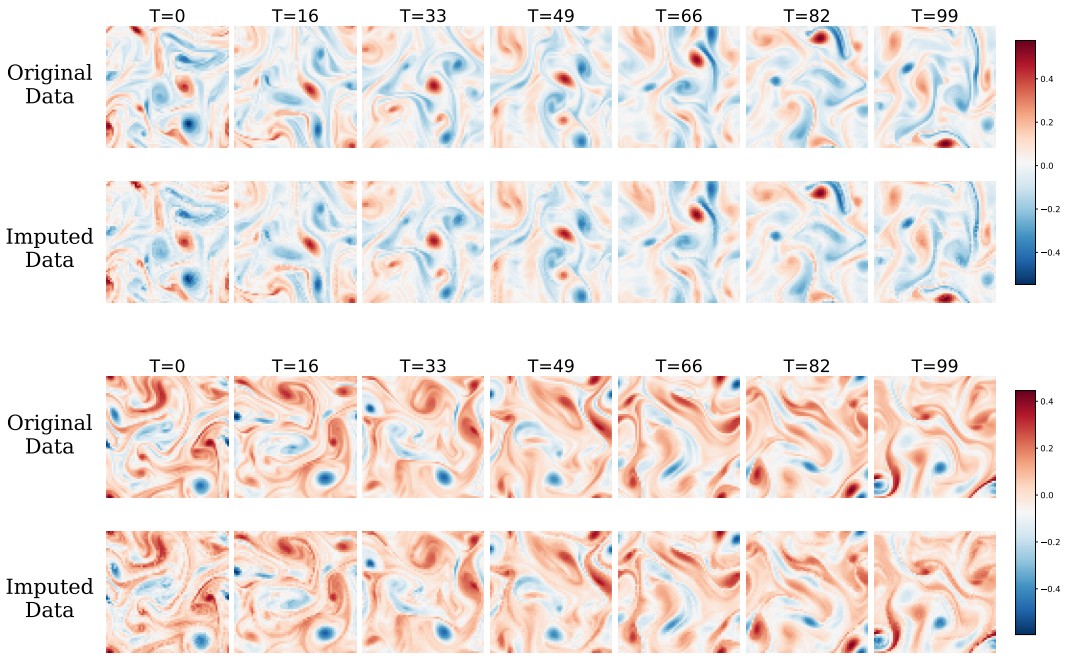

Figure 6: Sample imputation results on 2D Navier-Stokes dataset where 80% of the original data points are available for training.

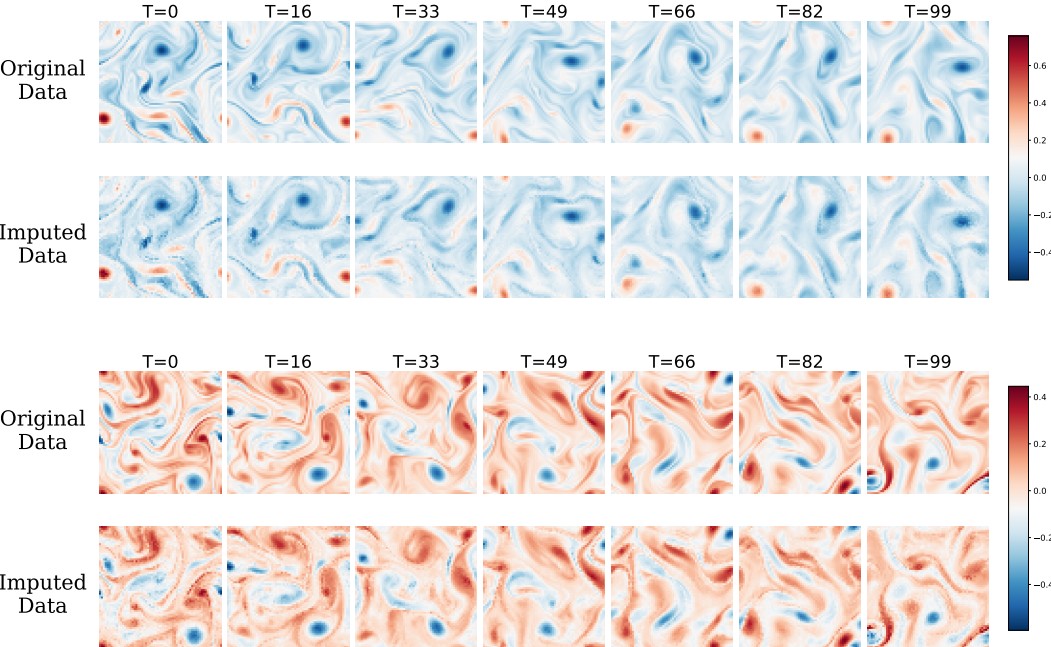

Figure 7: Sample imputation results on 2D Navier-Stokes dataset where 60% of the original data points are available for training.

On the theoretical front, our analysis provides guarantees for the asymptotic convergence of our training paradigm. We acknowledge that this analysis does not extend to a non-asymptotic regime. A more comprehensive theoretical investigation, such as deriving finite sample complexity bounds or formally quantifying the approximation error introduced by the neural network architecture, is considerably challenging. Such an analysis would need to account for the complex interplay between

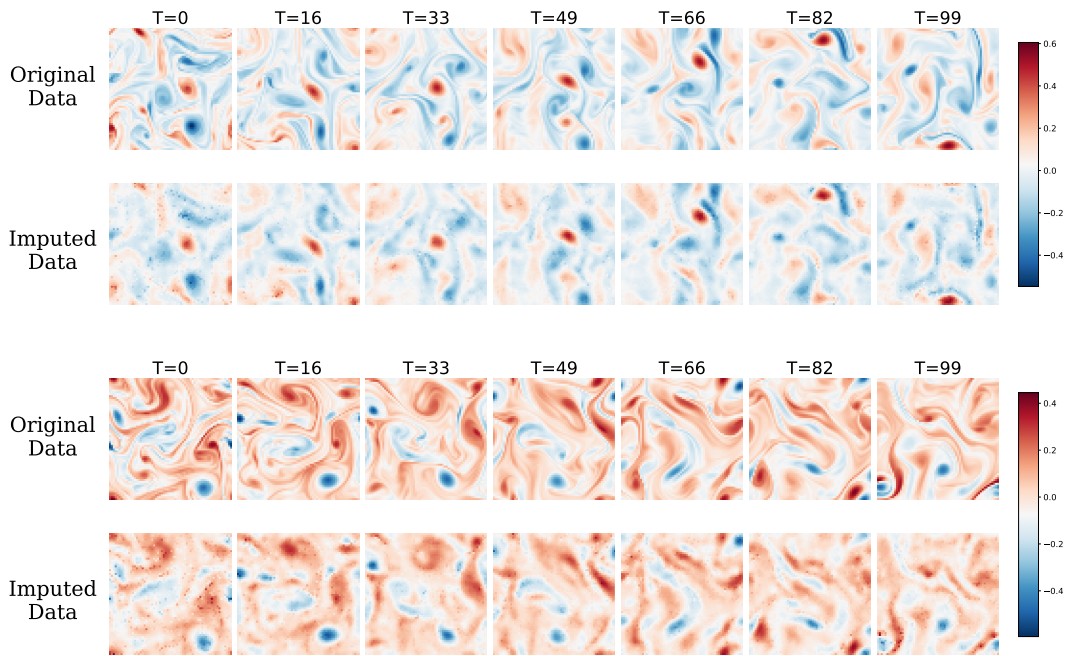

Figure 8: Sample imputation results on 2D Navier-Stokes dataset where 20% of the original data points are available for training.

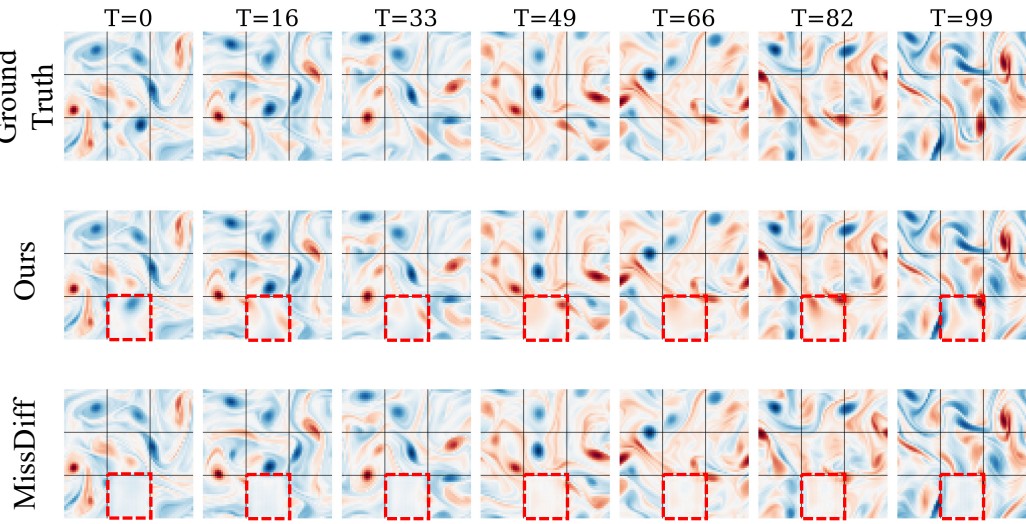

Figure 9: Comparison of original and imputed data from the Navier-Stokes dataset (one missing block). Each sample consists of 100 frames at $64 \times 64$ resolution.

the diffusion process, the context-query sampling strategy, and the function approximator. We leave this rigorous theoretical extension as an important open problem for future work.

## J  LLM USAGE STATEMENT

We used large language models (Claude) to assist with manuscript preparation in the following capacities: (1) improving the clarity and grammatical correctness of our writing through proofreading

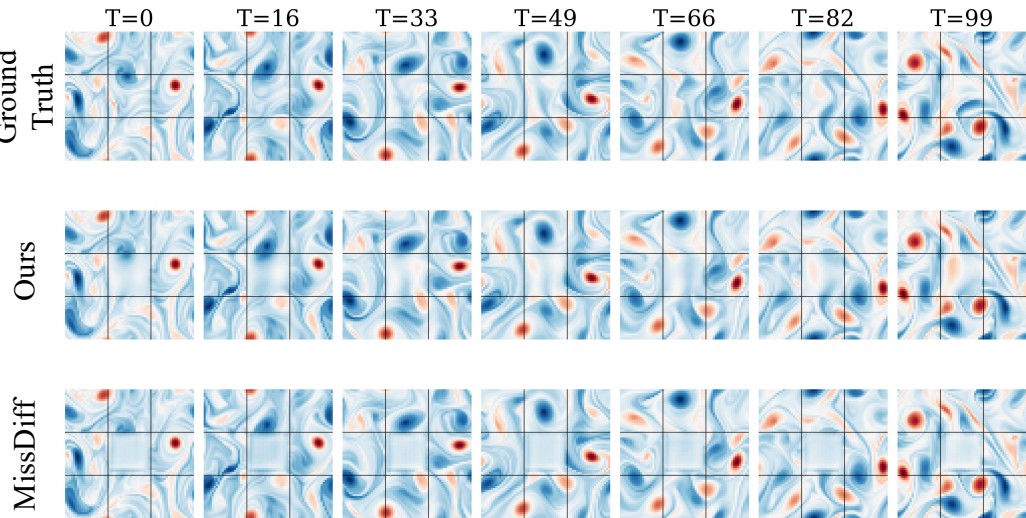

Figure 10: Imputation results for our method vs. MissDiff. The imputed block is the center one. The baseline method shows characteristic white central regions with no meaningful generation, confirming our theoretical prediction (Theorem. 1) of learning failure in these areas.

and copy-editing suggestions, (2) formatting LaTeX code for tables and equations, (3) reviewing mathematical proofs for logical consistency and clarity, and (4) identifying and correcting typographical errors throughout the manuscript.

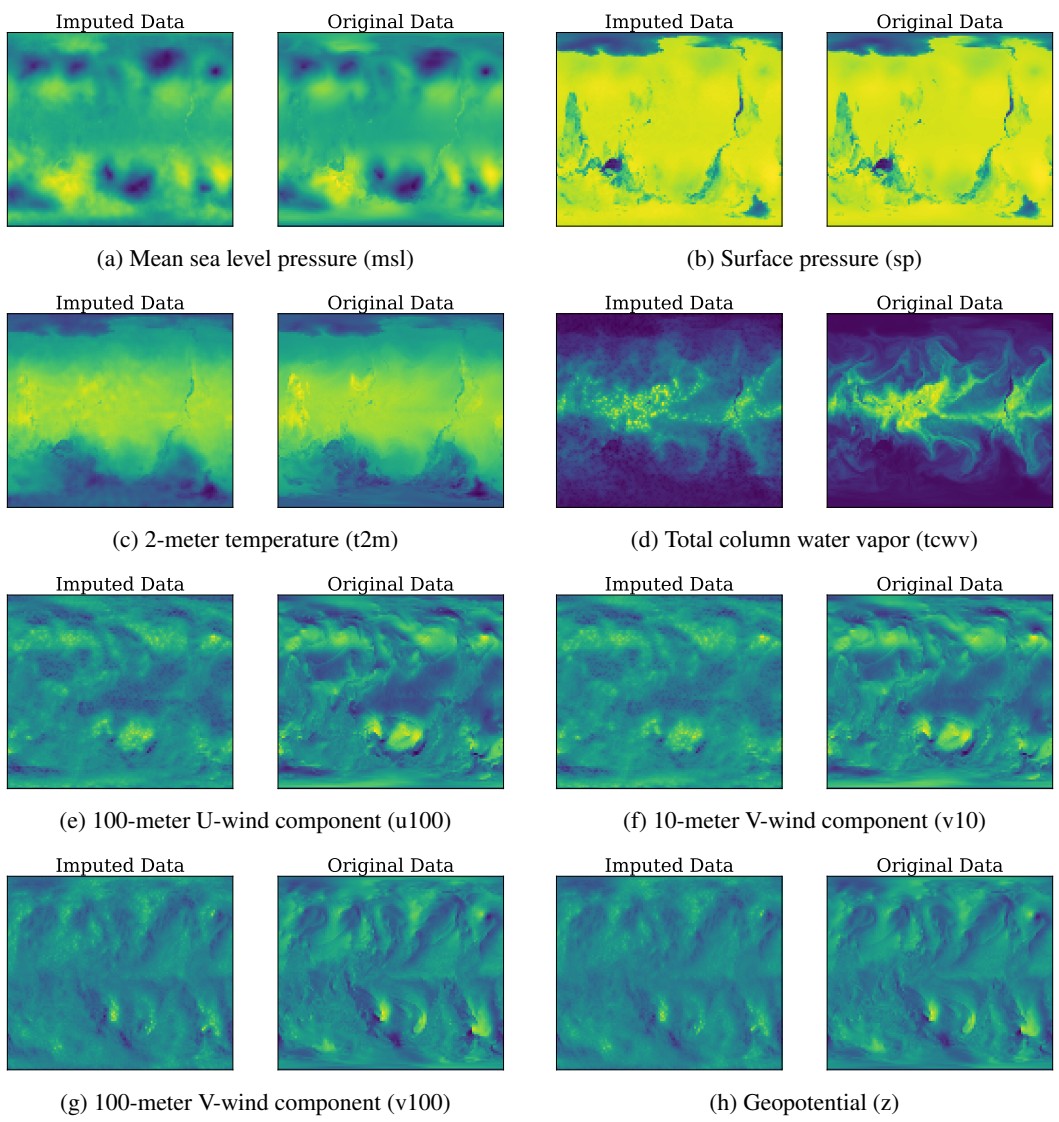

Figure 11: Imputation results on the ERA5 dataset with 20% observed points. Each subfigure shows a different atmospheric variable. The left column of each subfigure contains the imputed/reconstructed data, and the right column shows the original data.

