# OpenReview forum: "Incomplete Data, Complete Dynamics: A Diffusion Approach"
_ICLR.cc/2026/Conference — ICLR 2026 Poster_

### Official Review · Reviewer_bUgQ · 2025-10-28

**Soundness:** 4
**Presentation:** 3
**Contribution:** 3
**Rating:** 6
**Confidence:** 3

**Summary:**

This paper addressed the problem of imputing observational data under unknown dynamics. In diffusion-based imputation models, context mask and query mask are needed to train the model to extend the learned dynamics to impute unobserved data. The authors claim that the unobserved data is structured in reality and that diffusion and mask-based methods lack theoretical foundations, and then they introduce their novel pipeline to address the structured missing prior. The authors theoretically analyzed the correct strategy for context-query mask partitioning and extrapolation, and conducted comprehensive experiments on simulated and real-world datasets against competitive baseline methods. The results demonstrated the superiority of their method and the importance of catering to the structured missing prior in algorithm design. Their work could be a sound analysis platform for future diffusion-based and mask-based methods in data imputation.

**Strengths:**

S1: This paper gives a clear literature review and a contrasting comparison with similar models.
S2: The algorithm is backed by rigorous theorems and proofs, and additionally addresses the structured missing prior in reality.
S3: The experiments are thoughtful with ablation studies on the model design and components.

**Weaknesses:**

W1: The explanation of the mask selection strategy is confusing, and Figure 1 is hard to understand. In the series of training data with added noise, all data points seem to be observed once across time. No data point needs to be imputed.
W2: Sometimes, the mask distribution $p_{mask}$ depends on the data samples $x_0$. For example, a weather station may be damaged by a typhoon's direct impact. Also, cloud blocks weather satellite observing the sea temperature, and high sea temperature also promotes cloud generation.
W3: The authors proposed that the mask selection strategy for $M_{ctx},M_{qry}$ should align with the mask distribution prior $p_{mask}$ and used a block mask as an example. However, the authors didn't consider the case when the mask distribution pattern cannot tessellate on the whole space.
W4: The analysis of the paper's limitations is missing in the main text.
W5: The authors claim that some methods are not scalable, but they have not provided a scalability analysis and report on wall-time across baselines.

Minor remarks:
1. Line 723 VAE-based "and" GAN-based methods.
2. The definition of $x_\theta$ is unclear on line 138, and its input and output should be specified.
3. What is the definition of $i$ in eq. 2? Training instance or feature dimension?
4. A cross-reference to the proof of Thm 1 should be included in the paragraph starting from line 172.
5. On line 361, the improvement on MissDiff from noise matching to data matching needs to be quantified.
6. The margin of the caption of Figure 3 is too narrow.

**Questions:**

Q1: What is the term "feature" referring to in the main text? Does it refer to different data modalities or cell grid?
Q2: We observed a significant improvement from block-wise to pixel-level configuration on the Navier-Stokes dataset in Table 2. Is it because the authors tuned the hyperparameters on the NS dataset, as stated in tables 6-7? This could be a source of evaluation bias.
Q3: How is the physical dynamic learned? The data is not assumed to include the time dimension, and the "time" in the main text only refers to the diffusion time steps.

---

> ### Author Response · Authors · 2025-11-16
>
> # W1
>
> Thank you for your thoughtful feedback regarding the mask selection strategy and Figure 1. We appreciate the opportunity to clarify these points.
>
> **On the Mask Selection Strategy**:
>
> The core idea is simple: during training, we simulate different "views" of the same incomplete sample by randomly selecting subsets of observed points (context) and treating the rest as targets (query). This encourages the model to learn robust reconstructions from arbitrary observation patterns. The key is to ensure that every observed location has a chance to be queried during training. Otherwise, the model never learns to reconstruct from a limited context. This is not a hyperparameter tuning trick, but a principled way to expose the model to diverse partial observation settings.
>
> **On Figure 1**:
>
> We apologize for the confusion. The figure aims to contrast two sampling strategies:
> - **Top (Ineffective)**: Uniformly sampling context points can leave some observed regions never queried, so the model never learns to use them for reconstruction.
> - **Bottom (Effective)**: Sampling context in structured blocks (matching real-world sensor layouts) ensures all observed regions are queried over time, enabling the model to learn complete reconstruction.
>
> The “across time” refers to the accumulation of training steps—over many batches, the model sees all observed regions as queries.
>
>
> # W2, W3, W4
>
> Thank you for your thoughtful comments and for highlighting these important considerations.
>
> **Our contribution as foundational work**: We emphasize that our work is the first to provide a principled, theoretically grounded solution for learning from incomplete data in the foundational setting where the mask distribution is known and independent of the data. This setting, despite its importance, has lacked rigorous treatment in prior work. Establishing this foundation is a critical first step toward addressing more complex scenarios.
>
> **Extensions to more realistic cases**: The challenging cases you raised, where mask distribution depends on data (W2) or when mask patterns cannot tessellate the space (W3), are indeed important for real-world applications. Importantly, these can be addressed by building upon our framework through modular extensions:
> - For data-dependent masks, one could learn $p(M|x)$ as an independent module or jointly optimize both $x$ and $M$ in an end-to-end fashion
> - For non-tessellating patterns, adaptive masking strategies or learned mask distributions could be integrated into our context-query partitioning scheme
>
> Our work provides the theoretical foundation and principled methodology that makes these extensions feasible and well-motivated.
>
> **Addressing your suggestions**: We have added a Limitations section (Appendix I) that formally discusses these points and outlines them as important directions for future work. We have also corrected the mentioned typos.
>
> We appreciate you bringing these valuable perspectives to our attention, and we are committed to exploring these extensions in future work.
>
> # W5
> Please see the general response Evaluation on DiffPuter's Tabular Task. We also corrected minor typos. Thanks for mentioning.

---

> > ### Author Response · Authors · 2025-11-16
> >
> > # Q1
> >
> > The "feature" refers to "dimension". We have revised our paper to make this clear.
> >
> > # Q2
> >
> > We understand the concern about potential evaluation bias, but we can confirm that the observed performance improvement is **not a result of fine-tuning hyperparameters**. The significant performance gain stems directly from our core methodological contribution: **the strategic context-query partitioning strategy**. Here is a detailed explanation:
> > 1. **Source of Improvement**: The performance difference highlighted in Table 2 is almost entirely attributable to the change in the training strategy itself. The "pixel-level (incorrect)" and "block-wise (correct)" rows for our method were trained under identical settings and hyperparameters, with the only difference being the partitioning strategy.
> > 2. **The "Incorrect" Strategy Failure**: When data is missing in large, contiguous blocks (the "block-wise" setting), a naive "pixel-level" query strategy fails.
> >   - As our theoretical analysis in Theorem 1 predicts, if an entire block is missing, a random pixel-level query strategy will result in a zero-percent chance ($P(M_{qry}=1|M_{ctx}) = 0$) of those missing pixels ever being included in the query set.
> >   - Consequently, no gradients are ever backpropagated for these regions. The model fundamentally cannot learn to reconstruct the missing data.
> >   - This is the exact issue plaguing standard methods in this scenario and is visually confirmed in Figure 10, where the baseline method fails to generate any meaningful content in the missing block.
> > 3. **The "Correct" Strategy Success**: Our "block-wise (correct)" strategy is designed precisely to solve this.
> >   - By sampling entire blocks as context and query, we guarantee that all regions, including those originally unobserved, have a positive probability of being queried during training (as illustrated in Figure 1).
> >   - This allows the model to learn the true, complete data distribution, leading to the dramatic improvement in imputation quality seen in Table 2.
> > 4. **Purpose of Ablation Studies (Tables 6-7)**: The ablation studies on the NS dataset were not performed to find the single best hyperparameter combination for that dataset. Instead, their purpose was to demonstrate the robustness of our method. As shown in those tables, our method achieves state-of-the-art performance across a wide and reasonable range of parameters (e.g., context ratios from 50% to 90%), demonstrating that our approach is not overly sensitive to these choices.
> >
> > In summary, the performance jump is a direct and intended consequence of our theoretically-grounded partitioning method, which correctly provides learning signals for all parts of the data, rather than an artifact of hyperparameter tuning.
> >
> >
> > # Q3
> >
> > The reviewer is correct that "time" can refer to diffusion steps, but our data does include the physical time dimension, which is how the dynamics are learned. As detailed in Section 4.2 (Datasets), all datasets are based on PDE dynamics, meaning each sample is a spatio-temporal sequence (like a video). For example, the Navier-Stokes data has 100-frame sequences, and Shallow Water has 50 temporal frames. The "incompleteness" we address is spatial, not temporal. The observation mask is fixed across all temporal frames for a given sample. This simulates real-world fixed sensors.

---

> > > ### Comment · Reviewer_bUgQ · 2025-11-21
> > >
> > > My concerns have been adequately addressed and I have no further questions to ask. Good luck!

---

> ### Comment · Reviewer_bUgQ · 2025-11-21
>
> After seeing the responses from the authors, I would like to raise my rate to 8.

---

> > ### Author Response · Authors · 2025-11-21
> >
> > Thank you for your recognition and valuable feedback. We are pleased that you appreciate the core contribution of our method.

---

### Official Review · Reviewer_7yXe · 2025-10-31

**Soundness:** 2
**Presentation:** 3
**Contribution:** 2
**Rating:** 4
**Confidence:** 3

**Summary:**

This paper introduces a diffusion-based framework for learning physical dynamics when the training data itself is incomplete. The core contribution is a training strategy termed "context-query partitioning." This strategy partitions the observed data within each incomplete sample into a "context" for model input and a "query" for loss computation. Theoretically, the authors demonstrate that this approach asymptotically recovers the true data distribution without full supervision, provided the partitioning strategy ensures a non-zero probability for any dimension to be queried. For inference, the paper proposes an ensemble sampling method that reconstructs the complete sample by averaging predictions over multiple, randomly drawn context masks. Empirical results on several PDE and climate datasets demonstrate superior performance over existing baselines, especially in highly sparse observation regimes.

**Strengths:**

Problem Significance: The paper tackles a crucial and pervasive problem in scientific computing: learning from inherently sparse and incomplete observational data.
Theoretical Rigor: The paper provides a theoretical justification (Theorem 1) for its proposed training strategy. This principled approach is a clear advantage over prior works that often rely on heuristics.
Strong Empirical Results: The method demonstrates significant performance improvements across multiple benchmarks, particularly in sparse regimes with very low data coverage (e.g., 1%-20%).

**Weaknesses:**

- Insufficient Discussion of Related Work: The primary weakness is the failure to adequately discuss the connection to masked signal modeling, such as Masked Autoencoders (MAEs). The "context-query partitioning" is conceptually very similar to masking part of the visible data to predict another part. The absence of this discussion makes the core idea seem more novel than it is.
- Limited Baseline Comparison: While the paper includes recent diffusion-based baselines, the modification of some baselines (e.g., adapting MissDiff to a data matching framework) makes it difficult to assess if the comparison is entirely fair. Furthermore, excluding some theoretically rigorous but computationally expensive methods, while understandable, would be more convincing with a small-scale comparison or a deeper discussion.
- Scope of Theory: While Theorem 1 is a highlight, it focuses on asymptotic convergence. It doesn't analyze finite sample complexity or the approximation error introduced by the model architecture. The theorem's premise—that the union of query masks covers all dimensions—is satisfied by the design of the sampling strategy, which makes the argument slightly circular.

**Questions:**

1. Could the authors elaborate on the conceptual similarities and differences between the "context-query partitioning" strategy and self-supervised learning paradigms like Masked Autoencoders (MAEs)? This is crucial for accurately positioning the paper's novelty.
2. In the baseline comparisons, were the hyperparameters for all baseline methods tuned with the same rigor as the proposed method? Ensuring a robust comparison is key to validating the claimed performance gains.
3. The "distribution-preserving" sampling strategy mentioned in line 264 appears critical to success. In real-world scenarios where prior knowledge of P_mask(M) might be inaccurate, how sensitive is the method's performance to this "mask distribution mismatch"?

---

> ### Author Response · Authors · 2025-11-16
>
> # Connection with MAE
>
> Thank you for raising this important clarification. While the masking operations appear superficially similar, our approach differs fundamentally from MAE in purpose, data assumptions, and theoretical goals:
>
> ## Key Differences
> 1. **Data Completeness Assumption (Most Critical)**
> - **MAE**: Assumes access to complete data during training, artificially masks parts for self-supervised representation learning
> - **Our work**: Training data is inherently incomplete. No complete ground truth exists. This reflects real-world scientific measurements where complete observations are physically impossible (e.g., sparse sensor networks, satellite occlusions)
> 2. **Objective**
> - **MAE**: Learn robust representations via pretext tasks for downstream applications
> - **Our work**: Learn the complete generative distribution and enable accurate imputation from incomplete observations
> 3. **Masking Strategy Design**
> - **MAE**: Random masking to encourage learning general features
> - **Our work**: Strategic context-query partitioning guided by Theorem 1 to ensure **positive gradient updates for originally unobserved dimensions**. Without this design, the model learns arbitrary values for these dimensions (see Eq. 2).
> 4. **Theoretical Contribution**
> - **MAE**: Empirical effectiveness in representation learning
> - **Our work**: First theoretical proof (Theorem 1-2) that diffusion models trained on incomplete data can asymptotically recover the complete distribution under structured masking
>
> ## Why This Distinction Matters
>
> In MAE, masking is a **training trick** applied to complete data. In our setting, incompleteness is the **fundamental constraint** of the problem. Our context-query partitioning solves a theoretical challenge: **how to ensure the model learns meaningful predictions for dimensions that are permanently missing in training data**, which MAE never encounters.
>
> # Hyperparameter Tuning Rigor
>
> We ensured fair hyperparameter tuning across all methods:
>
> - **Unified framework**: All diffusion-based methods (MissDiff, AmbientDiff, and ours) were implemented within the same codebase and trained with identical experimental protocols (same training epochs, learning rates, batch sizes, etc.)
>
> - **Best performance reported**: For all baselines, we tested both single-step and multi-step sampling strategies and reported their best results, ensuring we did not disadvantage any baseline method
>
> - **Robustness to hyperparameters**: Our experiments show that the Diffusion UNet 3D transformer is relatively insensitive to hyperparameter choices, which helps ensure stable comparisons
>
> **Architecture-agnostic improvements**: To further validate that our gains come from the training methodology rather than architecture choices, we conducted experiments with two different backbones (Appendix G.2): Karras UNet (our main architecture) and Fourier Neural Operator (FNO)
>
> The results demonstrate that **our training approach consistently outperforms baselines regardless of backbone architecture**, confirming that the performance gains stem from our principled context-query partitioning strategy rather than favorable hyperparameter tuning.

---

> > ### Author Response · Authors · 2025-11-16
> >
> > # Fairness of MissDiff Modification
> >
> > We appreciate your concern about the fairness of modifying MissDiff to use data matching. **This modification was necessary and actually benefits MissDiff rather than disadvantages it.**
> >
> > **Why the original noise matching formulation fails in our setting**:
> >
> > We initially attempted to evaluate MissDiff with its original noise matching objective. However, we found that **the training loss failed to decrease at all** in our experimental setting. This failure stems from a fundamental difference between tabular and spatiotemporal data:
> >
> > - **MissDiff's original paper focused on tabular data** with relatively moderate missing ratios. In tabular data, each entry represents an independent feature, where missing one entry means completely losing information about that specific feature.
> >
> > - **Our PDE datasets involve spatiotemporal fields** (images/videos) with much higher missing data ratios (down to 1% observed). Critically, missing one pixel does not mean losing one independent feature. Instead, due to the inherent spatial smoothness and continuity of physical fields, neighboring pixels carry correlated information. A single missing pixel can often be reasonably inferred from its spatial context.
> >
> > However, this difference has profound implications for noise matching objectives:
> >
> > - While data matching can leverage spatial smoothness to predict reasonable values even from sparse observations
> > - **Noise matching requires predicting fine-grained noise patterns**, which demands much denser observations to capture local structure
> > - At extreme sparsity (e.g., 1% observed), the noise prediction task becomes ill-posed. There is simply insufficient local context to distinguish signal from noise
> >
> > This explains why MissDiff's noise matching completely fails (outputs all zeros) in our setting, while data matching can at least produce meaningful predictions by exploiting the smoothness priors inherent in physical systems. This modification **actually helps MissDiff perform better** in our challenging setting. Without this adaptation, MissDiff would completely fail in our experimental scenarios, making the comparison meaningless.
> >
> > # Scope of Theory
> >
> > Thank you for your comment. We agree that finite-sample analysis and architectural approximation error are important aspects not covered by Theorem 1. Deriving such non-asymptotic guarantees is highly non-trivial for diffusion models with structured missingness and remains an open challenge [1]. Indeed, even in existing convergence and error analyses of diffusion models, the learning error is often directly assumed rather than established, as seen in works like [2]. Our theoretical contribution focuses on establishing asymptotic consistency as a foundational step.
> >
> > [1]Gao X, Nguyen H M, Zhu L. Wasserstein convergence guarantees for a general class of score-based generative models[J]. Journal of machine learning research, 2025, 26(43): 1-54.
> >
> > [2]Benton J, De Bortoli V, Doucet A, et al. Nearly $ d $-linear convergence bounds for diffusion models via stochastic localization[J]. ICLR24
> >
> > # Sensitivity to mask distribution mismatch
> >
> > At this stage, we cannot fully characterize the sensitivity under inaccurate prior knowledge of $p_{mask}(M)$, as this would require a systematic sweep across varying degrees of distribution shift, which remains future work.
> > However, our ablation study in Table 2 provides a clear boundary case: when the mask distribution is completely mismatched (e.g., using pixel-level partitioning for block-wise observations), performance degrades substantially. The significant performance gap between the pixel-level (incorrect) and block-wise (correct) rows empirically validates that our method is highly effective when the context-query partitioning strategy aligns with the true observation pattern, underscoring the importance of our distribution-preserving design principle.

---

> > > ### Author Response · Authors · 2025-11-26
> > >
> > > Dear Reviewer 7yXe,
> > >
> > > We hope our responses have addressed your concerns. We would greatly appreciate it if you could take a moment to share your thoughts when you have time.
> > >
> > > Thank you for your consideration and efforts.

---

### Official Review · Reviewer_VhJM · 2025-10-31

**Soundness:** 2
**Presentation:** 2
**Contribution:** 2
**Rating:** 4
**Confidence:** 4

**Summary:**

This paper proposes a diffusion-based framework for learning physical dynamics from incomplete and irregularly sampled data, a common challenge in scientific domains. The core contribution is a novel training paradigm where each incomplete data sample is partitioned into a "context" set and a "query" set. A conditional diffusion model is then trained to reconstruct the query portions given the context, without requiring access to complete ground-truth data. The authors provide a theoretical analysis demonstrating that their method asymptotically converges to the true data distribution under certain conditions. Furthermore, they introduce an ensemble sampling technique for inference to reconstruct the complete data. The method is evaluated on several synthetic PDE datasets (Navier-Stokes, Shallow Water, Advection) and the real-world ERA5 climate dataset, showing superior performance over existing baselines, especially in highly sparse settings.

**Strengths:**

The paper tackles the critical and practical challenge of learning physical dynamics from inherently sparse and incomplete real-world data.

It provides a theoretical analysis (Theorem 1 and 2) that offers guarantees for the learning process, explaining how and why the model can learn from incomplete data. This is a notable strength compared to more heuristic approaches.

**Weaknesses:**

Limited Novelty: The core idea of context-query partitioning is conceptually similar to existing masked data training strategies in self-supervised learning and other generative models. The distinction from methods like Ambient Diffusion, which also operate on incomplete data, is not sufficiently pronounced. The contribution seems to be more about a specific, effective sampling strategy rather than a fundamentally new paradigm.

Lack of Clarity on Implementation: The "strategic context-query partitioning" is the cornerstone of the method, yet its implementation is vaguely described. The paper lacks a clear algorithm or pseudo-code explaining how M_ctx is sampled from a given observation mask M for different structural patterns (e.g., block-wise). This ambiguity hinders reproducibility.

Incomplete Baseline Comparison: Excluding recent methods like those by Chen et al. (2024b), Givens et al. (2025), and Zhang et al. (2025a) due to computational cost weakens the experimental validation. A more thorough comparison, even on a smaller scale, would be necessary to firmly establish state-of-the-art performance.

Unaddressed Inference Cost: The proposed ensemble sampling method for reconstruction requires running the model K times per sample. This introduces a significant computational overhead at inference time, which is a critical factor for many scientific applications. The paper does not analyze this trade-off or discuss how K is chosen.

**Questions:**

Could the authors please elaborate on the key conceptual difference between their method and Ambient Diffusion (Daras et al., 2023)? Both seem to train diffusion models by applying masks to incomplete data. What is the crucial insight that leads to the performance gains shown?

Could you provide a more concrete algorithm or pseudo-code for the "strategic context-query partitioning" procedure, especially for the block-wise missing data scenario? For a given observation mask M with 5/9 observed blocks, how exactly is M_ctx (e.g., with 4 blocks) sampled?

Regarding the excluded baselines: Could the authors provide a more detailed justification for their exclusion? Would it be possible to conduct a qualitative or small-scale quantitative comparison to give the reader a better sense of where the proposed method stands?

What is the inference-time cost of the ensemble sampling approach? How does the performance (e.g., MSE) and computational cost vary with the number of ensemble members K? What was the value of K used in the experiments?

---

> ### Author Response · Authors · 2025-11-16
>
> # Clarification on Limited Novelty
>
> We respectfully disagree with this characterization. Our contribution is fundamentally different from both self-supervised learning and Ambient Diffusion in problem setting, theoretical depth, and methodological rigor. Reviewer ztBx recognized this, noting that our "context-query partitioning strategy constitutes a genuine advancement" and our theoretical analysis demonstrates "notable rigor...which distinguishes the work from purely empirical approaches.
>
> ## Distinction from masked self-supervised learning
>
> The problem settings and objectives are fundamentally different:
>
> - **Self-supervised learning**: Trains on **complete data** that is artificially masked. The goal is to learn representations by predicting masked portions, and the complete ground truth is always available during training.
>
> - **Our setting**: Trains on **inherently incomplete observations** (e.g., sparse sensors, cloud occlusion). The goal is to reconstruct complete physical fields from permanently missing data. **No complete ground truth exists** during training.
>
> **Core challenge**: How can a model learn to reconstruct regions that were **never observed in any training sample**? This differs fundamentally from self-supervised learning, which reconstructs artificially masked complete data.
>
>
> ## Critical distinction from Ambient Diffusion
>
> **The crucial insight**: While both methods apply masks during training, the key difference lies in **how masks are sampled**. Ambient Diffusion uses uniform random sampling regardless of the observation structure, while we design context-query partitioning strategies that **align with the mask distribution**. This seemingly subtle distinction has profound theoretical and practical implications:
>
> 1. **Theoretical rigor vs. heuristics**:
> - Ambient Diffusion: Proposes "further corruption" heuristically; Theorem 4.1 in their paper only shows the optimal solution form but provides no guidance on when or why the method works.
> - **Our contribution**: Theorem 1 proves that dimension i learns meaningful expectations **only when** $P(M_{qry} = 1 | M_{ctx}) > 0$, directly answering which dimensions receive gradient updates and providing actionable design principles (Principle 1).
>
> 2. **Structure-aware training strategies**:
> - Ambient Diffusion: Applies the **same further corruption strategy** regardless of mask distribution structure (randomly drops pixels for both pixel-level and block-structured observations).
> - **Our contribution**: Develops context-query partitioning strategies **tailored to mask distributions**. Table 2 demonstrates this is not merely an implementation detail. Using misaligned strategies fails catastrophically (Navier-Stokes: error 1.49 vs 0.76). Figure 1 illustrates why this matters: with misaligned strategies, entire regions receive **zero query probability**, meaning those dimensions never get gradient updates and the model learns nothing about them. This directly explains the performance gap. It's not about better optimization, but about whether the model can learn at all in certain regions.
>   The performance gains stem from ensuring $P(M_{qry} = 1 | M_{ctx}) > 0$ for all dimensions through distribution-aligned partitioning. Without this, models fail to learn in originally unobserved regions, not due to optimization issues, but due to the complete absence of a supervision signal (Theorem 1, Eq. 5).
>
> 3. **Training-inference gap analysis**:
> - Ambient Diffusion: Uses fixed mask sampling without theoretical justification for the approximation $E[x_{0} | x_{t} ] ≈ E[ x_{0} | Ã x_{t}, Ã]$.
> - **Our contribution**: Theorem 2 provides explicit convergence analysis and bias-variance decomposition (Eq. 13) showing how ensemble averaging bridges this gap.
>
> **Summary**: Our work is not "a specific sampling strategy" but a **theoretically principled framework** addressing a distinct problem (learning from permanently incomplete data) with rigorous analysis of what enables learning in never-observed dimensions. Such insights are absent in both self-supervised learning and Ambient Diffusion.

---

> > ### Author Response · Authors · 2025-11-16
> >
> > # Context-Query Partitioning Implementation
> >
> > We added context and query mask selection algorithm implementation in Appendix H2.
> >
> > **Core Principle**: Given observation mask $M$, we sample context mask $M_{ctx}$ and query mask $M_{qry}$ by applying the same structural pattern as the original mask distribution $p_{mask}(M)$.
> >
> > ### Concrete Algorithm for Block-wise Scenario
> >
> > For the block-wise example mentioned (5/9 observed blocks in $M$):
> >
> > 1. **Identify observed blocks**: Extract which blocks are observed in $M$ (e.g., blocks {1,2,4,7,8})
> > 2. **Sample context blocks**: From these 5 observed blocks, randomly select 4 blocks for $M_{ctx}$
> > 3. **Sample query blocks**: From the 5 observed blocks, randomly select 4 blocks for $M_{qry}$
> > 4. **Construct masks**: $M_{ctx}$ and $M_{qry}$ contain only their respective sampled blocks
> >
> > Pseudo-code:
> > ```
> > Given: observation mask M with observed_blocks = {blocks where M==1}
> >
> > # Sample context mask
> > n_ctx = 4  # number of context blocks
> > sampled_ctx_blocks = randomly_select(observed_blocks, n_ctx)
> > M_ctx = create_mask(sampled_ctx_blocks)
> >
> > # Sample query mask
> > n_qry = 4  # number of query blocks
> > sampled_qry_blocks = randomly_select(observed_blocks, n_qry)
> > M_qry = create_mask(sampled_qry_blocks)
> >
> > Output: M_ctx, M_qry (both subsets of M)
> > ```
> >
> > ### Why This Works
> >
> > The rationale is that if $M$ is generated by randomly selecting 5/9 blocks, then $M_{ctx}$ should be generated by the same process applied to the available observations (selecting 4 blocks from those observed). This ensures that for any dimension not in $M_{ctx}$, there exist multiple possible observation masks $M$ that could contain it, guaranteeing positive query probability (Principle 1, Equation 6-7).
> >
> > ### General Case
> >
> > For any mask distribution:
> > - **i.i.d. pixel-level**: Randomly sample a subset of observed pixels in $M$ for $M_{ctx}$; independently sample another subset for $M_{qry}$
> > - **Block-structured**: Randomly sample a subset of observed blocks in $M$ for $M_{ctx}$; independently sample another subset for $M_{qry}$
> > - **Other patterns**: Apply the same structural sampling rule as $p_{mask}(M)$ to generate both $M_{ctx}$ and $M_{qry}$
> >
> > # Baseline Comparison
> >
> > We have now conducted additional experiments and provide a detailed justification below.
> >
> > ## Additional Experimental Comparison
> >
> > Following the reviewer's suggestion, we have conducted experiments under the DiffPuter (Zhang et al., 2025a) experimental setting. Please refer to the general response section for details.
> >
> >
> >
> > ## Detailed Justification for Original Exclusion
> >
> > **Computational feasibility**. The fundamental reason for excluding DiffPuter and related methods (Chen et al., 2024b; Givens et al., 2025) is the prohibitive computational cost when scaled to our high-dimensional PDE applications. Specifically:
> >
> > - **DiffPuter's iterative training paradigm** requires multiple cycles of: train model → generate data → retrain model with augmented data, typically around 10 iterations. This design works well for tabular generation, where each training round takes minutes.
> >
> > - **Our PDE tasks** require 10-20 hours per training round due to high-dimensional spatiotemporal data (e.g., 100×64×64 for Navier-Stokes). With multi-step diffusion sampling at inference, a single iteration of DiffPuter's approach would take days to weeks, making 10 iterations computationally infeasible.
> >
> > **Task mismatch**. Our method is designed for physical dynamics reconstruction where the information gap (Eq. 11 in our paper) between partial and complete observations is relatively small due to strong physical constraints. In contrast, DiffPuter targets tabular generation where this information gap is typically much larger, requiring their iterative refinement approach.
> >
> >
> > # Unaddressed Inference Cost
> >
> > Please refer to the general response section: Selection of K and Computational Trade-offs.

---

> > > ### Author Response · Authors · 2025-11-26
> > >
> > > Dear Reviewer VhJM,
> > >
> > > We hope our responses have addressed your concerns. We would greatly appreciate it if you could take a moment to share your thoughts when you have time.
> > >
> > > Thank you for your consideration and efforts.

---

### Official Review · Reviewer_ztBx · 2025-10-31

**Soundness:** 3
**Presentation:** 3
**Contribution:** 4
**Rating:** 8
**Confidence:** 3

**Summary:**

This paper tackles a genuinely important problem: learning physical dynamics when training data is inherently incomplete. The authors propose training diffusion models by strategically splitting each incomplete sample into "context" (what the model sees) and "query" (what it tries to predict), then using ensemble averaging at inference time to reconstruct complete fields.
The key idea is that if you carefully design how you partition incomplete observations during training, matching the partition strategy to the underlying observation pattern (e.g., block-wise vs. pixel-wise missing data), one can train a model that learns meaningful conditional expectations even for dimensions that were never observed in the training data. The authors provide a theoretical analysis that proves convergence to the true complete data distribution and validate it on both synthetic PDEs (Shallow Water, Advection, and Navier-Stokes) and real climate data (ERA5).

**Strengths:**

- The incomplete data setting represents a fundamental constraint in scientific applications where complete ground truth observations are physically impossible to obtain (e.g., global weather systems, ocean dynamics). The motivation is well-articulated and substantiated.

- The context-query partitioning strategy constitutes a genuine advancement over existing approaches. The key differentiation with existing lies in training-time strategic partitioning that adapts to observation patterns, rather than inference-time conditioning or generic loss masking.

- Theorem 1, concerning gradient scaling and parameter update frequency, provides meaningful insight into the significance of partitioning strategy selection. Theorem 2's ensemble convergence analysis demonstrates notable rigor, employing Martingale convergence theory rather than heuristic arguments. This theoretical grounding distinguishes the work from purely empirical approaches.

- The performance improvements documented in Table 2 for block-wise observations are substantial. The 1% observation regime results on ERA5 are particularly noteworthy, representing an extremely challenging setting.

**Weaknesses:**

A bit of nitpicking:
- The manuscript demonstrates conditions under which the method succeeds, but provides insufficient analysis of failure modes. Specifically, when does the information gap exceed the capacity of ensemble averaging to compensate? The cross-distribution experiments (Appendix G.3, Table 8) suggest performance degradation when the training and test observation ratios differ substantially; however, this warrants a more thorough discussion. Are there principled criteria for determining whether a test distribution lies outside the method's applicability range?

- Ensemble averaging during inference requires K forward passes through the model (the paper employs single-step sampling with K context masks). The selection of K and the resulting computational trade-offs are not discussed. For a method targeting scientific applications, these practical considerations are important. The multi-step sampling described in Appendix E further compounds this concern. Quantitative comparison of computational costs relative to baselines would be valuable.

**Questions:**

- How is K selected for ensemble averaging? Is there a principled approach to determining the number of context masks, or is selection based on empirical validation? Does the required K scale with observation sparsity?

- What is the computational overhead of the proposed method? Can you provide wall-clock training and inference times relative to baseline methods? This information is essential for practical implementation.

---

> ### Author Response · Authors · 2025-11-16
>
> Thank you for these thoughtful questions about failure modes, computational costs, and practical implementation considerations. We address your concerns below:
>
> # Failure Modes and Information Gap Analysis
>
> ## When does the information gap become too large?
>
> The information gap (Eq. 11) quantifies the difference between posterior variance given context versus full observations. Our method is particularly suited for physical dynamics because, unlike natural image generation, PDE solutions are well-constrained by observations due to underlying physical laws. When sufficient observations are available, the solution space is typically concentrated, making the posterior variance relatively small.
>
> The method would fail when:
> - The posterior distribution remains highly multi-modal or dispersed despite available observations
> - Observations are too sparse to meaningfully constrain the solution
> - The observation density is insufficient to capture characteristic scales of physical phenomena (e.g., correlation lengths in turbulent flows)
>
> While a universal criterion is challenging (as it depends on specific physics, observation patterns, and noise characteristics), a practical heuristic is: if observations cannot capture the dominant spatial/temporal scales of the system, the information gap will likely be too large. This is evident in our experiments, where 1% observation (Table 1) shows significantly larger errors than 20%, indicating we're approaching the method's applicability limits.
>
> # Cross-Distribution Generalization
>
> Regarding the performance degradation in Tab. 8 (original version, Tab. 9 in the newest version):
>
> The degradation stems primarily from training difficulty on sparser datasets, not from distribution shift. Learning from 20% observations is fundamentally harder than from 80%, regardless of test distribution.
>
> Our adaptive context sampling strategy explicitly addresses distribution mismatch:
> - At test time with observation ratio r_test, we sample context as a fraction of available observations
> - This ensures the model receives the same effective input ratio it saw during training
> - The strategy maintains consistent model behavior across different observation densities
>
> Results show models trained on denser observations (80%) generalize reasonably well to sparser tests (60%), while models trained on sparse data naturally struggle due to limited training information. This is expected and not a fundamental limitation of the method.
>
> **Principled criteria for applicability**:
>
> A test distribution lies within the method's applicability range when:
>
> - The observation pattern structure matches the training distribution
> - The test observation density is not substantially lower than the training density
> - The adaptive context sampling can maintain similar context-query ratios as during training
>
> # Selection of K and Computational Trade-offs
>
> Please refer to the general response section: Selection of K and Computational Trade-offs.

---

> > ### Author Response · Authors · 2025-11-26
> >
> > Dear Reviewer ztBx,
> >
> > We hope our responses have addressed your concerns. We would greatly appreciate it if you could take a moment to share your thoughts when you have time.
> >
> > Thank you for your consideration and efforts.

---

### Author Response · Authors · 2025-11-16
**General Response**

In response to the reviewers' feedback, we have updated our manuscript with the following revisions, mostly highlighted in color of red:

1. Corrected minor typos highlighted by the reviewers.
2. Provided a more detailed explanation for excluding certain baselines, including an expanded discussion on their prohibitively high computational costs, which render them unsuitable for our PDE dynamics generation task (see Appendix G).
3. Clarified the rationale for modifying MissDiff’s experimental setting, emphasizing that the adjustment was necessary to ensure its proper functionality (see Appendix H3).
4. Added new comparative experiments to investigate the impact of the ensemble size K on the results (see Appendix H4).
5. Included a discussion on the distinctions between our approach and related works in masked self-supervised learning (see Appendix C5).
6. Added context and query mask selection algorithm implementation (see Appendix H2)
7. Added Limitations and Future Work section (see Appendix I)



# Selection of K and Computational Trade-offs

## How is K selected?

As shown in Theorem 2 (Eq. 13), increasing K reduces variance terms in the prediction error. The optimal K balances accuracy and computational cost, which is problem-dependent due to varying magnitudes of data variance and model variance across different observation patterns and datasets.

Our experiments show that K=5-10 typically suffices across most scenarios (we set the default K=10). The required K does scale with observation sparsity, and sparser observations generally benefit from a larger K to reduce variance.


| K  | NS Error (80%) | NS Error (60%) | NS Error (20%) | Time Ratio |
|----|----------------|----------------|----------------|------------|
| 1  | 0.2239         | 0.5652         | 2.1446         | 1.00×      |
| 2  | 0.2147         | 0.5475         | 2.0822         | 1.81×      |
| 3  | 0.2119         | 0.5418         | 2.0640         | 2.65×      |
| 5  | 0.2094         | 0.5371         | 2.0462         | 4.26×      |
| 10 | 0.2076         | 0.5337         | 2.0343         | 8.32×      |
| 20 | 0.2068         | 0.5320         | 2.0277         | 16.48×     |
| 50 | 0.2062         | 0.5308         | 2.0240         | 41.06×     |

## Computational overhead analysis

**Training cost**: Our training procedure has a comparable computational cost to baseline diffusion methods (MissDiff, AmbientDiff) since the network architecture and input/output dimensions are the same. The main difference is in our context-query partitioning strategy during training, which adds negligible overhead.

**Inference cost**: The additional computational cost comes from sampling:

For **single sample generation** (common in scientific applications):
- The K forward passes can be executed in parallel since they are independent
- Wall-clock time increases sub-linearly with K rather than K-fold due to parallel GPU computations
- On an A800 GPU: K=10 requires 3.36× time of K=1 for 50-frame 32×32 sequences, and 8.32× for 100-frame 64×64 sequences
- The overhead depends on hardware parallelization efficiency and batch size

For **batch generation** of multiple samples:
- Computational cost scales approximately K times compared to baselines
- This represents a fundamental trade-off: our method enables learning from realistically incomplete data, a necessity in many scientific domains where complete measurements are physically impossible

---

> ### Author Response · Authors · 2025-11-16
>
> # Evaluation on DiffPuter's Tabular Task
>
> We conducted experiments on tabular data generation following DiffPuter's setup. Using their officially released code with default parameters, we ran experiments under the "missing at random 30%" setting (corresponding to Table 7 in the original paper). However, we were unable to reproduce the results reported in their paper. This issue has also been raised by other users in their GitHub repository. In the table below, we present the performance of their method as reproduced in our experiments, alongside the performance of our proposed method.
>
> It is worth noting that DiffPuter requires multiple rounds of model training (i.e., training several distinct models) to iteratively refine the imputed data. Moreover, in each round, they also require **multiple sampling steps and rounds**  to average the results. Therefore, the theoretical cost per iteration of their method is comparable to that of our model. However, our proposed method requires only a single iteration to achieve overall better performance than their method after 10 iterations.
>
> While our proposed method maintains a training cost comparable to that of standard diffusion models, other approaches, such as DiffPuter and several excluded baselines, demand tens of times more computational resources or become infeasible due to excessive CUDA memory requirements. We also provide a more detailed justification for excluding certain baselines, including an expanded discussion of their prohibitively high computational costs, which make them unsuitable for our PDE dynamics generation task (see Appendix G).
>
> | Method          | Adult   | Bean    | Default | Gesture | Letter | Magic  | News   | Shoppers |
> |-----------------|---------|---------|---------|---------|--------|--------|--------|----------|
> | Ours            | **5.3340**  | **0.3804**  | **0.9630**  | **0.7103**  | 0.5851 | **0.6314** | **0.6820** | **0.8658**   |
> | DiffPuter iter 1| 5.4949  | 0.7545  | 1.6294  | 1.0041  | 0.7733 | 0.8357 | 0.8610 | 0.9421   |
> | DiffPuter iter 2| 5.4932  | 0.6176  | 1.5436  | 0.9787  | 0.6944 | 0.7630 | 0.8004 | 0.9043   |
> | DiffPuter iter 3| 5.4925  | 0.5562  | 1.5016  | 0.9659  | 0.6599 | 0.7348 | 9.7733 | 0.8901   |
> | DiffPuter iter 5| 5.4967  | 0.5116  | 1.4630  | 0.9378  | 0.6278 | 0.7185 | 0.9239 | 0.8841   |
> | DiffPuter iter 10| 5.4940  | 0.4636  | 1.4296  | 0.8803  | **0.5739** | 0.6917 | 0.7421 | 0.8750   |

---

> > ### Author Response · Authors · 2025-12-01
> > **Author Summary for Meta-Review**
> >
> > Dear AC,
> >
> > We would like to highlight that our rebuttal has comprehensively addressed every concern raised by all four reviewers:
> > 1. **Novelty Clarification (Reviewers VhJM, 7yXe)**: We respectfully note that there appears to be a misunderstanding regarding our core contribution. While our context-query partitioning may superficially resemble masked pretraining, **the problem settings are fundamentally different**: MAE masks complete data for representation learning, whereas our method addresses the challenge of learning from **inherently incomplete** observations where ground truth never exists during training. Our contribution is not about leveraging masked pretraining for representation learning. Rather, it establishes a rigorous mathematical framework for a distinct problem. Theorem 1 proves precisely which dimensions can be learned and why, while Theorem 2 provides asymptotic convergence guarantees for the sampling process. This theoretical rigor was recognized by Reviewers ztBx and bUgQ, with ztBx noting our work "provides meaningful insight into the significance of partitioning strategy selection".
> > 2. **Implementation Clarity (Reviewer VhJM)**: We added complete pseudo-code for the context-query partitioning algorithm in Appendix H2.
> > 3. **Baseline Comparison (Reviewers VhJM, 7yXe)**: Following reviewer suggestions, we conducted additional experiments on DiffPuter's tabular benchmarks. **Our method outperforms DiffPuter on 7 out of 8 datasets**, despite DiffPuter requiring 10× iterative training cycles compared to our single-pass training.
> > 4. **Computational Cost (Reviewers ztBx, VhJM)**: We provided comprehensive wall-clock analysis showing training cost comparable to baselines, with inference overhead scaling sub-linearly due to GPU parallelization.
> > 5. **Reviewer bUgQ's Response**: After reviewing our rebuttal, Reviewer bUgQ explicitly stated "My concerns have been adequately addressed" and raised their score from 6 to 8.
> >
> > We believe the theoretical rigor (asymptotic convergence guarantees), strong empirical results (especially in sparse regimes), and the practical importance of learning from incomplete scientific data make this a solid contribution to ICLR.
> >
> > Thank you for your consideration.

---

### Meta-Review · Area_Chair_19Qq · 2026-01-05

**Summary:**

The paper introduces a framework for learning spatio-temporal physical dynamics from incomplete observations. The core idea relies on a masking-based training strategy: incomplete observations are partitioned into a context set and a query set, and training consists of predicting the query portion from the context using a conditional diffusion model. The authors show that this framework allows asymptotic convergence to the true data distribution, even when full observations are unavailable. This result further motivates the proposed context–query partitioning strategy during training. At inference time, the method relies on ensemble sampling to approximate the target conditional expectation. The approach is evaluated on simulated PDE benchmarks as well as on the ERA5 reanalysis dataset.

The reviewers acknowledge the importance of the problem, the theoretical contribution supporting the proposed strategy, and the significant improvements over selected baselines. Two reviewers, however, raise several concerns, including insufficient positioning of the work with respect to related literature (e.g., masking strategies used in self-supervised learning such as masked autoencoders), incomplete baseline comparisons, and a lack of technical detail in the algorithm description.

**Reviewer Concerns:**

The authors provide a detailed rebuttal addressing these points. In particular, they clarify the originality of their contribution with respect to existing masked-learning and related approaches and add new baseline comparisons as suggested by the reviewers. Overall, I consider that the authors’ responses and the corresponding revisions of the manuscript adequately address the main concerns raised during the review process, especially regarding the positioning of the contribution and the experimental evaluation. I am inclined to recommend acceptance.

**Reviewer Scores:**

RztBX, rating 8, would probably keep their rating

RVhjM, rating 4, all questions/ concerns addressed, would probably increase their score

R7yXe, rating 4, all questions/ concerns adressed, would probably increase their score

RbUgQ, rating 6, indicates that all their concerns have been adressed and that they are willing to increase the score to 8.

---

### Decision · Program_Chairs · 2026-01-26

Accept (Poster)